# Estimation of Cloud Condensation Nuclei number concentrations and comparison to in-situ and lidar observations during the HOPE experiments

Christa Genz[1,a,*], Roland Schrödner[1], Bernd Heinold[1], Silvia Henning[1], Holger Baars[1], Gerald Spindler[1], and Ina Tegen[1]

[1]Leibniz Institute for Tropospheric Research (TROPOS), Permoserstraße 15, 04318 Leipzig, Germany
[a]now at German Centre for Integrative Biodiversity Research (iDiv) Halle-Jena-Leipzig, Deutscher Platz 5e, 04317 Leipzig, Germany
[*]previously published under the name Christa Engler
**Correspondence:** christa.genz@idiv.de

**Abstract.** Atmospheric aerosol particles are the precondition for the formation of cloud droplets and therefore have large influence on the microphysical and radiative properties of clouds. In this work four different methods to derive or measure number concentrations of cloud condensation nuclei (CCN) were analyzed and compared for present-day aerosol conditions: (i) A model parameterization based on simulated particle concentrations, (ii) the same parameterization based on gravimetrical particle measurements, (iii) direct CCN measurements with a CCN counter and (iv) lidar-derived and in-situ measured vertical CCN profiles. In order to allow for sensitivity studies of the anthropogenic impact, a scenario to estimate the maximum CCN concentration under peak aerosol conditions of the mid 1980's in Europe was developed as well. In general, the simulations are in good agreement with the observation. At ground level, average values between $0.7 - 1.5 \times 10^9 \, \mathrm{CCN \, m^{-3}}$ at a supersaturation of 0.2 % were found with the different methods under present-day conditions. The discrimination of the chemical species revealed an almost equal contribution of ammonium sulfate and ammonium nitrate to the total number of CCN for present-day conditions. This was not the case for the peak aerosol scenario, in which it was assumed that no ammonium nitrate was formed while large amounts of sulfate present consuming all available ammonia during ammonium sulfate formation. The CCN number concentration at five different supersaturation values has been compared to the measurements. The discrepancies between model and in-situ observations were lowest for the lowest (0.1 %) and highest supersaturations (0.7 %). For supersaturations between 0.3 % and 0.5 %, the model overestimated the potentially activated particle fraction by around 30 %. By comparing the simulation with observed profiles, the vertical distribution of the CCN concentration was found to be overestimated by up to a factor of 2 in the boundary layer. The analysis of the modern (year 2013) and the peak aerosol scenario (expected to be representative for mid 1980's over Europe) resulted in a scaling factor, which was defined as the quotient of the average vertical profile of the peak aerosol and present day CCN concentration. This factor was found to be around 2 close to the ground, increasing to around 3.5 between 2 and 5 km and approaching 1 (i.e., no difference between present day and peak aerosol conditions) with further increasing height.

# 1 Introduction

Compared to today, in the 1980's the anthropogenic emission of aerosols and precursor gases was much higher (Vestreng et al., 2007; Smith et al., 2011). Presumably, during this time the loads of such aerosols over this region were at their maximum. At least since the 1990's anthropogenic emissions of aerosols and precursor gases in Central Europe have been decreasing (e.g.,
Smith et al., 2011).

Atmospheric aerosol particles play an important role in the microphysical processes of cloud formation (Köhler, 1936) and thus have a potentially large influence on cloud properties. However, the evaluation of their effects shows still large uncertainties (e.g., Boucher et al., 2013). In order to reduce those uncertainties, parameterizations to estimate the number concentrations of the CCN have been developed for application in models. For a realistic simulation of microphysical aerosol-
cloud-interactions and macroscopic cloud adjustment due to aerosol perturbations, a detailed representation of the aerosol in the models is required. To describe the activation of aerosol particles, the chemical composition, the number concentration and the size distribution of the aerosol particles have to be known. Parameterizations of the cloud droplet activation (e.g., Abdul-Razzak et al., 1998; Abdul-Razzak and Ghan, 2000; Petters and Kreidenweis, 2007) apply the Köhler-Theory (Köhler, 1936) and have been implemented into regional chemistry transport models (e.g., Bangert et al., 2011; Hande et al., 2016). The
influence of the droplet activation on the aerosol composition is described using the aerosol hygroscopicity, e.g., represented by the hygroscopicity parameter kappa ($\kappa$). These parameterizations enable the investigation of the interaction of the aerosol population with cloud microphysical properties.

For the regional chemistry transport model (CTM) that is used in this study (COSMO-MUSCAT, Wolke et al., 2012, see section 2.1.1), Sudhakar et al. (2017) extended the model system to allow aerosol-cloud-interactions applying the two-moment
cloud microphysics scheme by Seifert and Beheng (2006). This model version is online interactively coupled, making the activation of aerosol mass available for the two-moment scheme. However, the aerosol activation uses the bulk mass and does not explicitly consider online computed aerosol microphysical properties. The complex consideration of aerosols and aerosol-cloud-interactions including the particle size distribution and composition in models is expensive with regard to computation time and storage and thus not feasible in particular for long-term applications.

Therefore, Hande et al. (2016) applied a combination of two existing models to produce a CCN climatology for use in limited-area models, representing normal background conditions over Europe. First, the aerosol particle mass concentrations were simulated using a CTM with a mass-based aerosol scheme. Then, the CCN number concentration was calculated offline using the parametrization of Abdul-Razzak and Ghan (2000), utilizing assumed number size distributions and the modeled chemical composition of the aerosol.

Measurements of the CCN number concentration in the field are valuable in order to evaluate and constrain the ability of the models to describe the activation of aerosol particles. There are several recent studies of in-situ observations (e.g., Henning et al., 2014; Hammer et al., 2014; Friedman et al., 2013). In-situ CCN measurements were already performed in the 50s of the last century and compared to the predicted CCN number (e.g., Twomey and Squires, 1959). Also the influence of the source region and the variation of concentration with height and region has been investigated earlier (e.g., Squires and Twomey, 1966;

Hoppel et al., 1973). Furthermore, the derivation of vertical profiles of CCN with ground-based remote sensing methods is possible (e.g., Ghan et al., 2006; Shinozuka et al., 2015; Mamouri and Ansmann, 2016; Lv et al., 2018) with the development of first approaches in the late 1990's (Feingold et al., 1998). Such data sets can be used to evaluate the application of available aerosol activation parameterizations in atmospheric models. Evaluated against in-situ observations, the applied regional and

global models (e.g., Spracklen et al., 2011; Bègue et al., 2015; Schmale et al., 2019; Fanourgakis et al., 2019; Watson-Parris et al., 2019) tend to underestimate the observed CCN concentrations.

The aim of this study is to provide estimates of the concentrations of cloud condensation nuclei (CCN) representative for the mid 1980's over Germany and compare those to simulations and observations in the year 2013. The derived time varying 3D-CCN fields were used as input for high-resolution simulations over Germany in the framework of the High Definition Clouds

and Precipitation for advancing Climate Prediction (HD(CP)$^2$) project (see Heinze et al., 2017; Costa-Surós et al., 2020). A similar approach as in Hande et al. (2016) was applied to derive CCN from modeled aerosol mass concentrations. The mass concentrations of the aerosol species were simulated using the regional CTM COSMO-MUSCAT with a mass-based aerosol scheme for two periods of the HD(CP)$^2$ Observational Prototype Experiments (HOPE, Macke et al., 2017) in 2013. Based on the modeled aerosol mass concentrations and assumed particle number size distributions for each aerosol species, the CCN

number concentrations were calculated offline using the activation parametrization by Abdul-Razzak and Ghan (2000). The parameterization calculates the number of activated aerosol particles for an aerosol population consisting of multiple lognormal aerosol size distributions and multiple aerosol types. The number of activated aerosol particles depends on the number size distributions of the aerosol population, its chemical composition as well as the applied supersaturation (e.g., fixed or derived from updraft velocities). Thus, this approach is very versatile and can be applied for each type of aerosol mixture. The resulting

modeled CCN fields can be used in atmospheric models that do not treat aerosol transport explicitly to analyze clouds and their radiation effects. For this purpose, CCN fields of a variable degree of complexity can be generated, e.g., temporally and spatially constant CCN profiles, a 3D CCN field as a long-term average or even a 4D CCN field for temporally limited episodes. For the year 2013, the CCN number concentrations derived or measured with four different methods were compared: (i) CCN derived from COSMO-MUSCAT simulations of aerosol mass concentrations, (ii) CCN derived from gravimetrical

aerosol mass measurements, (iii) ground-based in-situ measurements of CCNC, and (iv) vertical CCN profiles derived from ground-based lidar remote sensing and observed by helicopter-borne in-situ measurements.

In order to estimate the CCN concentrations in the mid 1980's over Europe, the aerosol concentrations from the 2013 simulation were scaled based on emission estimates for Germany of the year 1985. The derived CCN fields for the mid 1980's scenario were compared to the 2013 simulation and the observations of the year 2013.

The manuscript is structured as follows. First, the applied CTM COSMO-MUSCAT as well as the different observation techniques are introduced and necessary assumptions are described. In section 3, the results of the comparison of CCN number concentrations obtained from the different methods are discussed. Conclusions and a summary can be found in section 4.

## 2 Methods

### 2.1 Model description

#### 2.1.1 COSMO-MUSCAT

For this study, the chemistry transport model system COSMO-MUSCAT (Wolke et al., 2012) was used. It consists of the
meteorological model COSMO (COnsortium for Small scale MOdelling), which is the operational forecast model of the
German Weather Service (DWD), and the chemistry transport model MUSCAT (MUltiScale Chemistry Aerosol Transport).
COSMO is driven by initial and boundary data from GME re-analysis (the global model of DWD operational in 2013, Majewski
et al., 2002). After a spin-up phase for COSMO of 24 hours, both models run coupled online for 48 hours. To ensure that the
meteorology stays close to the real meteorological conditions, the meteorological fields are then re-initialized for the next
simulation cycle. The trace gas and aerosol fields are kept from the last time step of the previous cycle to ensure a continuous
simulation. The online coupling has the advantage that the meteorological fields from COSMO are forwarded to MUSCAT in
every time step. The meteorological fields drive the chemical transformation and atmospheric transport treated in MUSCAT
for several gas phase and aerosol species. Transport processes include advection, turbulent diffusion, sedimentation, dry and
wet deposition. MUSCAT is based on mass balances, which are described by a system of time-dependent, three-dimensional
advection-diffusion reaction equations. Emissions of anthropogenic primary particles and precursors of secondary aerosols are
prescribed using emission fields from EMEP (European Monitoring and Evaluation Programme, EMEP, 2009). Emissions of
natural primary aerosols (Saharan desert dust, primary marine aerosol particles) are computed within the model (e.g., Heinold
et al., 2011), using meteorological fields (surface wind speed, precipitation) from the model itself in addition to information on
surface properties.

### 2.2 Model setup

The study presented here is part of the High Definition Clouds and Precipitation for advancing Climate Prediction (HD(CP)[2])
project. The main objective is to improve our understanding of clouds and precipitation, using a model for very high resolution
simulations. In the ICON-LEM (ICOsahedral Non-hydrostatic Large Eddy Model; Zängl et al., 2015; Dipankar et al., 2015;
Heinze et al., 2017), which is the model used in HD(CP)[2], there is no online aerosol transport scheme, which indicates the
need of prescribing the aerosol and CCN concentrations in order to be considered for aerosol-cloud interaction.

In order to provide time varying 3D fields of CCN concentrations for ICON-LEM, model simulations with COSMO-
MUSCAT covering most of Germany have been carried out for the time period of two intensive measurement campaigns
during HD(CP)[2]: HOPE. The resulting cloud properties in the ICON-LEM simulation using the derived CCN fields from this
study are analysed and discussed by (Costa-Surós et al., 2020). The HOPE campaigns cover the time periods between April
3 to May 31 and September 1 - 30, 2013 (see section 2.3). Data from the measurement site Melpitz, Germany, were used for
comparison during both campaigns. In addition, lidar-based CCN concentrations were available during the spring campaign in
Jülich, Germany.

The model domain investigated in this study is displayed in Fig. 1 and covers the area between 6-15°E and 48.25-54°N. The horizontal resolution was set to 7 km. In the vertical, the model treats 50 layers up to a height of 22 km. As lateral boundary conditions for the trace gases and aerosol species, modeled fields of atmospheric chemical composition originating from a coarser simulation on a European domain are utilized. This coarser surrounding simulation is driven by reanalysis data for

5   meteorology (reanalysis product of DWD using the GME model) and atmospheric chemical composition (CAMS (Copernicus Atmosphere Monitoring Service) reanalysis product; Inness et al., 2019). The temporal resolution for the model output was set to 1h. Besides the standard meteorological model output from COSMO, MUSCAT provides the mass concentrations of several gas phase and aerosol species.

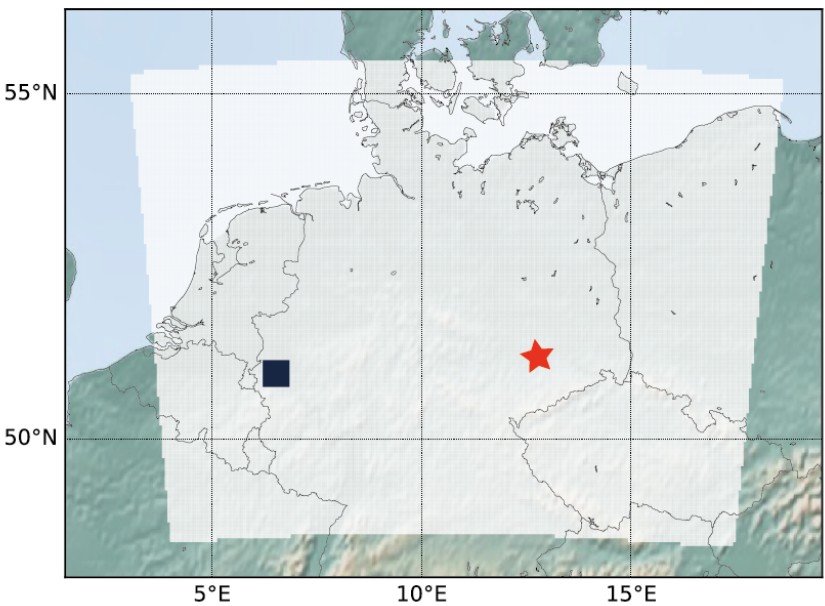

**Figure 1.** Model domain over Germany, which was used in this study (white area). The red star marks the research station Melpitz (12.93°E, 51.53°N) and the blue square the measurement site Jülich (50.88°N, 6.41°E).

### 2.2.1   Aerosol particle number estimation and CCN parametrization

10   Using the aerosol bulk scheme of COSMO-MUSCAT, the mass concentrations for the species considered are simulated. In order to compare the model results with in-situ particle measurements and to calculate number concentrations of CCN, particle number size distributions (PNSD) have to be estimated from those mass concentrations. For each species of the anthropogenic aerosol (ammonium sulfate (AS), ammonium nitrate (AN), sulfate (SU), organic (OC) and elemental carbon (EC)) and sea salt

(SS), individual log-normal size distributions are assumed. The size distribution of the mineral dust (DU) particles follows a sectional scheme (Heinold et al., 2011). A log-normal size distribution is explicitly defined with the three parameters diameter or radius ($d$ or $r$, respectively), standard deviation ($\sigma$) and total number concentration ($N$). For the externally mixed aerosols, the total number concentration of each species is calculated from the modeled mass of the aerosol species assuming an individ-

ual geometric mean radius and standard deviation. The choice of these parameters defines the aerosol number size distribution and is a critical source for uncertainty of aerosol and CCN number concentrations. Within the HD(CP)$^2$ framework, literature values, aerosol mass spectrometer (AMS) measurements, and particle number size distribution measurements in the diameter range 10 nm to 10 μm from the TROPOS site Melpitz, Germany (Poulain et al., 2011), which is representative for central Europe (e.g., Spindler et al., 2012; Engler et al., 2007), were used to define the parameters for the log-normal distributions. Adding

up the different size distributions of all considered species gives the total particle number size distributions. The calculations have been compared to observational data and showed a good agreement to the observed total size distribution at Melpitz between 50 and 200 nm (Hande et al., 2016), which is a very relevant size range for estimating CCN in the supersaturation range investigated in this study (0.1-0.7 %). The geometric mean radius, standard deviation and density for characterizing the particle number size distributions of the individual aerosol species is listed in Tab. 1, mostly according to the values used in

Hande et al. (2016).

The number size distributions of the aerosol species was now used to calculate the number of activated particles under certain conditions. The calculation of the CCN number concentration in this study follows the parameterization of Abdul-Razzak and Ghan (2000) for multi-modal aerosol distributions, which relates the particle number size distribution and composition to the number of activated particles as a function of supersaturation. The individual aerosols compete for the available liquid water,

determining the maximum supersaturation, which apart from the aerosol composition and individual size distributions depends on the updraft velocity. Abdul-Razzak et al. (1998) describe the parameterization for a single log-normal mode of aerosol particles (only for a single species), whereas Abdul-Razzak and Ghan (2000) developed an extended approach for multiple soluble and insoluble aerosol species, representing a multi-modal aerosol size distribution. The parameterization uses the hygroscopicity parameter $\kappa$ of each considered aerosol species. The $\kappa$ values used in this study can be found in Tab. 1 as well.

$\kappa$ was defined first in Petters and Kreidenweis (2007) as a single parameter to describe the relationship between the particle dry diameter, its hygroscopicity, and the CCN activation. In several laboratory studies, $\kappa$ has been determined experimentally. Highly hygroscopic particles can have a $\kappa > 1$, while for totally hydrophobic particles $\kappa = 0$. Petters and Kreidenweis (2007) reported $\kappa$ for a number of different compounds, e.g., ammonium sulfate being about 0.6 in the supersaturation regime. Further studies investigated $\kappa$ for other substances like sea salt (e.g., Niedermeier et al., 2008), coated soot (e.g., Henning et al., 2010)

and secondary organic aerosol (e.g., Wex et al., 2009; Duplissy et al., 2011) or in dependence on the mixing state of the particles (Wex et al., 2010).

The same size distributions as applied in this study to derive CCN concentrations from the modeled aerosol mass were utilized in a related study of the HD(CP)$^2$ project by Hande et al. (2016). They evaluated the aerosol size distribution at Melpitz and found good agreement in the size range between 50-200 nm. We therefore assume that the applied method generally

produces realistic CCN concentrations since the critical size of activation usually falls within this range for the supersatura-

**Table 1.** Physical and chemical aerosol properties used in this study. The values for the particle radius and standard deviation of the size distribution follow Poulain et al. (2011), Spindler et al. (2012) (non-dust species) and Heinold et al. (2011) (mineral dust). Several laboratory and model studies served as basis for the $\kappa$ values used in this study (ammonium sulfate: Ghan et al. (2001), Petters and Kreidenweis (2007); ammonium nitrate: Duplissy et al. (2011); sulfate: Petters and Kreidenweis (2007); OC: Ghan et al. (2001), Wex et al. (2009); sea salt, dust, and EC: Ghan et al. (2001)).

| Species | $\kappa$ | $\sigma$ | r (µm) | $\rho$ (kgm$^{-3}$) |
|---|---|---|---|---|
| Ammonium sulfate | 0.51 | 1.6 | 0.05 | 1.77 |
| Ammonium nitrate | 0.54 | 1.6 | 0.05 | 1.725 |
| Sulfate | 1 | 1.6 | 0.05 | 1.8 |
| Sea salt 1 | 1.16 | 1.8 | 0.065 | 2.2 |
| Sea salt 2 | 1.16 | 1.7 | 0.645 | 2.2 |
| EC | $5 \times 10^{-7}$ | 1.8 | 0.03 | 1.8 |
| OC | 0.14 | 1.8 | 0.055 | 1.0 |
| Mineral dust 1 | 0.14 | 2.0 | 0.2 | 2.65 |
| Mineral dust 2 | 0.14 | 2.0 | 0.6 | 2.65 |
| Mineral dust 3 | 0.14 | 2.0 | 1.75 | 2.65 |
| Mineral dust 4 | 0.14 | 2.0 | 5.25 | 2.65 |
| Mineral dust 5 | 0.14 | 2.0 | 15.95 | 2.65 |

tions applied in this study and aerosol particles in this range are usually more numerous than larger particles. However, the ambient aerosol size distribution varies in time and space and therefore the assumption of a spatially and temporally constant size distribution for the different aerosol species is a source of uncertainty. For the example of $1\,\mu\,\mathrm{gm}^{-3}$ ammonium sulfate aerosol, using the assumptions given in Tab. 1, the number of activated aerosols is $215\,\mathrm{cm}^{-3}$ at supersaturation of 0.2 %. By varying the geometric mean radius of the assumed size distribution by +/- 10 %, but keeping the total mass constant, the CCN concentration varies by +/- ~15 %. Widening the distribution using $\sigma$=1.7 instead of 1.6 leads to decrease of ~25 % in CCN number concentration at 0.2 % supersaturation since the total particle number decreases due to more large particles that are large in volume.

In order to evaluate these assumptions, the modeled CCN number concentrations were compared to measurements close to the ground for the TROPOS super-site Melpitz. For this purpose, the same supersaturations as applied in the CCN number concentration measurements with a cloud condensation nucleus counter (CCNC, Henning et al., 2014) were applied to the simulated particle number size distributions (see section 2.3.1).

**Table 2.** Annual emissions of dust, sulfur dioxide and ammonia for entire Germany for the years 1985 and 2013 in $\mathrm{Mt}$ as provided by Umweltbundesamt (German Federal Environmental Agency, UBA, Hausmann, 2017, personal communcation). So called dust also includes e.g., soot and resuspended material besides the natural mineral dust. The table also includes the factors, by which the concentrations in 2013 are scaled in order to estimate the concentrations for the mid 1980's scenario.

|  | 1985 | 2013 | ratio 1985/2013 |
| --- | --- | --- | --- |
| dust (incl. soot) | 2.65 | 0.35 | 7.7 |
| $SO_2$ | 7.73 | 0.41 | 19 |
| $NH_3$ | 0.86 | 0.74 | 1.2 |

### 2.2.2 Estimation of peak aerosol in the mid 1980's

In order to allow for sensitivity studies on the impact of anthropogenic pollution on CCN concentrations, a scenario to estimate aerosol concentrations over Central Europe in the mid 1980's was developed. Due to the maximum emissions of aerosols and precursor gases in Europe during the 1980's, the year 1985 was taken as a reference year for the emissions. In the early 1990s, environmental protection became much more important and efficient emission reduction strategies were developed. Furthermore, many aerosol and precursor sources simply disappeared after the liquidation of several industry sites in Eastern Germany and the former East-bloc countries after the political change in 1990.

The calculations for the mid 1980's were carried out offline with the model run from 2013 as a basis. The annual emissions of sulfur dioxide and ammonia during the years 1985 and 2013 (see Tab. 2) were utilized for these estimations (Hausmann, 2017, Umweltbundesamt (UBA, German Federal Environmental Agency), personal communication). The scaling factors derived in order to estimate the aerosol concentrations for the mid 1980's scenario based on the present day simulation are summarized in Tab. 3. The model implementation of the formation of ammonium sulfate ($(NH_4)_2SO_4$) and ammonium nitrate ($NH_4NO_3$) is described by Hinneburg et al. (2009) and follows Simpson et al. (2003). Particulate ammonium sulfate can be formed in the atmosphere from sulfuric acid (formed after oxidation of $SO_2$) and ammonia. In the model, first ammonium sulfate is formed until either ammonia or sulfuric acid is consumed. In case there is still ammonia left after this reaction, ammonium nitrate can be formed as well. As can be seen from Tab. 2, almost 20 times more $SO_2$ was emitted in Germany during the 1980's compared to 2013, whereas $NH_3$ emissions remained almost unchanged. For this reason, there was much more sulfuric acid available in the atmosphere than necessary for the transformation of the total available ammonia to ammonium sulfate. In 2013, $SO_2$ and $NH_3$ react to ammonium sulfate until $SO_2$ is consumed leading to the formation of $0.85\,\mathrm{Mt}$ ammonium sulfate. For the mid 1980's conditions, in the implemented scheme, first $NH_3$ is consumed and in total $3.32\,\mathrm{Mt}$ ammonium sulfate are formed. This results in a scaling factor for ammonium sulfate of 3.9. In this $SO_2$ limited regime in 2013, there would not be any $NH_3$ left to produce ammonium nitrate. The inhomogeneous distribution and the time-dependent formation would still enable nitrate formation in reality. However, since assumed density, size distribution and hygroscopicity of ammonium sulfate

**Table 3.** Assumptions for the estimation of the aerosol conditions for the 1980's over Germany.

|  | 2013 | mid 1980's scenario |
|---|---|---|
| Ammonium sulfate | $AS_{2013}$ | $AS_{2013} \cdot 3.9$ |
| Ammonium nitrate | $AN_{2013}$ | 0 |
| Sulfate | $SU_{2013}$ | $AS_{2013} \cdot 5.3$ |
| EC | $EC_{2013}$ | $EC_{2013} \cdot 2$ |
| OC | $OC_{2013}$ | $OC_{2013}$ |
| Sea salt | $SS_{2013}$ | $SS_{2013}$ |
| Mineral dust | $DU_{2013}$ | $DU_{2013}$ |

and ammonium nitrate are similar, exchanging part of the ammonium sulfate with ammonium nitrate and vice versa would not introduce strong changes to the calculated CCN number concentration, which is the aim of this study. This is why, the production of ammonium nitrate was set to zero for the mid 1980's scenario. The ammonium sulfate formation leaves $6.1\,\mathrm{Mt}$ $SO_2$ unconsumed. Half of this excess $SO_2$ left after ammonium sulfate formation in the mid 1980's is assumed to be oxidized

to sulfuric acid. Sulfuric acid is assumed to entirely partition to the particulate phase and is therefore accounted for as sulfate. The approach described above is also encouraged by the serious acid rain problem in the 1980's (e.g., Seinfeld and Pandis, 1998, p. 1030ff). Since no excess sulfate is present in the 2013 simulation, we calculate the sulfate concentration for the mid 1980's scenario based on the 2013 ammonium sulfate concentration. The ratio between the formed sulfate in the mid 1980's scenario ($4.68\,\mathrm{Mt}$) and the formed ammonium sulfate in 2013 ($0.85\,\mathrm{Mt}$) results in a scaling factor of 5.3. Since no emission

data for elemental carbon for the 1980's were available, the particle concentrations were assumed to be twice as high as in 2013. This is only justified by the fact that aerosol concentrations in the 1980's over Central Europe were higher than today mainly caused by combustion processes for heating and energy production. Organic carbon, sea salt and dust are supposed to result mostly from natural sources and thus remain unchanged for the mid 1980's scenario.

Due to lack of observational data of aerosol size distributions in the 1980's in the study region to generalize size distributions

during this time, for this study the same size distributions for the mid 1980's scenario and 2013 were assumed. Since the size distribution is crucial in order to translate modeled aerosol mass into particle numbers and finally derive CCN numbers, this assumption is likely an important source of uncertainty, which is difficult to quantify reliably.

The above scaling approach, instead of conducting actual simulations for the 1980's, implies that any observed differences between mid 1980's and 2013 aerosol and CCN concentrations are due to changes in the emissions only and are not caused by

differences in meteorological conditions. However, the results have to be interpreted carefully and can only represent a rough, general estimate of the mid 1980's conditions, and hence are not representative for the specific conditions of a particular year or period of the 1980's. The results of the comparison of the number concentrations in 2013 and the mid 1980's are presented in section 3.

## 2.3 Measurements during HOPE

The present study utilizes observational data from the extensive measurements conducted during the two HOPE campaigns (April 3 to May 31 and September 1 - 30, 2013) at the TROPOS research station Melpitz and the measurement site near Jülich, Germany. At Melpitz additonal long-term measurements of in-situ aerosol PNSD, CCN concentrations and chemical composition of the aerosol particles are available. The rural-background site Melpitz (12.93°E, 51.53°N, 86 m a.s.l.) is located in Germany, ~40 km east of Leipzig in the East German lowlands. The site at a meadow is surrounded by agricultural land. It is representative for a large area in Central Europe and long-term studies with consideration of marine or continental air mass inflow enables the investigation of the influence of different spatially distributed emission sources and long-range transport on particulate matter (PM) concentrations (Engler et al., 2007; Spindler et al., 2013). The Melpitz site is integrated in the infrastructure network ACTRIS (Aerosols, Clouds, and Trace gases Research Infrastructure Network, www.actris.eu) and EMEP (Co-operative Programme for Monitoring and Evaluation of the Long-Range Transmission of Air Pollutants in Europe Tørseth et al., 2012). From the spring campaign at Jülich only the lidar measurements were used to derive vertical profiles of CCN concentrations.

The idea behind the HOPE campaigns was to gain a comprehensive dataset of observations for evaluation of the new German operational forecast model ICON at the scale of a couple hundred meters (ICON-LEM). The campaign focused on the convective atmospheric boundary layer, especially the connection of clouds and precipitation. Technically, HOPE aimed at combining most of the surface flux and mobile ground-based remote-sensing observations available in Germany within a single domain for the purpose of describing the vertical structure and horizontal variability of wind, temperature, humidity, aerosol particles and cloud droplets in a high temporal and spatial resolution.

Additionally, during the fall campaign, in-situ observations with the helicopter-borne platform ACTOS (Airborne Cloud Turbulence Observation System, Siebert et al., 2006) were combined with aerosol and cloud properties observed with remote sensing at the LACROS (Leipzig Aerosol and Cloud Remote Observations System, Bühl et al., 2013) supersite. This dataset allows for the investigation of the relationship between tropospheric clouds and aerosol conditions.

Detailed information on the meteorological conditions during the two campaigns can be found in Macke et al. (2017), Tab. 3 and 4. The weather situations during the spring campaign changed from a few high-pressure systems with high-level cirrus clouds, interrupted by several frontal passages (warm and cold fronts) at the beginning of the campaign, and followed by more shallow convective clouds later on. The fall period was dominated by low-level overcast clouds.

### 2.3.1 In-situ CCNC measurements - ground-based and airborne

Ground-based in-situ measurements with the CCNC are operational in Melpitz since August 2012 (Schmale et al., 2017) and the results were available for model evaluations within this study. The ambient CCN number concentration at Melpitz station was determined by means of size segregated activation measurements as described in detail in Henning et al. (2014), following the ACTRIS SOP (standard operating procedures, Gysel and Stratmann, 2013). Briefly, the set-up is as follows, downstream of the aerosol inlet and the drier unit, an aerosol flow of $1.5 \, \mathrm{L \, min^{-1}}$ is size-selected with a DMPS system (Differential

Mobility Particle Sizing system) and afterwards divided between a condensation particle counter ($1\,\mathrm{L\,min^{-1}}$ working flow, CPC 3010, TSI Aachen Germany) and a cloud condensation nucleus counter ($0.5\,\mathrm{L\,min^{-1}}$ working flow, CCNC, CCN-100, Boulder, USA). With the CCNC, a stream-wise thermal gradient cloud condensation nucleus counter (Roberts and Nenes, 2005), the supersaturation-dependent activation of the particles is investigated at 0.1, 0.2, 0.3, 0.5, 0.7 and 1 % supersaturation.

The ratio between the CCN number and the total particle number as counted by the CPC (condensation nuclei, CN) gives the activated fraction (AF) of the particles. The AF was corrected for multiply charged particles up to three charges by subtracting their apparent fraction from the AF using the charge equilibrium (Wiedensohler, 1988). This multiple charge corrected AF is calculated for each particle diameter and results in a size dependent activation curve for each supersaturation. This curve is fitted with a sigmoidal function describing the activation curve with the four parameters lower activation limit, upper limit, sigma ($\sigma$)

and the critical diameter ($D_c$). Multiplying the activation curve (CCN/CN) with the ambient size distribution integral results in the ambient CCN number concentration at the given supersaturation. One measurement per supersaturation is available every two hours.

During the fall measurement campaign of HOPE also the helicopter-borne measurement platform ACTOS was deployed in Melpitz. The experimental set-up and the flight characteristics are described in detail by Düsing et al. (2018). Within this study

we use the vertically resolved in-situ data of the light weight mini cloud condensation nuclei counter (mCCNc, custom built by Gregory C. Roberts, working principal as described by Roberts and Nenes, 2005), which has been applied successfully on ACTOS before (e.g., Wex et al., 2016). The miniCCNc measured the CCN number concentration at a supersaturation of 0.2 %. Vertical profile measurements are available for 8 flights between September 12 - 27, 2013.

### 2.3.2 Daily PM$_{10}$ sampling at Melpitz site

Particles with aerodynamic diameter up to $10\,\mathrm{\mu m}$ (PM$_{10}$) were sampled daily at the Melpitz site. PM-High-Volume quartz filter samples for PM$_{10}$ were collected using a High-Volume sampler (DIGITEL DHA-80, Walter Riemer Messtechnik, Germany), having a sampling flux of about $30\,\mathrm{m^3 h^{-1}}$. The filter type is a MK 360 quartz fibre filter (Munktell, Grycksbo, Sweden). The measurement techniques to determine the particle mass, water soluble ions and carbonaceous particles are described by Spindler et al. (2013, 2012). The particle mass determination was performed gravimetrically. The conditioned filters (72 hours

at 20°C and 50 % relative humidity) were weighted with a microbalance as tare (blank) and after sampling of particles as gross weight. Main water-soluble ions ($NO^{3-}$, $SO_4^{2-}$, $Cl^-$, $Na^+$, $NH^{4+}$, $K^+$, $Mg^{2+}$, $Ca^{2+}$) were analyzed by ion chromatography. The determination of organic and elemental carbon (OC and EC) was performed by a two-step thermographic method using a carbon analyzer (behr Labor-Technik, Germany). OC was vaporized at 650°C for 8 minutes under nitrogen atmosphere and catalytically converted to $CO_2$ and the remaining EC was combusted further in 8 minutes with $O_2$ at 650°C. The formed $CO_2$

was than quantitatively determined by a non-dispersive infrared detector (modified German standard VDI method 2465 part 2).

### 2.3.3 CCN concentrations derived by lidar measurements

During the HOPE campaigns, PollyXT lidar systems (Engelmann et al., 2016) were used to measure automatically and continuously the vertical state of the atmosphere in terms of aerosol particles and clouds. Lidar observations were performed in Melpitz (fall campaign) and Jülich (spring campaign) with the 12 channel-multiwavelength-polarization lidar PollyXT_OCEANET. Hourly averaged profiles of the particle backscatter and extinction coefficient as well as the particle depolarization ratio were calculated automatically for the whole measurement period as described in Baars et al. (2016). As the particle depolarization ratio was close to zero (indicator for spherical particles) for the whole period, one can conclude that no dust intrusion was occurring during the intensive field campaigns. Thus, the CCN concentration profiles were calculated following the continental aerosol branch in Mamouri and Ansmann (2016).

For this approach, the lidar-derived particle backscatter profiles are converted to extinction profiles by using a lidar ratio of 50 sr as a typical value for continental sites (Baars et al., 2017). The aerosol number concentration profiles for particles with a dry radius > 50 nm ($n_{50}$) are calculated using

$$n_{50,c,dry}(z) = c_{60,c}\sigma_c^{X_c}(z)$$

with $c_{60,c}$=25.3 cm$^{-3}$ and $X_c$=0.94 (see Mamouri and Ansmann (2016) for details). Finally, the CCN concentration at supersaturations < 0.2 % is estimated by multiplying $n_{50}$ with an enhancement factor of f = 1. The uncertainty of this estimation is at a factor of 2-3 according to Mamouri and Ansmann (2016).

## 3 Results

### 3.1 Aerosol optical thickness

The simulations described in this work were also evaluated by Costa-Surós et al. (2020). They present a comparison of aerosol optical thickness (AOT) over the North and Baltic Sea as observed by the AVHRR (Advanced Very High Resolution Radiometer) instrument onboard of different NOAA satellites and modeled by COSMO-MUSCAT. The observational platform represents a good opportunity to evaluate the modeled aerosol load for both 2013 and the 1980's conditions since the data set dates back to 1981. The modeled and observed AOT were shown to agree well for both the 2013 period and the mid 1980's conditions (using the observational example of the year 1985). The observed median AOT values over the Baltic sea for 1985 and 2013 were 0.30 and 0.14, respectively, and over the North Sea 0.25 and 0.14. The modeled values were 0.30 and 0.11 for the Baltic Sea and 0.22 and 0.09 for the Baltic Sea. It can therefore be concluded that the model, using the assumptions discussed in this work, is able to represent the average aerosol loads of 2013 and particularly the 1980's.

### 3.2 Composition of CCN

As described above, number concentrations of CCN over Germany for two time periods of the year 2013 have been calculated offline from aerosol particle number concentrations based on simulated mass concentrations of 7 different compounds: am-

**Table 4.** Average CCN number concentration ($\mathrm{m}^{-3}$) and average contribution (%) of the considered species to the total CCN number concentration at ground level for a supersaturation of $0.2\,\%$ at the HOPE site Melpitz for the two 2013 campaigns and the corresponding periods of the mid 1980's scenario. The values were calculated from aerosol mass concentrations modeled with COSMO-MUSCAT and from aerosol mass concentrations observed by gravimetrical measurements. In addition, the average in-situ measured CCN number concentration is shown for comparison.

| Data base / scenario | $N\_CCN_{0.2\%}$, $\mathrm{m}^{-3}$ | AS | AN | SU | EC | OC | SS | DU |
|---|---|---|---|---|---|---|---|---|
| Modeled aerosol mass concentrations (mid 1980's) | $5.2 \times 10^9$ | 36 | 0 | 64 | 0 | 0.4 | 0.3 | 0.001 |
| Modeled aerosol mass concentrations (2013) | $9.4 \times 10^8$ | 51 | 46 | 0.007 | 0 | 2.3 | 1.6 | 0.008 |
| Measured aerosol mass concentrations (2013) | $1.5 \times 10^9$ | 35 | 53 | 0 | 0 | 7.4 | 0.3 | 4.0 |
| Direct observation of CCN with CCNC (2013) | $1.1 \times 10^9$ | | | | | | | |

monium sulfate, ammonium nitrate, sulfate, organic and elemental carbon, sea salt and mineral dust. Similarly, representing a peak aerosol scenario over Europe, aerosol concentrations have been calculated representative for the mid 1980's based on the simulations for the year 2013 (see section 2.2.2). Furthermore, the CCN parameterization has been applied to observed particle mass concentrations. The modeled CCN number concentrations were compared to ground-based in-situ measurements

by a CCNC, and to vertical profiles derived from lidar and helicopter-borne in-situ observations. Table 4 lists the total number concentration of CCN and the contribution of the individual compounds as average values for the simulated time period. Nowadays, the contribution of ammonium nitrate and ammonium sulfate are almost balanced. Due to the assumption that ammonium nitrate was not formed in the mid 1980's scenario, there is no contribution from ammonium nitrate to CCN in this time period. The concentration of ammonium sulfate in the atmosphere was far higher than today (see also section 2.2.2), resulting

in almost no ammonia being available for the formation of ammonium nitrate. Instead, much more sulfuric acid could form during this time period.

Comparing the two different methods of estimating todays CCN concentrations, differences can be seen especially for ammonium sulfate, organic carbon and mineral dust. The dust concentrations resulting from the gravimetrical methods are usually higher than simulated, because they result from the difference of the total gravimetric mass and the sum of the masses

of the individual species and are not directly measured. This is why the error is quite large due to losses of the other species during the analytical processes. Furthermore, they may contain other undetected material than only mineral dust and also re-emitted soil dust, which is not included in the emission data used in the model simulations. The difference in CCN from OC is partly due to the absence of secondary organic aerosol (SOA) in the model approach. SOA generally can contribute a large fraction to the total concentration of organic aerosol mass with an average contribution over Europe ranging from ~20 to more

than $50\,\%$ (Jimenez et al., 2009). Also at Melpitz in summer, organic matter is the major fraction (59 %) of the $PM_1$ aerosol and is strongly influenced by SOA (Poulain et al., 2011).

Fig. 2 shows the time series of derived CCN from the model simulation (upper panel) and from gravimetrical aerosol measurements (lower panel) for both the spring and fall period in comparison to the CCNC measurements at a supersaturation of 0.2 %. The same plot for a supersaturation of 0.3 % is shown in Fig. A1 in the supplement. On average (see Tab. 4), CCN concentrations derived from modeled and observed aerosol mass deviate from the CCNC measurements by a factor of around 1.2 (16 % underestimation) and 1.4 (37 % overestimation), respectively. Taking into account the uncertainty due to assumptions in converting observed or modeled aerosol mass into number, the used CCN parametrization is concluded to work reasonably well on average. However, as can be seen in Fig. 2a and b the performance of the model differs between the two time periods with the tendency to underestimate CCN concentrations in the spring period and overestimate it in the fall period (see also Fig. 3). In order to evaluate the applied method of deriving CCN concentrations from aerosol mass and its assumptions, the size distributions and the activation parameterization are applied to gravimetrical measurements of aerosol mass (Fig. 2c and d). The CCN concentrations derived from the gravimetrical measurements catch better the peaks in the first half of the spring episode and do not show the strong under- and overestimation, respectively, for the two periods as seen for CCN derived from the modeled aerosol mass. Since the activation parameterization is applied to both the modeled and observed aerosol mass, differences in the derived CCN concentrations between the upper and the lower panel of Fig. 2 correspond directly to uncertainties in the actual aerosol simulation with the atmospheric transport model. Particularly in the first half of the spring episode, ammonium nitrate and ammonium sulfate concentrations and thus their contribution to the CCN number concentration were clearly underestimated (see also Hande et al. (2016) Fig. 2). However, during the fall period the model often overestimates the concentration of ammonium sulfate and ammonium nitrate and hence the CCN concentrations. In particular ammonium nitrate is sometimes strongly overestimated by up to a factor of 5. Deviations in ammonium nitrate might arise due to uncertainties of both modeling and observation. The emission of ammonia is depending on agricultural activity (e.g., manuring). Hence, the magnitude and timing of observed ammonium nitrate concentration peaks cannot be represented by the model, which uses monthly emission estimates. Since nitrate is volatile, high temperature within the sampling unit can lead to partial evaporation from the filters.

An interesting episode occurred between day-of-year (doy) 255 and 257 (September 12-14, 2013) in the fall period, resulting in clearly overestimated CCN number concentrations in the model. This was caused by a small-scale low pressure system, which moved south-eastward over the measurement station doy 255 and then continued eastward. The location of this low pressure system was not correctly simulated and the corresponding precipitation in North-western Germany and The Netherlands on doy 254 and 255 was underestimated. This region represents one of the main ammonia sources and hence is important for the formation of ammonium nitrate and sulfate in the atmosphere. Due to the lack of precipitation in this region, wet deposition of aerosol particles and precursors was missing, resulting in an overestimation of aerosol mass concentration and hence CCN number concentration. The airmass rich in particularly ammonium nitrate travelled during doy 254-25 towards the measurement site at Melpitz. The underestimated wet deposition represents a likely cause for the overestimation seen during the three days. However, other potential causes such as wrong emission or overestimated formation cannot be ruled out. Since also the gravimetrical observations show a strong peak during these three days, it can be concluded that the overall situation (emission, formation, transport) is still model reasonably.

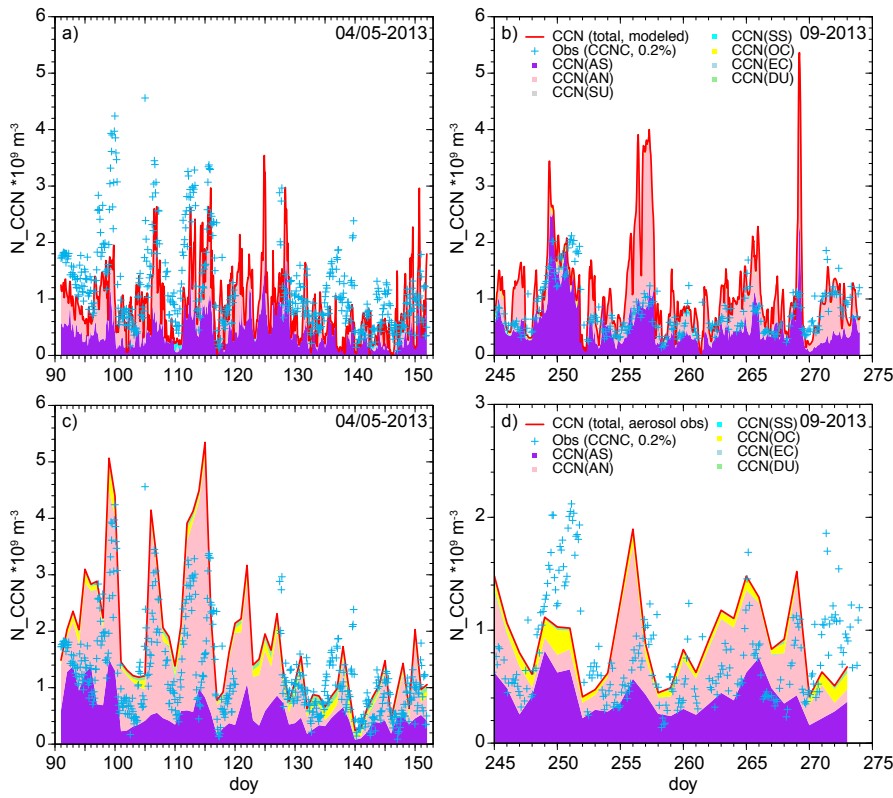

**Figure 2.** Simulated and measured CCN number concentrations in Melpitz at a supersaturation of 0.2 % during the two HOPE campaigns (April to May and September 2013). The upper panel (a and b) shows the CCN number concentrations resulting from the simulated aerosol concentrations, the lower one (c and d) the CCN numbers resulting from measured aerosol concentrations using the same CCN parametrization. The colors represent the contributions to CCN of different species. The blue crosses indicate the CCN number concentrations using the CCNC. Please note the different time resolution for the observations, as well as the different scale for the CCN number concentration in plot d.

## 3.3 Comparison to in-situ CCN measurements

For a more evident comparison of the absolute CCN number concentrations, Fig. 3 displays the derived and measured CCN number concentrations at a supersaturation of 0.2 % as a scatter plot for both episodes. As already seen in the time series plots in Fig. 2, the model tends to underestimate the CCN numbers of the in-situ CCN measurements in the spring episode (on average by 29 %). For the fall episode, an overestimation of 37 % was found (20 % without the outliers of the two days discussed above). In contrast, the CCN number concentration estimated from the gravimetrically measured aerosol masses tends to overestimate the direct measurements in both periods (50 % in spring, 15 % in fall). Together, Figs. 2, 3, and A1 show that the model underestimates the observed CCN concentration at least partly due to an underestimation of aerosol mass,

mainly ammonium nitrate and ammonium sulfate, in the spring episode and overestimates the CCN concentration because of an overestimation of these aerosol species in the fall episode.

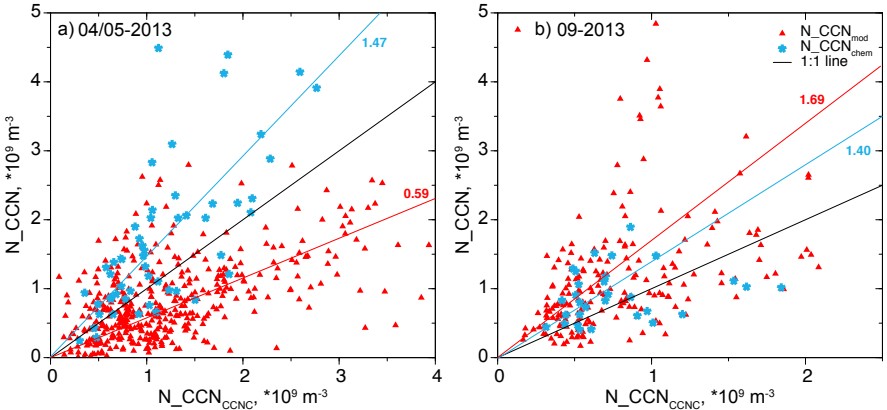

**Figure 3.** Comparison of derived and measured CCN number concentrations in Melpitz at a supersaturation of 0.2%. Red triangles show results from the aerosol simulations, blue stars from applying the CCN parameterization to the gravimetrically measured aerosol mass concentrations. The colored lines are the linear regressions. The slope of the fits are given at the regression lines.

In Fig. 4 the ratio of the number concentrations of CCN (N_CCN) and the total aerosol particles (N_CN) larger than a certain size is shown as comparison between simulation and observation. The upper panels display the fractions for a supersaturation of 0.2 % and particles larger than 110 nm for both episodes, the lower panels for a supersaturation of 0.3 % and particles larger than 80 nm, respectively. A ratio of exactly 1.0 means, that as many particles would activate at the respective supersaturation as aerosol particles with a diameter larger than the threshold diameter of 110 nm ($N\_CN_{110nm}$) and 80 nm ($N\_CN_{80nm}$), respectively, are present in the atmosphere at this time. For the rural observation site Melpitz, this ratio is usually close to 1.0 for 0.2 % and 110 nm, as well as 0.3 % and 80 nm (S. Henning, 2017, personal communication), which is why these two size threshold values were chosen. The $N\_CCN_{0.2\%}$ to $N\_CN_{110nm}$ ratios compare very well (on average 1.03 (observation) and 0.98 (model), respectively), but the model tends to overestimate the $N\_CCN_{0.3\%}$ to $N\_CN_{80nm}$ ratios for both episodes (on average, 0.93 (observation) and 1.26 (model), respectively). This can be the result of the model either overestimating the CCN concentration or underestimating the aerosol particle number in the size range larger than 80 nm in diameter. For both 0.2 % and 0.3 % supersaturation, the model underestimates the CCN concentration in total for both periods by a similar magnitude of 13 and 11 %, respectively (see also Figs. 2 and A1). The size distributions used to convert modeled aerosol mass to number were developed with data at Melpitz. They are able to represent the average total particle number concentration at 100 nm (Fig. 3c in Hande et al., 2016). For particles larger than 110 nm, the observed total particle number concentrations is underestimated by  10 %. However, the number concentration of particles larger than 80 nm is underestimated by  35 %. Hence, the underestimated modeled number concentration of aerosol particles in the size range between 80 and 110 nm in diameter is likely the main reason for the different behavior between the $N\_CCN_{0.2\%}$ to $N\_CN_{110nm}$ ratio and the $N\_CCN_{0.3\%}$ to $N\_CN_{80nm}$ ratio. From Fig. 4, it can be seen that there is no difference in the comparison to the observation between the spring

and the fall episode. Hence, the different under- and overestimation of the CCN concentration between the spring and the fall episode seen in Figs. 2, 3, and A1 is more likely linked to uncertainties of the modeled aerosol mass than the assumptions made to derive CCN.

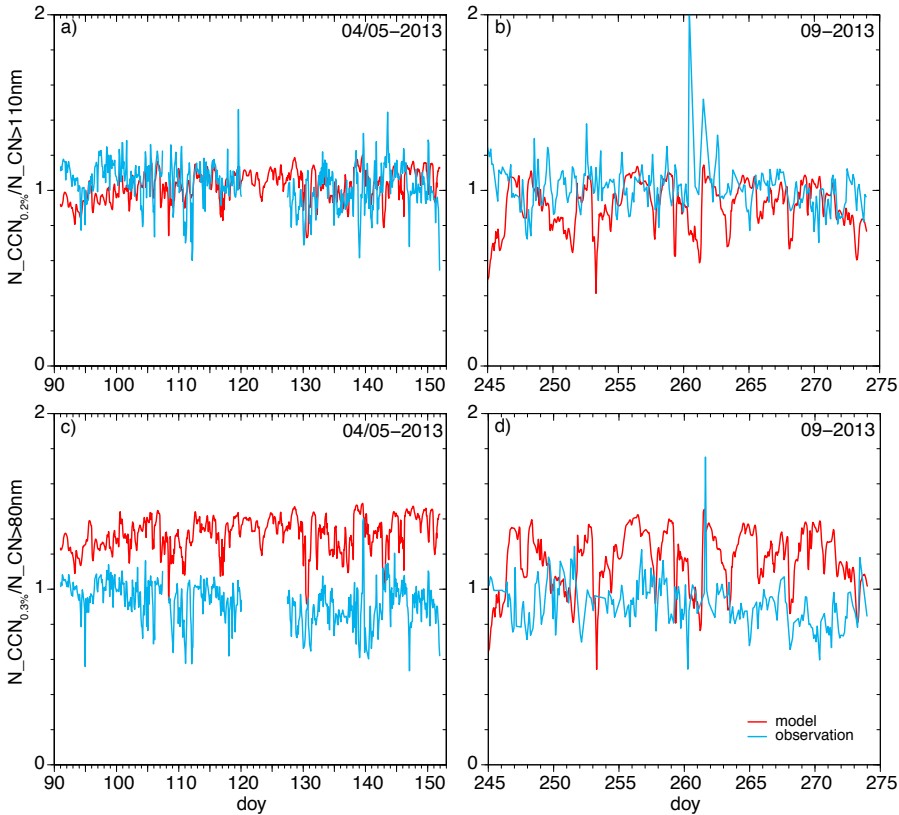

**Figure 4.** Comparison of the modeled and observed activated fraction (N_CCN / N_CN) at a supersaturation of 0.2 % (a and b) and 0.3 % (c and d), respectively. As number of total CN, the number concentration of CN > 110 nm (a and b) and > 80 nm (c and d), respectively, was used.

Fig. 5 shows the average N_CCN-to-N_CN ratio for five different supersaturations between 0.1 and 0.7 % for a cut-off diameter of 40 nm. It can be seen from this graph, that at a low supersaturation of 0.1 %, only very few particles activate, whereas almost all particles activate at a high supersaturation of 0.7 %. In the model, more of the available aerosol particles activate at the respective supersaturation, which is most pronounced in the medium range of supersaturations between 0.3 % and 0.5 %. The assumed size distributions are known to lack of particles much smaller and much larger than 100 nm. Hence, for very low and high supersaturations, both the number of particles and the CCN concentration are underestimated similarly. For supersaturations in between, for which the critical size of activation is in the size range where the assumed size distribution matches the average observations quite well (i.e., around 100 nm), the modeled CCN concentration is less underestimated on average.

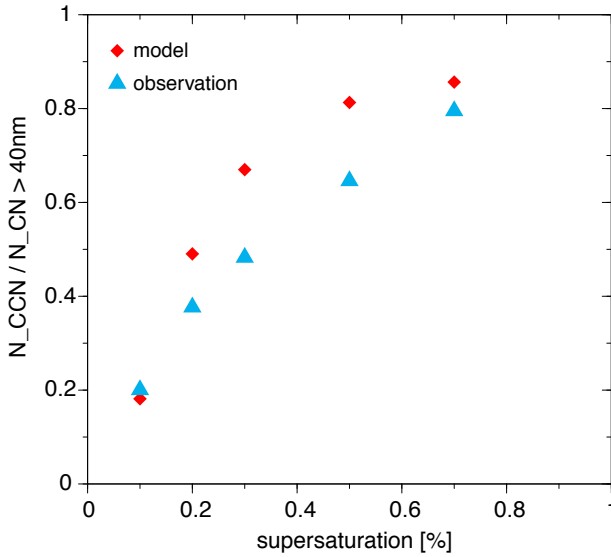

**Figure 5.** Simulated and observed fraction of CCN number concentration to the total number concentration of particles with a diameter larger than 40nm (N_CCN/N_CN) as a function of supersaturation.

### 3.4 Evaluation of the vertical structure of CCN

In order to evaluate the vertical distribution of the CCN concentrations and investigate its change since the 1980's, the modeled vertical profiles are compared to measurements. Fig. 6 compares the simulated and observed vertical profiles of the CCN number concentration for the two periods in 2013. Fig. 6a shows the comparison to CCN derived from lidar observation during

the spring period at Jülich, and Fig. 6b the comparison to the in-situ observations by the helicopter-based platform ACTOS during the fall period at Melpitz. Displayed are the median values as well as the 0.25- and 0.75-quantiles. For the spring period and close to the ground, the average CCN number concentration is overestimated by less than 50 %, which is in the range of the observation uncertainty of up to a factor of 2-3. However, up to a height of ~1.3 km, marking the average height of the boundary layer, the overestimation increases up to a factor of ~2. Nevertheless, the displayed 0.25-0.75 quantile range still overlaps in

the boundary layer. Above, the observed and modeled CCN concentrations start to decrease considerably, but clearly more strongly in the lidar observations. The model seems to transport too much aerosol mass into the free troposphere. In contrast to the model, the CCN number concentration derived from the lidar are on average negligible at heights above 4 km. Nevertheless, the variability of the observed CCN number concentrations is higher in the free troposphere. This is mainly an expression of increased detection uncertainty. The comparison to the in-situ observations by ACTOS during the fall period displayed in

Fig. 6b reveals a stronger overestimation also close to the ground by a factor of ~2. Also for this comparison, the modeled CCN number concentration does not as strongly decrease with height above the boundary layer (~1.5 km), hence increasing the overestimation. Note, that the larger variability of the median with height and the smaller 0.25-0.75 quantile range is caused by the smaller sample size of only 8 distinct cases compared to the 48 days with several hours of lidar observations during

the spring period. Furthermore, the ACTOS observation have a general uncertainty of only ~10 %. This, therefore, manifests the tendency of the model to overestimate the average CCN concentrations in the boundary layer by up to a factor of 2 and higher above the boundary layer. The general overestimation could be reduced by assuming different aerosol size distributions, which are used to convert modeled aerosol mass into number. However, the utilized size distributions were derived from data

5  at Melpitz and any other size distribution would therefore be less justified. It can be expected that the size distribution is not constant in time and in space as currently applied. Simulations that treat the aerosol in a size-resolved manner including aerosol microphysics are a useful tool to provide more insight into the temporal and spatial variability of the aerosol size distribution and hence the CCN number concentration. However, due to the increased degrees of freedom and similar assumptions, such as the size distribution during the emission, the results are not necessarily more accurate. Overall, although the model tends

10  to overestimate the average CCN concentrations, the modeled present day-CCN number concentration is in-line with the observations, whereas the estimated profile for the 1980's is far outside today's observational range (cf. Figs. 6 and 7). This indicates the influence of anthropogenic air pollution on the CCN number.

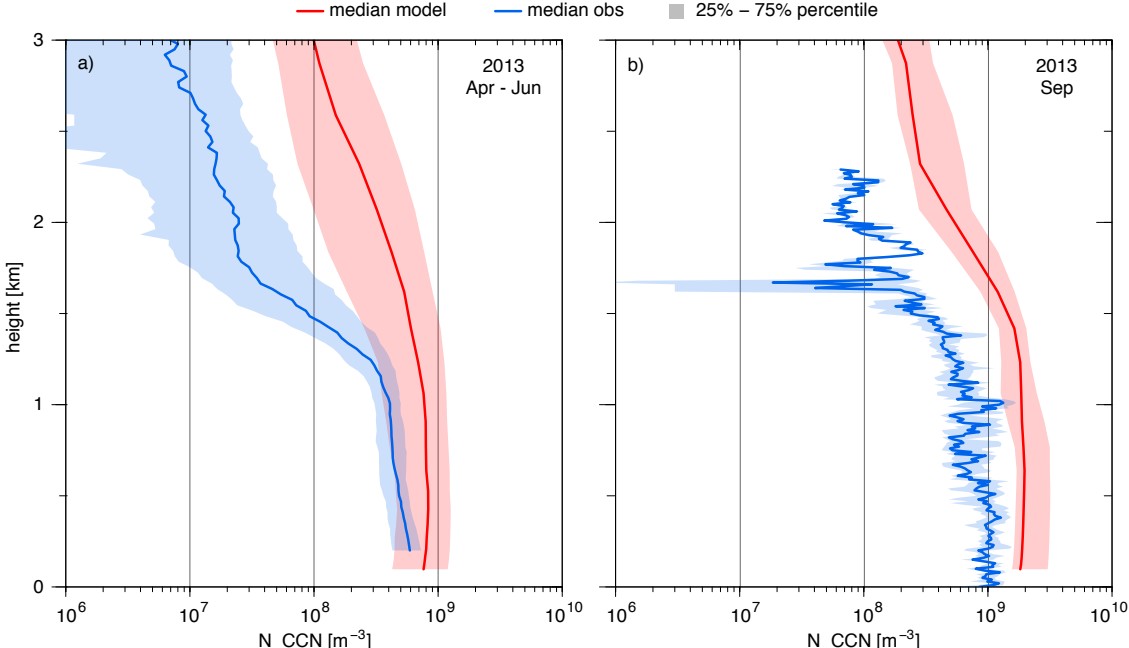

**Figure 6.** Comparison of the simulated vertical profiles of CCN number concentration (red) to profiles derived from observations (blue) of (a) lidar (04/05 2013) at Jülich, Germany, and (b) ACTOS (09/2013) at Melpitz, Germany. The CCN number concentrations were calculated or measured for a supersaturation of 0.2 %. The shading depicts the range between the 0.25- and the 0.75-quantile. On 48 and 8 different days, 335 and 27 model profiles (instantaneous hourly output), which matched the time of observations, could be taken into account for the spring and fall period.

### 3.5 Present day and historic vertical CCN profiles

For each of the two periods, a temporally and spatially averaged vertical profile of the CCN concentration was calculated for the year 2013 and the mid 1980's scenario, which is displayed together with the 0.05, 0.25, 0.75 and 0.95 quantiles in Fig. 7a - d. For the calculation, a vertical velocity of $1\,\mathrm{ms^{-1}}$ was assumed. This is an example for the CCN fields that are required as input for the ICON-LEM simulations within the HD(CP)$^2$" project. In contrast to the previous analysis, the applied supersaturation, and hence the critical size of activation, is not fixed but now result from the competition of the aerosol particles for the available water vapor. Therefore, the supersaturation and critical size of activation depend on the aerosol composition and vary temporally and spatially. The shape and values of the profiles show no major differences for the spring and fall episode. Close to the ground, where aerosol particles are emitted, the number concentrations of CCN are higher than in the free troposphere. With increasing height, the number of aerosol particles and thus also that of CCN is decreasing. This is the case for both the 2013 and the mid 1980's scenario. In 2013, the concentrations are almost constant up to a height of $1\,\mathrm{km}$ (around $1.0 \times 10^9\,\mathrm{m^{-3}}$) due to the well mixed boundary layer and decrease above (Fig. 7a, b). This is less pronounced in the mid 1980's scenario (Fig. 7c, d), in which the concentrations close to the ground are much higher (around $3 \times 10^9\,\mathrm{m^{-3}}$) and decrease almost immediately with height. At the top of the uppermost simulated layer ($8\,\mathrm{km}$), similar concentrations of $5 \times 10^7$ to $1 \times 10^8\,\mathrm{m^{-3}}$ were found for both, the present day and peak aerosol scenario. Due to different vertical distribution of the aerosol constituents, the aerosol composition and, hence, aerosol hygroscopicity deviates between the mid 1980's and 2013. Therefore and since Fig. 7 presents the CCN concentration for a fixed vertical velocity leading to variable supersaturations, the shape of the CCN profiles in the two scenarios differs.

Based on the CCN profiles, a scaling factor for the CCN concentration was calculated, which varies with height (Fig. 7e, f). This scaling factor describes the difference in CCN number concentration between the past peak aerosol in the mid 1980's and present day conditions in Europe and is useful for sensitivity studies. The difference in the vertical profile of the CCN number concentrations between the 2013 and the mid 1980's scenario is the reason for the curvature in the plot of the scaling factor at around $1\,\mathrm{km}$ height (Fig. 7e, f), because at this height, also the concentrations in the 2013 simulations start to decrease. The efficacy of pollution reduction policies and the breakdown of industrial production in the former East-bloc countries at the end of the 1980's becomes evident in terms of CCN. Close to the ground, a factor of around 2 was found. The relative difference between mid 1980's and 2013 is most pronounced in the height between 2 and $5\,\mathrm{km}$, where a scaling factor of up to 3.5 was found. In the upper troposphere, the scaling factor decreases to around one, which means there is no difference between the 1980's and present day concentrations.

### 4 Summary and conclusions

The CCN number concentrations from different simulation estimates and observation techniques were compared for two periods of the HOPE field experiments in Germany in spring and fall 2013. Based on simulations of the mass concentrations of different aerosol species (ammonium sulfate, ammonium nitrate, sulfate, organic carbon, elemental carbon, sea salt, and mineral dust) using the regional chemistry-transport model COSMO-MUSCAT, the CCN number was computed offline using

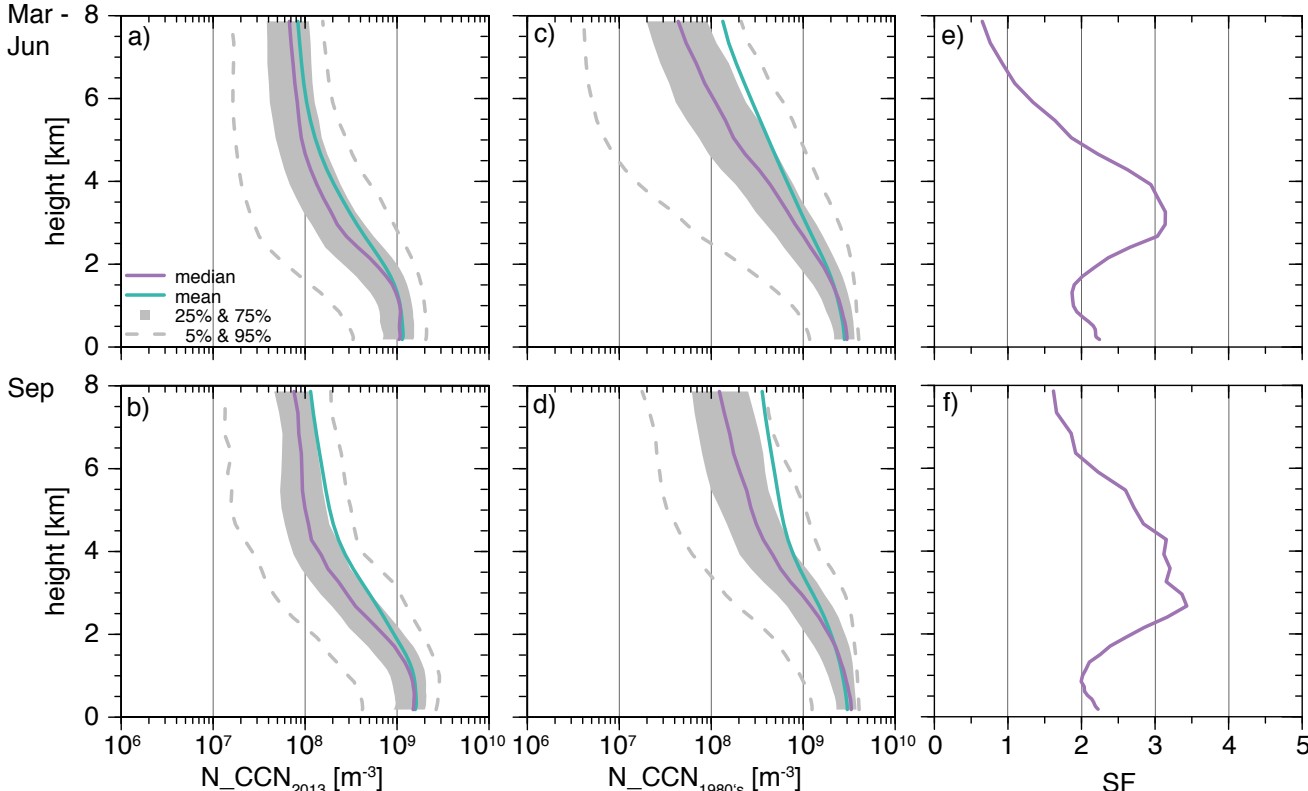

**Figure 7.** Spatial and temporal averaged vertical profile of the CCN number concentration as computed by COSMO-MUSCAT for the spring and fall period in 2013 (a and b), the estimation for the respective mid 1980's peak aerosol scenario (c and d) and scaling factor (SF) for the two scenarios (SF=N_CCN$_{1980s}$ / N_CCN$_{2013}$; e and f). For the calculation of the CCN number concentration, a vertical velocity of $1\,\mathrm{ms^{-1}}$ was assumed.

a state-of-the-art parameterization for cloud droplet activation. The resulting CCN number concentrations were compared to direct CCN measurements with a CCN counter, CCN number concentrations derived from applying the activation parameterization to gravimetrically measured aerosol mass concentrations, and vertical profiles derived from lidar observations and helicopter-borne in-situ measurements. In addition, CCN number concentrations representative for the mid 1980's, when the anthropogenic air pollution in Central Europe was highest, for the two periods were computed based on the COSMO-MUSCAT simulations of the year 2013. Comparing the results for the year 2013 and the mid 1980's scenario allows to investigate the impact of anthropogenic air pollution and the potential of the applied reduction measures on the atmospheric CCN budget.

At the ground and averaged over the full investigation period, the model-derived CCN concentration (for a supersaturation of 0.2 %) were about 16 % lower than the directly measured CCN concentrations and 37 % lower than the CCN concentrations derived from aerosol mass measurements. Hence, model and observation agree well for the longterm average. However, the deviations were different for the individual periods with 29 % underestimation of the measured CCN concentrations by the

model in the spring period and 37 % overestimation for the fall period. Discrepancies between observed and modeled CCN concentrations likely resulted mostly from uncertainties in the modeled aerosol mass and composition as well as the assumptions for the conversion from particle mass into number size distributions, which do not allow for the necessary flexibility to consider weather and tranport-related heterogeneity. The comparison of the ratio of the CCN number concentration and the total particle number of particles larger than $110\,\text{nm}$ in diameter shows a good agreement between model and observation for $0.2\,\%$ supersaturation. However, for supersaturations between $0.2\,\%$ and $0.7\,\%$ and smaller threshold sizes to define CN (e.g., particles larger than $40\,\text{nm}$), the model overestimates the activated particle fraction. Since the assumed prescribed size distributions were developed to correctly predict the average number of accumulation mode particles, which are the most relevant for deriving CCN number concentrations, the number of particles smaller than $\sim100\,\text{nm}$ is very likely underrepresented. As a non-linear process, aerosol activation depends strongly on the current ambient aerosol size distribution, which can vary considerably both temporally and spatially. Hence, the application of fixed size distributions in order to convert modeled aerosol mass to number concentrations is a source of uncertainty, which only for longterm averages might cancel out.

At the measurement station Melpitz, Germany, the model-derived average CCN concentration for the mid 1980's scenario was more than 5 times higher than for the year 2013. The underlying aerosol load of the 1980's scenario is expected to be reasonable since a comparison of modeled to satellite-based AOD (Costa-Surós et al., 2020) showed good agreement on average. Again, the application of fixed prescribed parameters for the number size distributions likely is a source of uncertainty since the aerosol size distribution in 2013 and the 1980's were not necessarily similar.

Within the boundary layer, the simulated vertical profiles of the present-day CCN concentration are within the variability range of the CCN derived from lidar measurements but do deviate from the in-situ helicopter-borne CCN measurements outside their 0.25-0.75 quantile range (and up to a factor of 2 for the median). The strong decrease of the observed CCN concentrations above the boundary layer could not be met by the model, hence strongly overestimating the CCN concentration in the free troposphere. The mid 1980's scenario, however, has much larger CCN number concentration far outside the variability range of the present-day observations.

By comparing the CCN concentrations modeled for the year 2013 and the mid 1980's scenario, the effect of strict emission reduction policies and reorganization of industrial production in Eastern Europe after 1990 becomes apparent. A domain and time averaged vertically resolved scaling factor for the CCN concentration between the year 2013 and the mid 1980's was computed, which is well suited for application in model sensitivity studies, in particular for studies that do not consider aerosol transport and chemistry explicitly. The scaling factor for estimating the CCN concentrations during the 1980's from current simulations is not vertically homogeneous. Close to the ground, a scaling factor of 2 was determined, increasing to 3.5 between 2 and $5\,\text{km}$ height. Towards the upper troposphere at around $8\,\text{km}$ height, the scaling factor decreases again to 1. The vertical variability of the CCN scaling factor is caused by the changed chemical composition of the aerosol due to the 1980's emission estimates. Especially the height range of up to $5\,\text{km}$, where a very high CCN number concentration during the 1980's was found, is important for cloud and precipitation formation in the mid-latitudes (e.g., Lebo, 2014; Marinescu et al., 2017). A significantly higher number of CCN points to large differences in the cloud droplet number concentration and thus the radiative

properties of the clouds as well as the precipitation probability during that time. The analysis of the radiative impacts including effects on cloud cover and albedo effects should be subject of future studies.

*Data availability.*  Data used in this manuscript can be provided upon request by email to the corresponding author, Christa Genz (christa.genz@idiv.de).

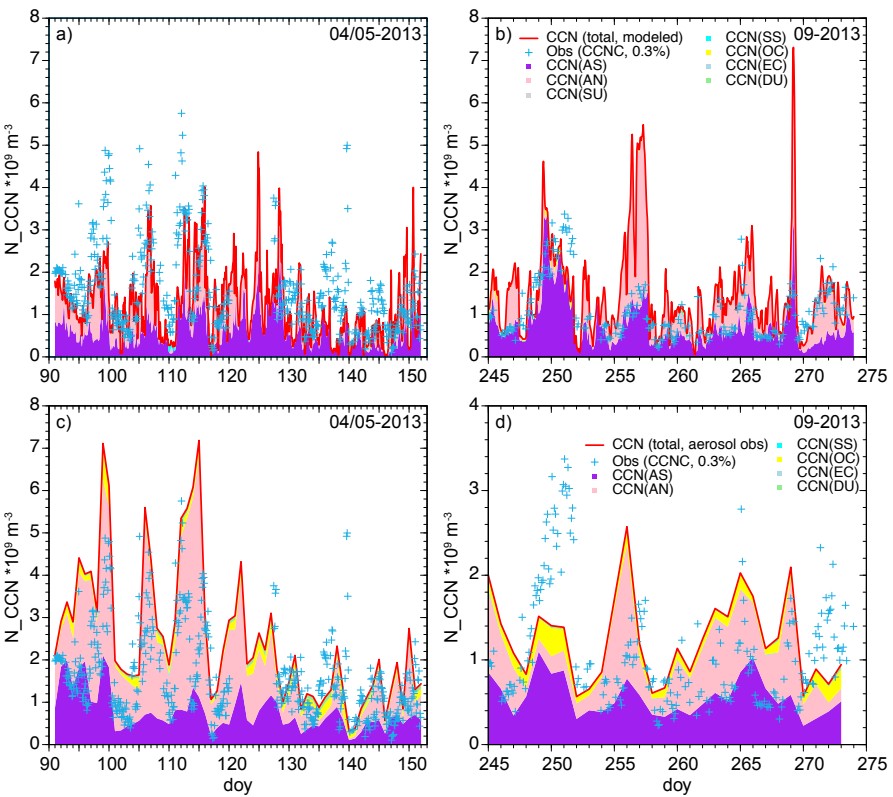

**Figure A1.** Simulated and measured CCN number concentrations in Melpitz at a supersaturation of 0.3 % during the two HOPE campaigns (April to May and September 2013). The upper panel (a and b) shows the CCN number concentrations resulting from the simulated aerosol concentrations, the lower one (c and d) the CCN numbers resulting from measured aerosol concentrations using the same CCN parametrization. The colors represent the contributions to CCN of different species. The blue crosses indicate the CCN number concentrations using the CCNC. Please note the different time resolution for the observations, as well as the different scale for the CCN number concentration in plot d.

*Author contributions.* Christa Genz working with Ina Tegen and Bernd Heinold ran the COSMO-MUSCAT model, and performed the
5   aerosol evaluation and CCN concentration calculation. Roland Schrödner joined during the analysis and coordinated the revision. Silvia Henning provided the CCNC measurements, Holger Baars the lidar derived CCN profiles and Gerald Spindler obtained the chemical measurements and analysis. Christa Genz and Roland Schrödner prepared the manuscript with contributions from all co-authors.

*Competing interests.* The authors declare that they have no conflict of interest.

*Acknowledgements.* This work was funded by the Federal Ministry of Education and Research in Germany (BMBF) through the research
5   programme "High Definition Clouds and Precipitation for Climate Prediction - HD(CP)$^2$" (FKZ: 01LK1503F, 01LK1502I, 01LK1209C
and 01LK1212C). The simulation data was generated using Copernicus Atmosphere Monitoring Service Information. The authors wish
to thank Luke Hande for providing a version of CCN-calculation code, which served as basis for the estimations shown here. We also
acknowledge good cooperation and support from the German Weather Service (Deutscher Wetterdienst, DWD) and the high performance
computing center Jülich. Furthermore the authors acknowledge support from ACTRIS under grant agreement no. 262 254 of the European
Union Seventh Framework Programme (FP7/2007-2013).

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
