# Peer review of "Estimation of Cloud Condensation Nuclei number concentrations and comparison to in-situ and lidar observations during the HOPE experiments"

_Atmospheric Chemistry and Physics, 2019_

## Referee Comment (RC1) · Anonymous Referee #1 · 25 Oct 2019

This paper presents estimation of CCN number concentrations based on 1) simulated mass concentrations of major particle-phase species using COSMO-MUSCAT and 2) measured mass concentrations of major particle-phase species obtained during HOPE campaign in 2013. The ground-level model simulations and measurements were used to estimate the vertical profile of CCN number concentrations based on parameterizations reported in previous studies assuming log-normal number-size distribution of externally mixed particles.

The estimated vertical profiles of CCN were compared with ground-based and airborne measurements during HOPE. In-situ CCN measurements were performed at the ground and on a helicopter. In addition, vertical CCN concentrations were estimated using lidar measurements. The vertical profile estimation agreed with observation near the ground but differed significantly at higher altitudes (Figure 6). The model was applied to peak aerosol scenario in Germany in 1985 to evaluate the anthropogenic impacts on CCN concentrations. No observation was available in 1985. They showed that higher anthropogenic emissions in 1985 resulted in higher CCN numbers compared to 2013 represented by the scaling factor (SF) > 1 (Figure 7).

I believe that the manuscript is technically sound and will be of interest to readers. However, I found the manuscript somewhat confusing and missing some details. I support publication after authors address the following specific comments.

1. A significant limitation of the CCN estimation appears to be that it neglects aerosol microphysics. If I understood correctly, the CCN prediction is linearly proportional to mass concentration since the size distribution is fixed. A large body of literature has shown that dependence of CCN number on emission is highly non-linear especially when microphysics of the nucleation and the growth of sub-CCN size particles into CCN size range are considered. For instance, Pierce and Adams (2009) showed that when primary emissions rates are varied by a factor of 3, tropospheric average CCN (at 0.2% supersaturation) only varied by 17% because higher primary emissions results in higher condensation and coagulation sinks. Although I recognize that such microphysics is beyond the scope of this paper, there should some discussion on the potential effects of the non-linearity of CCN numbers and emissions. In other words, the estimation of CCN based on peak emission in 1985 is likely to be significantly overestimated because higher condensation/coagulation sinks would prevent the growth of sub-CCN particles into CCN size range.

2. This study assumes ammonium nitrate concentration was zero in 1985. P6. L6 makes the following justification: "Particulate ammonium sulfate can be formed in the atmosphere from emitted sulfur dioxide and ammonia. In case there is still ammonia

left after this reaction, ammonium nitrate can be formed as well. As can be seen from Tab. 1, almost 20 times more SO2 was emitted during the 1980s compared to 2013. For this reason, there was much more sulfate available in the atmosphere than necessary for the transformation of the total available ammonia to ammonium sulfate. This is why in this study the production of ammonium nitrate was set to zero and half of the additional sulfur was transformed to sulfuric acid for the 1985 scenario."

Is there any literature to support this assumption? Why would not ammonium nitrate formation occur simultaneously with ammonium sulfate formation? According to their logic, when SO2 emission is very high, one would not observe nitrates in particles, but nitrates are commonly observed in a present-day polluted city with high SO2 with AMS measurements (e.g., Beijing) (Jimenez et al. 2009).

3. This study assumes additional sulfur was transformed to sulfuric acid for the 1985 scenario (p7. L3). How does that lead to "Sulfate" value of AS_2013 * 5.3 in Table 3?

4. Does "Sulfate" in Table 1 and 3 means "Sulfuric acid"?

5. Table 1: Shouldn't sigma ($\mu$m) in the log-normal distribution be dimensionless?

6. P8. Line 21. How are multiple-charge particles accounted for in in-situ CCNC measurements?

7. Is there any additional information on the mini CCN counter other than "custom built by Gregory C. Roberts"? Has it been used in any prior published study?

8. Conclusion: "In conclusion, the simulated profiles of the present-day simulation are within the variability range of the measurement-based profiles and thus represent realistic condition" This seems to be an over-statement. Figure 6 clearly shows the model and observation are outside 25-75% regions.

Typo

p.7 L22. Unnecessary comma (Spindler et al., 2013),.

p.12 L4 repeated "in"

p.14 Line 4. "cf. Figs.6 and ??" What is "??". Should it be 7?

References

Jimenez, J. L., Canagaratna, M. R., Donahue, N. M., Prevot, A. S. H., Zhang, Q., Kroll, J. H., DeCarlo, P. F., Allan, J. D., Coe, H., Ng, N. L., Aiken, A. C., Docherty, K. S., Ulbrich, I. M., Grieshop, A. P., Robinson, A. L., Duplissy, J., Smith, J. D., Wilson, K. R., Lanz, V. A., Hueglin, C., Sun, Y. L., Tian, J., Laaksonen, A., Raatikainen, T., Vaattovaara, P., Ehn, M., Kulmala, M., Tomlinson, J. M., Collins, D. R., Cubison, M. J., Dunlea, E. J., Huffman, J. A., Onasch, T. B., Alfarra, M. R., Williams, P. I., Bower, K., Kondo, Y., Schneider, J., Drewnick, F., Borrmann, S., Weimer, S., Demerjian, K., Salcedo, D., Cottrell, L., Griffin, R. J., Takami, A., Miyoshi, T., Hatakeyama, S., Shimono, A., Sun, J. Y., Zhang, Y. M., Dzepina, K., Kimmel, J. R., Sueper, D., Jayne, J. T., Herndon, S. C., Trimborn, A. M., Williams, L. R., Wood, E. C., Middlebrook, A. M., Kolb, C. E., Baltensperger, U., Worsnop, D. R. (2009). Evolution of organic aerosols in the atmosphere. Science 326:1525-1529.

Pierce, J. R. and Adams, P. J. (2009). Uncertainty in global CCN concentrations from uncertain aerosol nucleation and primary emission rates. Atmos. Chem. Phys. 9:1339-1356.

---

## Referee Comment (RC2) · Anonymous Referee #2 · 30 Oct 2019

**Review of acp-2019-742**

Estimation of Cloud Condensation Nuclei number concentrations and comparison to in-situ and lidar observations during the HOPE experiments

This study compares CCN estimates from various methods. One method involves converting bulk mass measurements from both the COSMO-MUSCAT model and from observations to CCN using assumed size distributions. CCN was estimated from these size distributions and compared to CCN observations (surface CCN, lidar, and in-situ measurements on a helicopter) during the HOPE observational period in 2013. A second comparison involves comparing the 2013 estimates to estimates for the year 1985. Overall, I am unclear of the takeaways of this study, as their results seem to be primarily due to how the set up their methods, which are often not clearly stated or justified, and similar to a study that the authors were recently involved in (Hande et al. 2016). Generally, I found it somewhat difficult to assess the results and discussion and think the authors need to provide more details in several locations throughout the manuscript, including more explanations. This being said, I think there is a lot of interesting data and methods and I do think comparisons between models and parameterizations with observations can be useful. However, in the way this study was written and presented, I wasn't able to clearly discern this study's scientific contributions.

**Major Concerns:**

**The calculation of the 1985 estimates**

One of the main parts of this study is comparing results from 2013 to 1985. However, if I am understanding correctly, the manner in which the authors determine the 1985 values is by taking the 2013 simulated aerosol mass results and simply scaling them up by certain factors (Table 3). Then, the authors present the result that the CCN is higher for their 1985, but isn't this simply because they scaled up the mass concentrations in their methods. By simply scaling the 2013 model solution up and down, the authors may not be considering the different aerosol processes present in the model that may change with different emissions and concentrations within the model. For example, can the authors justify that the size distributions that are used to convert aerosol mass to the number should be the same in 2013 as in 1985. This may change their results significantly. I wonder if it would not make more sense to make assumptions about the emissions in 1985 and allows the model to run with those emissions, such that the processes are better represented with these higher concentrations are better represented.

Also in Table 3, the authors do not explain where the scaling factors (3.9, 5.3, 2) come from? They also do not explain why they assume elemental carbon should be twice as high, since they themselves state there were no emission data to support this. More details are necessary to properly assess this study, especially since these are the primary methods in terms of estimating the 1985 concentrations.

**Unclear and unjustified methods**

I appreciate that the authors' comprehensive study and comparing models to observations, which is a difficult task. However, throughout reading the manuscript, there were many instances where I either did not understand what the authors were doing and/or why they were doing it, which made it difficult to assess their results and conclusions. Although I have included some of these instances directly below, in general, I think the writing in the manuscript could be made much more clear and suggest the authors work on this.

P4, L15. The authors need to convert aerosol mass to a number size distribution, and therefore, must assume some size distribution shape. The authors end up choosing a one-modal size distribution for each species, with parameters given in Table 1. Is it a good assumption to use one modal size distribution in this region? I am not particularly familiar with aerosol size distributions in this region, but I think the authors need to justify why using a one mode size distribution is valid and better justify the parameters chosen in Table 1, as opposed to just providing references. This assumption leads to one of their main results -- that they cannot CCN at higher supersaturation doesn't compare well, because they only used one mode at accumulation mode particle sizes.

P8, L24-32: The authors clearly explain the process in which they estimate CCN. However, why do the authors not simply add up the CCN measured from from the different size selections to get the total CCN? The authors seem to currently convert the measured CCN to activated fraction, just to convert it back to CCN again? Can the authors clarify why they did this in this manner?

P5, L20: "To achieve the maximum supersaturation (as a function of vertical velocity), accounting for particle growth before and after activation, the supersaturation balance is used." I do not understand this sentence and it seems to be important for the CCN calculation. What is the supersaturation balance? This paragraph in general, I think is very important, since it goes over how the estimated aerosol size distribution is converted to CCN, and therefore, I think it needs to be made much more clear how you are doing this. In its current form, I find it difficult to follow.

P5, L25-27: I do not understand these sentences. What model are you referring to, and why would producing realistic supersaturations necessarily mean that you are also producing realistic CCN -- it also depends on if your aerosol number concentrations are realistic? Furthermore, why do the authors assume this model producing realistic supersaturations for stratiform clouds? I am confused by these sentences.

**3: More explanations of the results**

The authors present many comparisons, but often do not fully explain why there comparisons are the same or different. I have included some of these instances below.

P 11, L15-16: The authors state that nitrate is problematic to simulate (especially in spring). However, they just discussed how it was difficult to simulate in the fall. Therefore, why especially in the spring?

P11, L15-16: The authors state that nitrate is difficult to measure (especially in the fall). Why is nitrate more difficult to measure as compared to other species? Why especially in the fall?

P12, L3: The authors state that a "too large number of CCN could result from too many large particles". However, instead of speculating, the authors have the simulation data and the conversions to particle number size distributions and the observations to confirm whether this is the case and explain why.

P13, L5-7: This seems to be one main result of this manuscript, but it primarily based on data and results from a prior study (Hande et al., 2016). A possible contribution from this study would be to explain why their model is producing too many particles in the size range from 80-110nm?

Figure 6 and P13 L13-20: One of the main conclusions of this study is that the vertical profiles compare well with the model and observations. However, all that is compared is an average over a month time period, and the authors conclude that it compares well because it is within a factor of 2 from the observations. Given the high temporal variability (seen in Figure 2), some temporal analysis should be considered here. Furthermore, it is difficult to read Figure 6 and see what the values and differences actually are.

Figure 7: One of the main results in this study is the shape of the CCN profile in Figure 7 and the differences between the model and observations. However, the authors don't really explain why it is different.

**4: References**
**4A: Lack of reference**
There is very little background included in this manuscript, many of which is likely very important to the authors results. On P2 L27, the authors state that comparison of modeled CCN to observations of CCN are sparse. However, since this is the entire point of this study, the authors should present the background literature here, as those results will likely be relevant to the results in this study.

**4B: Inconsistent referencing practices.**
Furthermore, the authors have many inconsistent referencing practices. For example, they include references from some instruments and projects, and do not include references for others. The authors should take some time to include relevant references throughout the manuscript. Some specific examples are listed below:
P3 L19: What is GME?
P7 L22 - P8 L2: Reference should be given for ACTRIS and EMEP.

P8 L3-4: References should be given to substantiate what HOPE is about.

P8 L9-12: References

P9, L1-2: Reference for this new instrument that the authors are using in the helicopter platform. If there is no reference, the authors should include more details about this instrument here, such as its specifications and where it was placed on the helicopter. The authors provide a lot of details in terms of ground measurements, but provide no details about the helicopter measurements.

Table 2: Is the reference for this only personal communication? This seems rather weak, and given that this is the basis of the 1985 estimates, I think the authors should supply a better reference. Who is Kevin Hausman, for whom this personal communication was with?

**Other Comments:**

**Qualitative, subjective explanations:** In general, the authors make several subjective statements instead of providing qualitative results that would be more useful to the reader. I have included a few instances below.

P11, L4: The authors state that the analyses shown in Figure 2 are in good agreement. However, this is qualitative and subjective. Can the authors provide quantitative, objective measures to describe their results here?

P11, L16: "results compare satisfactorily." This is subjective.

P11, L20: The authors use a 1-to-1 scatter plot to compare their results, and state that the model data underestimates CCN compared to the other methods. Can the authors add more quantitative details here. Underestimate by how much? Can the authors possibly do least squares fits for their data in Figure 3 to accomplish this or some other method in order to provide some more concrete results?

P12, L5: "Figures 2 and 3 show that the CCN number concentration is in similar agreement." Can the authors quantify this result.

P13, L17-18: The authors state that the CCN number if overestimated by less than 50% and that this is quite well. This is subjective. Can the authors put this into context, in terms of how being 50% off would impact cloud processes or better explain why they think this is quite well?

P13, L19: "decrease considerably." How much is considerably?

**Unclear discussions**

P12, L5-8: The authors first state that the results are in similar agreement, but then state that the differences are a factor of 2, and then state the difference is about 13%. I found these statements to be quite confusing. They must be comparing different things, but I wasn't sure

what was being compared. I think the authors should be careful about making it very clear what they are referring to throughout their manuscript, since there are a lot of different data being presented in this study.

P13, L17-18: The authors state that the ccn number if overestimated by less than 50%, but that seems to only be the case of the lidar measurements but not the ACTOS measurements. Can the authors clarify? A much different picture is present in the ACTOS measurements, which are not really discussed.

P11, L25-32: One large uncertainty in this analysis and comparison revolves around the size distributions being used in the model conversion from mass to number and whether these are representative for this site. As such, I think this should be included here.

P14, L1-2. The authors state that there is increased variability in CCN number conc. In the free troposphere, which I think is based on the increased 25%-75% quartile range in Figure 5. However, the authors state this is "mainly an expression of the considerably increased detection uncertainty". However, the same trend is present in the model data, which does not have this detection uncertainty. Can the authority clarify why they believe this to be the case?

**Additional Specific Comments**

P11, L22-24: Since SOA is so important to this region, why wasn't it included in the COSMO-MUSCAT simulations?

P2 L2: "For a realistic simulation of cloud adjustment..." What is cloud adjustment?

P2 L12-14: I do not understand what the authors are describing with these two sentences.

P2 L32: Why 1985? Can the authors provide background information here as to why one would be interested in the year 1985?

P3 L2: "This implies...". Can the authors clarify what this refers to? I am unclear and generally confused by this sentence?

P3 L4-9: The authors state they will be comparing 5 CCN estimates, but they seem to be estimates of different things. However, it is unclear why they are comparing these 5 items? As this is the introduction to the authors study, the authors should make this clear. What is the ultimate goal of this comparison and study?

P3 L18: What is a composition cycle?

P3 L19: The authors state that the simulations will be reinitialized every 48 hours, and provide other details, without yet describing the basics of the simulation. Therefore, it is difficult to

understand why a 48 initialization time is reasonable. It is suggested that the authors provide more details at the beginning of this section to make this section more readable.

P4 L8: Why do the authors only use model data up to 8km? The authors should provide a reason to justify or explain why they are not using the full model data?

Figure 1: Why is only Melpitz shown, when Julich is also mentioned? Were measurements taken at the different city, Julich? I was confused about this throughout the manuscript.

P4 L2: "in order to be considered" for what?

P4 L3: The authors state the model simulations were run. Are these ICON-LES simulations, which were just mentioned in the previous sentence or COSMO-MUSCAT simulations? I think it is the COSMO-MUSCAT simulations, but can the authors make this more clear?

P4 L17-18. It seems that the authors imply that total mass concentrations is converted to total number concentrations before the log-normal size distribution parameters are set? However, these size distribution would be necessary for the initial conversion to mass to number. So I am ultimately confused by what the authors are actually doing?

P5 L7: Which of the values are not according to Hande et al. (2016) and why?

P5, L14: The authors mentioned that ammonium sulfate has a kappa of 0.6, but then use 0.51 in their table. Why?

P5, L33-34 -- this should be included in the introduction.

P5, L7: The authors reference Table 1, but mean to reference Table 2.

Table 4: How were the measured aerosol mass concentrations obtained? Can the authors include this information in the caption, similar to how they mention that the modeled concentrations came from the COSMO-MUSCAT simulations.

P10, L16-19: The authors discuss why the measured dust was larger than the modeled dust. Can the authors make these statements more clear. For example, why is it OK to assume the difference in total mass from the measured species must be from dust?

P11, L8: The authors state the nitrate was overestimated by a factor of 2. Can the authors present this information more clearly, possible on the same figure to make this more clear.

P11, L8-16: A majority of the discussion is focused on a 3 day period, which is very difficult to see in Figure 2. Can the authors create a new figure that zooms in on this period, to allow the reader to more easily follow along in the figure.

Figure 6: Why isn't the fall period shown here?

P14, L8. The authors state that for this calculation a vertical velocity of 1 m/s is used. I don't understand what this is referring too.

Figure 7: What supersaturation is used for this analysis?

Can the authors changes their units to $cm^3$ from $m^3$, such that it is easier to comprehend the values presented?

---

## Author Response (AR1)

*Referee Comment:* This paper presents estimation of CCN number concentrations based on 1) simulated mass concentrations of major particle-phase species using COSMO-MUSCAT and 2) measured mass concentrations of major particle-phase species obtained during HOPE campaign in 2013. The ground-level model simulations and measurements were used to estimate the vertical profile of CCN number concentrations based on parameterizations reported in previous studies assuming log-normal number-size distribution of externally mixed particles.

The estimated vertical profiles of CCN were compared with ground-based and airborne measurements during HOPE. In-situ CCN measurements were performed at the ground and on a helicopter. In addition, vertical CCN concentrations were estimated using lidar measurements. The vertical profile estimation agreed with observation near the ground but differed significantly at higher altitudes (Figure 6). The model was applied to peak aerosol scenario in Germany in 1985 to evaluate the anthropogenic impacts on CCN concentrations. No observation was available in 1985. They showed that higher anthropogenic emissions in 1985 resulted in higher CCN numbers compared to 2013 represented by the scaling factor (SF) > 1 (Figure 7).

I believe that the manuscript is technically sound and will be of interest to readers. However, I found the manuscript somewhat confusing and missing some details. I support publication after authors address the following specific comments.

**Author Response:** The authors would like to thank the anonymous Referee #1 for the consideration of the manuscript to ACP and for the constructive comments and suggestions made to improve the manuscript. According to the referee's comments, the authors have revised the manuscript. All comments and changes in the manuscript are addressed in the following.

*Referee Comment:* 1. A significant limitation of the CCN estimation appears to be that it neglects aerosol microphysics. If I understood correctly, the CCN prediction is linearly proportional to mass concentration since the size distribution is fixed. A large body of literature has shown that dependence of CCN number on emission is highly non-linear especially when microphysics of the nucleation and the growth of sub-CCN size particles into CCN size range are considered. For instance, Pierce and Adams (2009) showed that when primary emissions rates are varied by a factor of 3, tropospheric average CCN (at 0.2% supersaturation) only varied by 17% because higher primary emissions results in higher condensation and coagulation sinks. Although I recognize that such microphysics is beyond the scope of this paper, there should some discussion on the potential effects of the non-linearity of CCN numbers and emissions. In other words, the estimation of CCN based on peak emission in 1985 is likely to be significantly overestimated because higher condensation/coagulation sinks would prevent the growth of sub-CCN particles into CCN size range.

**Author Response:** The fact, that an increase in emissions does not cause an increase of the concentrations of particle mass by the very same factor, is considered as well in our study. The increase of the $SO_2$ emission rates of a factor of 19 resulted in estimated mass concentration increases of 3.9 and 5.3 for ammonium sulfate and sulfuric acid, respectively, which in the end, under the assumption that the size distribution of each chemical species is the same in 1985 and 2013, results in roughly a factor of 2 increase in CCN number concentration. Although the size distributions are assumed to be the same in 1985 and 2013, the aerosol composition is different, and hence at least this non-linear effect of changed emissions on the calculated number of CCN is considered.

We are aware that the size distribution between the two scenarios 1985 and 2013 is likely different. The CCN concentrations might be overestimated for the 1985 scenario and should therefore be considered as an upper border estimate for the highly polluted period in Europe in the 1980s. In the revised manuscript we will discuss this more clearly.

**Change in manuscript (p.8, l.14 - 18):** Due to lack of observational data of aerosol size distributions in the 1980s in the study region to able to generalize 2010s and 1980s size distributions, for this study the same size distributions for 1985 and 2013 were assumed. Since the size distribution is crucial in order to translate modeled aerosol mass into number and finally derive

==highlight== CCN number, this assumption is likely an important source of uncertainty, which is difficult to reliably quantify. ==highlight==

*Referee Comment: 2. This study assumes ammonium nitrate concentration was zero in 1985. P6. L6 makes the following justification: "Particulate ammonium sulfate can be formed in the atmosphere from emitted sulfur dioxide and ammonia. In case there is still ammonia left after this reaction, ammonium nitrate can be formed as well. As can be seen from Tab. 1, almost 20 times more SO2 was emitted during the 1980s compared to 2013. For this reason, there was much more sulfate available in the atmosphere than necessary for the transformation of the total available ammonia to ammonium sulfate. This is why in this study the production of ammonium nitrate was set to zero and half of the additional sulfur was transformed to sulfuric acid for the 1985 scenario." Is there any literature to support this assumption? Why would not ammonium nitrate formation occur simultaneously with ammonium sulfate formation? According to their logic, when SO2 emission is very high, one would not observe nitrates in particles, but nitrates are commonly observed in a present-day polluted city with high SO2 with AMS measurements (e.g., Beijing) (Jimenez et al. 2009).*

**Author Response:** As mentioned by you, ammonium nitrate can also from under very high concentrations of sulfur. We oriented our calculations at the representation of the ammonium sulfate and ammonium nitrate formation in the model (Hinneburg et al., 2009). The implementation follows Simpson et al. (2003) who state (p. 39): "In the model ammonium sulphate is formed instantaneously from $NH_3$ and $SO_4$, only limited by the availability of the least abundant of the two species. Any excess $NH_3$ may then react with $HNO_3$, forming ammonium nitrate ($NH_4NO_3$) in an equilibrium reaction." Therefore the model would not form any ammonium nitrate in the 1985 scenario (see calculation in the response to the next comment). Ammonium nitrate is the only particulate nitrate compound formed in the model. Therefore, the model is likely to underestimate the nitrate concentration under conditions with high $SO_2$ emissions.

Furthermore, the density, size distribution and activation potential of ammonium sulfate and ammonium nitrate are similar. Therefore, exchanging part of the ammonium sulfate with ammonium nitrate would introduce only minor changes to the calculated CCN number concentration.

The revised manuscript will be improved with a more detailed explanation of the assumptions and calculation of the 1985 aerosol conditions.

Hinneburg, D., E. Renner, and R. Wolke (2009), Formation of secondary inorganic aerosols by power plant emissions exhausted through cooling towers in Saxony, Environ Sci Pollut R, 16(1), 25-35.

Simpson, D., et al. (2003), Transboundary acidification, eutrophication and ground level ozone in Europe PART I: Unified EMEP model description., EMEP Report 1/2003, Norwegian Meteorological Institute, Oslo, Norway.

**Change in manuscript (p.6, l.20 - p.8, l.21):** The whole section was revised.

*Referee Comment: 3. This study assumes additional sulfur was transformed to sulfuric acid for the 1985 scenario (p7. L3). How does that lead to "Sulfate" value of AS_2013 * 5.3 in Table 3?*

**Author Response:** Since no access sulfuric acid after formation of ammonium sulfate was left in the 2013 scenario, scaling up the 2013 sulfuric acid mass could not be done. Therefore, assumptions were made to calculate the sulfuric acid mass for 1985 from the ammonium sulfate concentrations in 2013:

- In the model, the formation of ammonium sulfate is described by paring ammonia and sulfate until one of the two species is consumed completely.
    - 1985: High $SO_2$ concentrations lead to all $NH_3$ is going to ammonium sulfate.
    - 2013: Lower $SO_2$ concentrations leave a certain amount of ammonia unconsumed, which is then transformed to ammonium nitrate with the available nitrate.
- In the 1985 scenario, no ammonia is left for the formation of ammonium nitrate and we assume that no ammonium nitrate is formed. As mentioned above, due to the similar

hygroscopic properties for ammonium sulfate and ammonium nitrate, this simplification is not expected to have significant impacts on the CCN concentration.

- 50% of the $SO_2$ leftover after ammonium sulfate formation is assumed to form sulfuric acid. This is a source of uncertainty and will be discussed in the revised manuscript.

The calculation of the scaling factors between 2013 and 1985 is as follows:

- 2013: The $SO_2$ emissions of 0.41 Mt are completely transformed to ammonium sulfate $((NH_4)_2SO_4)$, which then consumes 0.22 Mt $NH_3$ to form 0.845 Mt ammonium sulfate.
- 1985: The emitted $NH_3$ (0.86 Mt) is completely transformed to ammonium sulfate, which results in 3.32 Mt ammonium sulfate. This process consumes 1.61 Mt $SO_2$, leaving 6.12 Mt $SO_2$ unconsumed.
- The scaling factor between 1985 and 2013 for ammonium sulfate is then calculated as 3.9 (3.32 Mt / 0.845 Mt).
- 50 % of the leftover $SO_2$ after ammonium sulfate formation (3.06 Mt) is transferred to sulfuric acid resulting in 4.68 Mt sulfuric acid. Since no further sulfate (sulfuric acid) is present in 2013, we calculate the 1985 sulfate concentration from the 2013 ammonium sulfate concentration. The ratio between the formed sulfate in 1985 (4.68 Mt) and the formed ammonium sulfate in 2013 (0.845 Mt) results in a scaling factor of 5.3.

We will improve the description of the assumptions and calculations of the 1985 aerosol concentration estimates in the revised manuscript accordingly.

**Change in manuscript (p.6, l.20 - p.8, l.21):** The whole section was revised.

*Referee Comment: 4. Does "Sulfate" in Table 1 and 3 means "Sulfuric acid"?*

**Author Response:** Sulfuric acid is formed in the model. Due to its low vapor pressure, it is entirely partitioned to the particle phase as sulfate. The accounted molar mass is that of sulfate. We will make this clearer in the revised version to avoid confusion.

**Change in manuscript (p.8, l.5 - 6):** This is why in this study the production of ammonium nitrate was set to zero and half of the additional sulfur was transformed to sulfuric acid for the 1985 scenario. Sulfuric acid is assumed to entirely partition to the particulate phase and is therefore accounted for as sulfate.

*Referee Comment: 5. Table 1: Shouldn't sigma (m) in the log-normal distribution be dimensionless?*

**Author Response:** Yes, this is correct and will be changed accordingly in the revised manuscript.

**Change in manuscript:** Changed accordingly in Table 1.

*Referee Comment: 6. P8. Line 21. How are multiple-charge particles accounted for in in-situ CCNC measurements?*

**Author Response:** The monodisperse CCNc measurements were corrected for multiple-charged particles up to three charges. For the whole diameter size range the apparent connected diameter of doubly and triply charged particles was calculated via the linear relationship between single and doubly as well as between single and triply charged particles (Wiedensohler, 1988). For a given activation curve (CCN/CN) the fraction of apparently doubly and triply charged particles is than subtracted from the activated fraction. The resulting curve is valid for single charged particles and was fitted with a sigmoidal fit to get the activation diameter (50% of particles activated).

**Change in manuscript (p.10, l.6-9):** The ratio between the CCN number and the total particle number as counted by the CPC (condensation nuclei, CN) gives the activated fraction (AF) of the particles. The AF was corrected for multiply charged particles up to three charges by subtracting their apparent fraction from the AF using the charge equilibrium (Wiedensohler, 1988). This

multiple charge corrected AF is calculated for each particle diameter and results in a size dependent activation curve for each supersaturation.

Wiedensohler, A. (1988), An Approximation of the Bipolar Charge-Distribution for Particles in the Sub-Micron Size Range, J. Aerosol Sci., 19(3), 387-389.

**Referee Comment:** *7. Is there any additional information on the mini CCN counter other than "custom built by Gregory C. Roberts"? Has it been used in any prior published study?*

**Author Response:** This mini CCN instrument is working after the same principal as the commercially available CCNC-100 by the DMT, which was used for the ground-based CCN measurements (Roberts and Nenes, 2005). It has also been used during other campaigns like in the CARRIBA campaign on Barbados (Wex et al, 2016).

**Change in manuscript (p.10, l.15-18):** Within this study we use the vertically resolved in-situ data of the light weight mini cloud condensation nuclei counter (mCCNc, custom built by Gregory C. Roberts, working principal as in Roberts and Nenes, 2005), which has been applied successfully on ACTOS before (e.g. Wex et al. 2016).

Roberts, G. C., and A. Nenes (2005), A continuous-flow streamwise thermal-gradient CCN chamber for atmospheric measurements, *Aerosol Sci. Technol.*, *39*(3), 206-221.

Wex, H., et al. (2016), Aerosol arriving on the Caribbean island of Barbados: physical properties and origin, *Atmospheric Chemistry and Physics*, *16*(22), 14107-14130, doi:10.5194/acp-16-14107-2016.

**Referee Comment:** *8. Conclusion: "In conclusion, the simulated profiles of the present-day simulation are within the variability range of the measurement-based profiles and thus represent realistic condition" This seems to be an over-statement. Figure 6 clearly shows the model and observation are outside 25-75% regions.*

**Author Response:** Thank you for the important comment. The section is re-written in the revised manuscript and will focus on the representation in the boundary layer. In the boundary layer model and observations agree mostly within a factor of two. However, the model generally overestimates the observations. Considering the factor of 2-3 uncertainty of the observed CCN number concentrations based on the LIDAR retrieval (Mamouri and Ansmann, 2016), we think that there is reasonable agreement within the boundary layer. For the few days of the in-situ CCN-profile observed by the helicopter-borne platform ACTOS, the model and observation 25-75% range do not overlap, however, the overestimation of a factor of 2 in the boundary layer is generally acceptable for the scope of the study.
In the free troposphere, the model more strongly overestimates the observations. In particular the sharp decrease in CCN number concentration at around 1.5 km height is not represented as sharp in the model. We will include a more detailed discussion in the revised manuscript.

**Change in manuscript (p.20, l.7-9):** Within the boundary layer, the simulated vertical profiles of the present-day simulation are within the variability range of the CCN derived from lidar measurements but do deviate from the in-situ helicopter-borne CCN measurements outside their 0.25-0.75 quantile range.

**Referee Comment:**
*Typo*
*p.7 L22. Unnecessary comma (Spindler et al., 2013),.*
*p.12 L4 repeated "in"*
*p.14 Line 4. "cf. Figs.6 and ??" What is "??". Should it be 7?*

**Author Response:** Thank you for these findings. The mentioned typos will be corrected in the revised manuscript.

***Referee Comment:*** *Estimation of Cloud Condensation Nuclei number concentrations and comparison to in situ and lidar observations during the HOPE experiments. This study compares CCN estimates from various methods. One method involves converting bulk mass measurements from both the COSMO MUSCAT model and from observations to CCN using assumed size distributions. CCN was estimated from these size distributions and compared to CCN observations (surface CCN, lidar, and in situ measurements on a helicopter) during the HOPE observational period in 2013. A second comparison involves comparing the 2013 estimates to estimates for the year 1985. Overall, I am unclear of the takeaways of this study, as their results seem to be primarily due to how the set up their methods, which are often not clearly stated or justified, and similar to a study that the authors were recently involved in (Hande et al. 2016). Generally, I found it somewhat difficult to assess the results and discussion and think the authors need to provide more details in several locations throughout the manuscript, including more explanations. This being said, I think there is a lot of interesting data and methods and I do think comparisons between models and parameterizations with observations can be useful. However, in the way this study was written and presented, I wasn't able to clearly discern this study's scientific contributions.*

**Author Response:** The authors would like to thank the anonymous Referee #2 for the constructive comments and suggestions made to improve the manuscript. According to the referee's comments, the authors have further improved the manuscript. All comments and changes in the manuscript are addressed below.

***Referee Comment:***

***Major Concerns:***

*The calculation of the 1985 estimates*

*One of the main parts of this study is comparing results from 2013 to 1985. However, if I am understanding correctly, the manner in which the authors determine the 1985 values is by taking the 2013 simulated aerosol mass results and simply scaling them up by certain factors (Table 3). Then, the authors present the result that the CCN is higher for their 1985, but isn't this simply because they scaled up the mass concentrations in their methods. By simply scaling the 2013 model solution up and down, the authors may not be considering the different aerosol processes present in the model that may change with different emissions and concentrations within the model. For example, can the authors justify that the size distributions that are used to convert aerosol mass to the number should be the same in 2013 as in 1985. This may change their results significantly. I wonder if it would not make more sense to make assumptions about the emissions in 1985 and allows the model to run with those emissions, such that the processes are better represented with these higher concentrations are better represented. Also in Table 3, the authors do not explain where the scaling factors (3.9, 5.3, 2) come from? They also do not explain why they assume elemental carbon should be twice as high, since they themselves state there were no emission data to support this. More details are necessary to properly assess this study, especially since these are the primary methods in terms of estimating the 1985 concentrations.*

**Author Response:** We are aware that the size distribution between the two scenarios 1985 and 2013 is likely different. We chose to assume the same size distribution because we found no justification to choose a particular different size distribution. Estimating a realistic dynamic size distribution requires explicit simulations of microphysical aerosol transformation processes, which was beyond the scope of this work, but is planned for a follow-up study. In the revised manuscript, we will discuss the shortcomings concerning the size distribution in a more extended manner.
A model run with emissions of 1985 was not conducted, since only limited emission information are available for this time period and also historical emission scenarios are affected by large uncertainties, in particular in the eastern European countries during Soviet times. Emissions would be needed spatially and temporally resolved, but only countrywide spatial average values of the annual sums were available. Scaling the emissions was not applied, since the spatial distribution of the modern emission data base does likely not fit to the 1985 industrial emitters. The emission change is only known for the whole country, not distinguished by emission sectors or regions.

Additionally, the mass-based model does not allow for the free evolution of the aerosol size distribution under altered emissions. Therefore, we chose to scale the concentrations of only that species for which we know how much the countrywide emissions have changed.

The derivation of the scaling factors between 2013 and 1985 is as follows:

- In the model, the formation of ammonium sulfate is described by pairing ammonia and sulfate until one of the two species is consumed completely.
    - 1985: High $SO_2$ concentrations lead to all $NH_3$ is going to ammonium sulfate.
    - 2013: Lower $SO_2$ concentrations leave a certain amount of ammonia unconsumed, which is then transformed to ammonium nitrate with the available nitrate.
- 2013: The $SO_2$ emissions of 0.41 Mt are completely transformed to ammonium sulfate $((NH_4)_2SO_4)$, which then consumes 0.22 Mt $NH_3$ to form 0.845 Mt ammonium sulfate.
- 1985: The emitted $NH_3$ (0.86 Mt) is completely transformed to ammonium sulfate, which results in 3.32 Mt ammonium sulfate. This process consumes 1.61 Mt $SO_2$, leaving 6.12 Mt $SO_2$ unconsumed.
- The scaling factor between 1985 and 2013 for ammonium sulfate is then calculated as 3.9 (3.32 Mt / 0.845 Mt).
- 50 % of the $SO_2$ left after ammonium sulfate formation (3.06 Mt) is transferred to sulfuric acid resulting in 4.68 Mt sulfuric acid. Since no further sulfate (sulfuric acid) is present in 2013, we calculate the 1985 sulfate concentration from the 2013 ammonium sulfate concentration. The ratio between the formed sulfate in 1985 (4.68 Mt) and the formed ammonium sulfate in 2013 (0.845 Mt) results in a scaling factor of 5.3.
- The doubling of the EC concentration is a rough estimate to address the strong decrease of EC emissions due to extensive reconstruction measures, replacements and shutdowns or state of the art emission control systems of power and industrial plants in the 1980s and 1990s. Since no detailed emission data are available, an exact treatment of EC is not possible. For this reason, a doubling of the EC mass concentration was chosen as a rough estimate. The EC concentration is not expected to significantly change the CCN concentration due to the generally low hygroscopicity of EC (see Table 1).

We will improve the description of the assumptions and calculations of the 1985 aerosol concentration estimates in the revised manuscript accordingly.

**Change in manuscript (p6, l20 - p8, l21):** The whole section was revised.

*Referee Comment:*

*Unclear and unjustified methods*

*I appreciate that the authors' comprehensive study and comparing models to observations, which is a difficult task. However, throughout reading the manuscript, there were many instances where I either did not understand what the authors were doing and/or why they were doing it, which made it difficult to assess their results and conclusions. Although I have included some of these instances directly below, in general, I think the writing in the manuscript could be made much more clear and suggest the authors work on this.*

**Author Response:** Thank you for the important comments. We will improve the revised manuscript accordingly.

*Referee Comment: P4, L15. The authors need to convert aerosol mass to a number size distribution, and therefore, must assume some size distribution shape. The authors end up choosing a one modal size distribution for each species, with parameters given in Table 1. Is it a good assumption to use one modal size distribution in this region? I am not particularly familiar with aerosol size distributions in this region, but I think the authors need to justify why using a one mode size distribution is valid and better justify the parameters chosen in Table 1, as opposed to just providing references. This assumption leads to one of their main results    that they cannot*

*CCN at higher supersaturation doesn't compare well, because they only used one mode at accumulation mode particle sizes.*

**Author Response:** The assumptions made for the size distributions are explained, discussed and compared to measurements in more detail in Hande et al. (2016). They are based on AMS measurements of ambient concentration of the individual species (Poulain et al., 2011). A comparison of the estimated individual size distributions to observations can also be found in Hande et al. (2016). As can be seen in their Fig. 3c, the observed total aerosol size distribution at Melpitz is indeed bimodal with peaks at ~30 nm and ~100 nm diameter. The modeled total aerosol size distribution is combined of the ones of the individual species (which have different shapes and geometric mean radii) and is therefore multi-modal, and partly accounting for the Aitken mode range. The combined size distribution of the different species was found to match the observed size distribution well between 50 and 200 nm (i.e., the peak region of the observed accumulation mode), which is most relevant for estimating CCN. Applying a size-resolved aerosol transport model for a similar study would be a useful next step, which would also avoid uncertainty by using the same size distributions in the whole domain and for the 1980s and present day scenarios.

**Change in manuscript (p.4, l.29 - p.5, l.13):** The whole paragraph whose revised.

*Referee Comment: P8, L24-32: The authors clearly explain the process in which they estimate CCN. However, why do the authors not simply add up the CCN measured from from the different size selections to get the total CCN? The authors seem to currently convert the measured CCN to activated fraction, just to convert it back to CCN again? Can the authors clarify why they did this in this manner?*

**Author Response:** Thanks for pointing out this ambiguity in our description. In principal you are correct: It would be easier to just add up all measured CCN over the size range. However, if measuring monodisperse CCN one also has to account for multiple charged particles. These particles can have more than one electrical charge and as the particles are selected in the DMA by their mobility diameter, particles with more charges but larger diameter enter the CCNC at the same time. This effect influences the activation curve, because larger particles activate at lower supersaturation. Our multiple charge correction works in a way that the apparent diameter of doubly and triply charged particles is calculated and the actual fraction of those for each activation curve is calculated and subtracted from the activated fraction (AF). By doing so, we end up with the AF curve for single charged particles, which we use to derive the charge corrected CCN number.

**Change in manuscript (p.10, l.6-9):** The ratio between the CCN number and the total particle number as counted by the CPC (condensation nuclei, CN) gives the activated fraction (AF) of the particles. ==The AF was corrected for multiply charged particles up to three charges by subtracting their apparent fraction from the AF using the charge equilibrium (Wiedensohler, 1988).== This ==multiple charge corrected== AF is calculated for each particle diameter and results in a size dependent activation curve for each supersaturation.

Wiedensohler, A. (1988), An Approximation of the Bipolar Charge-Distribution for Particles in the Sub-Micron Size Range, J. Aerosol Sci., 19(3), 387-389.

*Referee Comment: P5, L20: "To achieve the maximum supersaturation (as a function of vertical velocity), accounting for particle growth before and after activation, the supersaturation balance is used." I do not understand this sentence and it seems to be important for the CCN calculation. What is the supersaturation balance? This paragraph in general, I think is very important, since it goes over how the estimated aerosol size distribution is converted to CCN, and therefore, I think it needs to be made much more clear how you are doing this. In its current form, I find it difficult to follow.*

**Author Response:** Overall, the paragraph was supposed to give only a little insight on the background of the widely used method derived by Abdul-Razzak et al. (1998) and Abdul-Razzak

and Ghan (2000). We will improve this paragraph in the revised manuscript by making clearer what is actually used and avoiding unnecessary distracting information, but rather refer the reader to the two papers. The supersaturation balance was utilized by Abdul-Razzak et al. (1998) to determine the maximum supersaturation of an air parcel rising adiabatically at uniform speed and given aerosol type. However, the term itself (used by Abdul-Razzak and Ghan (2000)) is not widely used and is therefore perhaps distracting. The formulations were extended for multiple aerosol types by Abdul-Razzak and Ghan (2000). Each aerosol particle can activate at a certain critical supersaturation. A formulation is derived, which describes how the critical radius of the smallest aerosol particle activated can be determined by the vertical velocity of an adiabatically rising air parcel: the critical supersaturation of the smallest activated particle is equal to the maximum supersaturation resulting from a uniform vertical velocity of an air parcel. Details of the derivation of the equations can be found in the papers of Abdul-Razzak et al. (1998) and Abdul-Razzak and Ghan (2000), to which it is referred to in the manuscript. Finally, in our study, the relation for the number of particles activated under given vertical velocity, aerosol composition and size distribution (Abdul-Razzak and Ghan (2000), Equation 13), is used.

**Change in manuscript (p.5, l.14 - p.6, l.9):** The whole paragraph whose revised.

*Referee Comment: P5, L25-27: I do not understand these sentences. What model are you referring to, and why would producing realistic supersaturations necessarily mean that you are also producing realistic CCN    it also depends on if your aerosol number concentrations are realistic? Furthermore, why do the authors assume this model producing realistic supersaturations for stratiform clouds? I am confused by these sentences.*

**Author Response:** The word "model" was misleading. We refer to the activation parametrization, which was already evaluated by Hande et al. (2016). The calculated supersaturation here is depending on updraft velocity and the size distribution and hygroscopicity of the aerosol particles. Stratiform clouds are often horizontally more homogenic and can be better described with grid scale vertical winds than convective clouds. Overall, the activation parameterization would also work for convective clouds if the updraft velocity would be known. We agree that this section was written confusingly and will improve it.

**Change in manuscript (p.6, l.10-14):** The same method to derive CCN concentrations from the modeled aerosol mass as applied in this study was utilized in a related study of the HD(CP)$^2$ project to parameterize the CCN concentrations as a function of vertical velocity (Hande et al., 2016). As written above, they evaluated the aerosol size distribution at Melpitz and found good agreement in the CCN size range. We therefore assume that the applied method generally produces realistic CCN concentrations. However, the ambient aerosol size distribution varies in time and space and therefore the assumption of a spatially and temporally constant size distribution for the different aerosol species is a source of uncertainty.

*Referee Comment:*

*3: More explanations of the results*

*The authors present many comparisons, but often do not fully explain why there comparisons are the same or different. I have included some of these instances below.*

*P11, L15-16: The authors state that nitrate is problematic to simulate (especially in spring). However, they just discussed how it was difficult to simulate in the fall. Therefore, why especially in the spring?*

*P11, L15-16: The authors state that nitrate is difficult to measure (especially in the fall). Why is nitrate more difficult to measure as compared to other species? Why especially in the fall?*

**Author Response:** The terms "especially in spring" and "especially in fall" are misleading and not entirely correct. We will delete these terms in the revised manuscript and will clarify the uncertainties of both observations and modelling.

Today, the concentration of ammonium nitrate in agricultural regions is depending on the available ammonia. The ammonia emissions are in the short term uncertain since the exact timing of bringing out manure is usually not known. Hence, in particular the magnitude and timing of observed ammonium nitrate concentration peaks cannot be expected to match by applying ammonia emission data bases (time variation covers only the general seasonal cycle).

For the measurements the aerosol is collected on filters in High-Volume samplers without cooling. Since nitrate is volatile, warmer temperatures within the sampling and storage units than in the ambient air can lead to partial evaporation of nitrate from the filter. Therefore, the measured ammonium nitrate concentrations are a lower border of the actual ammonium nitrate concentrations.

**Change in manuscript (p.12, l.11 - p.13, l.7):** The whole paragraph was revised taking into account also other comments.

*Referee Comment: P12, L3: The authors state that a "too large number of CCN could result from too many large particles". However, instead of speculating, the authors have the simulation data and the conversions to particle number size distributions and the observations to confirm whether this is the case and explain why.*

**Author Response:** Yes, this is true. We will re-write and extend the explanation in the revised manuscript. The model tends to underestimate the CCN concentrations, both for 0.2 and 0.3 % supersaturation by a similar percentage (13 and 11 %). The particle concentrations are underestimated by the model as well, but this effect is more pronounced for smaller particles. The underestimation for particles larger 110 nm is about 10 % but 35 % for particles larger than 80 nm. This explains the larger difference in the ratio of $N\_CCN_{0.3\%}$ / $N\_CN > 80nm$ compared to $N\_CCN_{0.2\%}$ / $N\_CN > 110nm$ (Fig. 4).

**Change in manuscript (p.14, l.3 - p.16, l.2):** The whole section was revised taking into account also other comments.

*Referee Comment: P13, L5-7: This seems to be one main result of this manuscript, but it primarily based on data and results from a prior study (Hande et al., 2016). A possible contribution from this study would be to explain why their model is producing too many particles in the size range from 80 -110nm?*

**Author Response:** The model underestimates both the particles larger than 80 nm and particles larger than 110 nm (see answer to previous comment) and therefore also particles between 80 nm and 110 nm. Since the applied chemistry transport model describes the mass concentrations of the different aerosol species, the calculated aerosol number size distribution, in particular its shape, depends on the assumed size distribution. The applied size distributions for the different species were compared to observations by Hande et al. (2016). The described underestimation is basically due to the utilized assumptions. These assumptions, however, have been proved by Hande et al. (2016) and were derived from observational data at the station Melpitz. Of course, we are aware that the chosen average size distribution is likely not representative for the whole domain and specific points in time. The present study, however, had the aim to provide estimates of the CCN concentrations in the 1980s using an offline method based on these assumptions. The respective paragraph was meant to discuss reasons for deviations between observation and modelled CCN concentrations. We will enhance the discussions at different parts of the manuscript and give more detailed information on the assumptions and known shortcomings. While there is always the possibility to design and conduct more sophisticated model experiments, we still believe that, despite the simplifying assumptions, our study provides valuable information on the CCN budget in the 1980s, which are of interest for the broader scientific community.

**Change in manuscript (p.14, l.3 - p.16, l.2):** The whole section was revised taking into account also other comments.

**Referee Comment:** *Figure 6 and P13 L13-20: One of the main conclusions of this study is that the vertical profiles compare well with the model and observations. However, all that is compared is an average over a month time period, and the authors conclude that it compares well because it is within a factor of 2 from the observations. Given the high temporal variability (seen in Figure 2), some temporal analysis should be considered here. Furthermore, it is difficult to read Figure 6 and see what the values and differences actually are.*

**Author Response:** A temporal analysis is included by displaying the temporal variability through the 0.25- and 0.75-quantiles of both the observations and the model. The plot does not show just the monthly mean, but mean of all measured profiles within the considered time periods. The model profiles were chosen for the exact times, for which the observations were available. We will improve Fig. 6 by adding the time periods and the summed observation time in the plot and the caption. Furthermore, we will make the plot wider and give vertical lines at the main ticks of the CCN number axis. The text will be revised by a better description of the data displayed in the Fig. 6.

**Change in manuscript (p16, l.5 - p.17, l.16):** The whole section was revised taking into account also other comments.

**Referee Comment:** *Figure 7: One of the main results in this study is the shape of the CCN profile in Figure 7 and the differences between the model and observations. However, the authors don't really explain why it is different.*

**Author Response:** Figure 7 does not show observational data, but the comparison between the modelled CCN concentrations in 2013 and 1985. We assume, this comment refers to Figure 6. The specific reasons for the differences are not known. There are several reasons that likely all come to play, however their contribution to the overall deviation between model and observation is speculative: i) the observational methods have general uncertainties (e.g., factor 2-3 for the lidar retrieved CCN concentrations), therefore it is not possible to compare to a well-known truth ii) the modelled aerosol concentrations are uncertain since the emissions are not known for short temporal and spatial scales, iii) the ambient aerosol size distribution is likely to deviate substantially from the applied mean size distribution, iv) the modelled boundary layer height is not as sharp as seen in the lidar observations (again for a number of potential reasons, which cannot be weighted by this study). We will enhance the discussion by giving potential reasons for the deviation between model and observation.

**Change in manuscript (p16, l.5 - p.17, l.16):** The whole section was revised taking into account also other comments.

**Referee Comment:**

*4: References*

*4A: Lack of reference*

*There is very little background included in this manuscript, many of which is likely very important to the authors results. On P2 L27, the authors state that comparison of modeled CCN to observations of CCN are sparse. However, since this is the entire point of this study, the authors should present the background literature here, as those results will likely be relevant to the results in this study.*

**Author Response:** Agreed. In the revised manuscript we will add more references to previous studies that have contributed to evaluating model representations of CCN activation.

**Change in manuscript (p.2, l.33 - p.3, l.2 and bibliography):** Such data sets can be used to evaluate the application of available aerosol activation parameterizations in atmospheric models. Evaluated against in-situ observations, the applied regional and global models (e.g., Spracklen et al., 2011; Bègue et al., 2015; Schmale et al., 2019; Fanourgakis et al., 2019; Watson-Parris et al., 2019) tend to underestimate the observed CCN concentrations.

Bègue, N., Tulet, P., Pelon, J., Aouizerats, B., Berger, A., and Schwarzenboeck, A.: Aerosol processing and CCN formation of an intense Saharan dust plume during the EUCAARI 2008 campaign, Atmospheric Chemistry and Physics, 15, 3497–3516, https://doi.org/10.5194/acp-15-3497-2015, https://www.atmos-chem-phys.net/15/3497/2015/, 2015.

Fanourgakis, G. S., Kanakidou, M., Nenes, A., Bauer, S. E., Bergman, T., Carslaw, K. S., Grini, A., Hamilton, D. S., Johnson, J. S., Karydis, V. A., Kirkevåg, A., Kodros, J. K., Lohmann, U., Luo, G., Makkonen, R., Matsui, H., Neubauer, D., Pierce, J. R., Schmale, J., Stier, P., Tsigaridis, K., van Noije, T.,Wang, H.,Watson-Parris, D.,Westervelt, D. M., Yang, Y., Yoshioka, M., Daskalakis, N., Decesari, S., Gysel-Beer,M., Kalivitis, N., Liu, X.,Mahowald, N.M., Myriokefalitakis, S., Schrödner, R., Sfakianaki,M., Tsimpidi, A. P.,Wu,M., and Yu, F.: Evaluation of global simulations of aerosol particle and cloud condensation nuclei number, with implications for cloud droplet formation, Atmospheric Chemistry and Physics, 19, 8591–8617, https://doi.org/10.5194/acp-19-8591-2019, https://www.atmos-chem-phys.net/19/8591/2019/, 2019.

Schmale, J., Baccarini, A., Thurnherr, I., Henning, S., Efraim, A., Regayre, L., Bolas, C., Hartmann,M.,Welti, A., Lehtipalo, K., Aemisegger, F., Tatzelt, C., Landwehr, S., Modini, R. L., Tummon, F., Johnson, J. S., Harris, N., Schnaiter,M., Toffoli, A., Derkani, M., Bukowiecki, N., Stratmann, F., Dommen, J., Baltensperger, U., Wernli, H., Rosenfeld, D., Gysel-Beer, M., and Carslaw, K. S.: Overview of the Antarctic Circumnavigation Expedition: Study of Preindustrial-like Aerosols and Their Climate Effects (ACE-SPACE), Bulletin of the American Meteorological Society, 100, 2260–2283, https://doi.org/10.1175/BAMS-D-18-0187.1, https://doi.org/10.1175/BAMS-D-18-0187.1,25 2019.

Spracklen, D. V., Carslaw, K. S., Pöschl, U., Rap, A., and Forster, P. M.: Global cloud condensation nuclei influenced by carbonaceous combustion aerosol, Atmospheric Chemistry and Physics, 11, 9067–9087, https://doi.org/10.5194/acp-11-9067-2011, https://www.atmos-chem-phys.net/11/9067/2011/, 2011.

Watson-Parris, D., Schutgens, N., Reddington, C., Pringle, K. J., Liu, D., Allan, J. D., Coe, H., Carslaw, K. S., and Stier, P.: In situ constraints on the vertical distribution of global aerosol, Atmospheric Chemistry and Physics, 19, 11 765–11 790, https://doi.org/10.5194/acp-19-11765-2019, https://www.atmos-chem-phys.net/19/11765/2019/, 2019.

**Referee Comment:** *4B: Inconsistent referencing practices.*

*Furthermore, the authors have many inconsistent referencing practices. For example, they include references from some instruments and projects, and do not include references for others. The authors should take some time to include relevant references throughout the manuscript. Some specific examples are listed below:*

*P3 L19: What is GME?*

**Author Response:** GME is the operational global icosahedralehexagonal gridpoint model, which was operationally used by the German Weather Service (DWD) previously to ICON. The reference Majewski et al. (2002) will be added to the manuscript. The abbreviation GME is a combination of its predecessor models GM (global model) and EM (a model for central Europe of DWD). Therefore, the abbreviation is not explained further and we refer to the reference.

Majewski, D., Liermann, D., Prohl, P., Ritter, B., Buchhold, M., Hanisch, T., Paul, G., Wergen, W., Baumgardner, J., 2002. The operational global icosahedralehexagonal gridpoint model GME: description and high-resolution tests. Monthly Weather Review 130, 319e338.

**Change in manuscript (p.3, l.33 - p.4, l.2 and bibliography):** COSMO is driven by initial and boundary data from GME re-analysis (the global model of DWD operational in 2013, Majewski et al., 2002).

**Referee Comment:** *P7 L22 - P8 L2: Reference should be given for ACTRIS and EMEP.*

**Author Response:** The references ACTRIS (www.actris.eu) and Tørseth et al. (2012) will be added to the manuscript.

Tørseth, K., Aas, W., Breivik, K., Fjæraa, A.M., Fiebig, M., Hjellbrekke, A.G., Lund Myhre, C., Solberg, S., Yttri, K.E.: Introduction to the European Monitoring and Evaluation Programme (EMEP) and observed atmospheric composition change during 1972–2009. Atmos. Chem. Phys. 12, 5447–5481 (2012).

**Change in manuscript (p.9, l.10-12 and bibliography):** Added references.

*Referee Comment: P8 L34: References should be given to substantiate what HOPE is about.*

**Author Response:** The reference Macke et al., 2017 for HOPE and its long name were given on page 2, line 31. The wrong brackets will be corrected in the revised manuscript.

**Change in manuscript (p.3, l.9-10):** Corrected brackets at first mentioning of HOPE.

*Referee Comment: P8 L9-12: References*

**Author Response:** The reference for ACTOS (Siebert et al., 2006) was given later in this paragraph and will be moved to this earlier mentioning of ACTOS. The reference for LACROS (Bühl et al., 2013) will be added to the manuscript.

Johannes Bühl, Patric Seifert, Ulla Wandinger, Holger Baars, Thomas Kanitz, Jörg Schmidt, Alexander Myagkov, Ronny Engelmann, Annett Skupin, Birgit Heese, André Klepel, Dietrich Althausen, and Albert Ansmann "LACROS: the Leipzig Aerosol and Cloud Remote Observations System", Proc. SPIE 8890, Remote Sensing of Clouds and the Atmosphere XVIII; and Optics in Atmospheric Propagation and Adaptive Systems XVI, 889002 (17 October 2013); https://doi.org/10.1117/12.2030911.

**Change in manuscript (p.9, l.20-23 and bibliography):** Added references.
Additionally, during the fall campaign, in-situ observations with the helicopter-borne platform ACTOS (Airborne Cloud Turbulence Observation System, Siebert et al., 2006) were combined cloud properties observed with remote sensing at the LACROS (Leipzig Aerosol and Cloud Remote Observations System, Bühl et al., 2013) supersite

*Referee Comment: P9, L12: Reference for this new instrument that the authors are using in the helicopter platform.*
*If there is no reference, the authors should include more details about this instrument here, such as its specifications and where it was placed on the helicopter. The authors provide a lot of details in terms of ground measurements, but provide no details about the helicopter measurements.*

**Author Response:** This mini CCN instrument is working by the same principle as the commercially available CCNC-100 by the DMT, which was used for the ground-based CCN measurements (Roberts and Nenes, 2005). It has also been used during other campaigns like in the CARRIBA campaign on Barbados (Wex et al, 2016).

Roberts, G. C., and A. Nenes (2005), A continuous-flow streamwise thermal-gradient CCN chamber for atmospheric measurements, *Aerosol Sci. Technol.*, *39*(3), 206-221.

Wex, H., et al. (2016), Aerosol arriving on the Caribbean island of Barbados: physical properties and origin, *Atmospheric Chemistry and Physics*, *16*(22), 14107-14130, doi:10.5194/acp-16-14107-2016.

**Change in manuscript (p.10, l.15-18):** Within this study we use the vertically resolved in-situ data of the light weight mini cloud condensation nuclei counter (mCCNc, custom built by Gregory C. Roberts, working principal as in Roberts and Nenes, 2005), which has been applied successfully on ACTOS before (e.g., Wex et al. 2016).

***Referee Comment:*** *Table 2: Is the reference for this only personal communication? This seems rather weak, and given that this is the basis of the 1985 estimates, I think the authors should supply a better reference. Who is Kevin Hausman, for whom this personal communication was with?*

**Author Response:** The countrywide emission estimates were gained directly in personal communication from the German Environment Agency (Umweltbundesamt, UBA). Our contact person was Kevin Hausmann. From the time before 1990, no reliable emission records for especially the German Democratic Republic ("East Germany") were available. Therefore, the emissions for the 1980s with the assumed peak emission year 1985 needed to be estimated by Umweltbundesamt for the area of today's Germany based on data of the Federal Republic of Germany ("West Germany"). Kevin Hausmann provided the annual emission sums as spatial average over entire Germany. This information will be added to the references list in the revised version.

**Change in manuscript:** Updated bibliography entry.

***Referee Comment:***

*Other Comments:*

*Qualitative, subjective explanations: In general, the authors make several subjective statements instead of providing qualitative results that would be more useful to the reader. I have included a few instances below.*

*P11, L4: The authors state that the analyses shown in Figure 2 are in good agreement. However, this is qualitative and subjective. Can the authors provide quantitative, objective measures to describe their results here?*

**Author Response:** This will be addressed in the revised manuscript.

**Change in manuscript (p.12, l.11 - p.13, l.7):** The whole paragraph was revised taking into account also other comments.

***Referee Comment:*** *P11, L16: "results compare satisfactorily." This is subjective.*

**Author Response:** On average the measured CCN concentrations are overestimated by 37 % (20.1 % without outliers) for the fall period, and underestimated by 29 % for the spring period by the model. We will add this information in the revised manuscript.

**Change in manuscript (p.12, l.11 - p.13, l.7):** The whole paragraph was revised taking into account also other comments.

***Referee Comment:*** *P11, L20: The authors use a 1 to 1 scatter plot to compare their results, and state that the model data underestimates CCN compared to the other methods. Can the authors add more quantitative details here. Underestimate by how much? Can the authors possibly do least squares fits for their data in Figure 3 to accomplish this or some other method in order to provide some more concrete results?*

**Author Response:** We calculated linear regressions. For the spring period (Fig 3a), a slope of 0.59 and 1.47 was found for the comparison of CCNc observations to CCN number concentration derived from modelled and gravimetrically measured aerosol mass, respectively. For the fall period (Fig 3b), the slopes are 1.21 (without the outliers) and 1.40, respectively. We will add this information to the plot and the text.

**Change in manuscript (p.14, l.3 - p.16, l.2):** The whole section was revised taking into account also other comments. Furthermore, regression lines have been added to Fig. 3 and the figure caption has been changed accordingly.

***Referee Comment:*** *P12, L5: "Figures 2 and 3 show that the CCN number concentration is in similar agreement." Can the authors quantify this result.*

**Author Response:** This sentence refers to the two different supersaturations of 0.2 and 0.3 % (Figs. 2 and A1). We also calculated the average deviation between directly measured and calculated CCN number concentrations for 0.3 % supersaturation. Comparing the CCN number concentration derived from modelled aerosol mass to CCNc observations we found an underestimation of 25 % for the spring period and an overestimation of 32 % for the fall period. These are comparable to the values in Fig. 2 (29 % underestimation and 37 % overestimation, respectively).
We agree, that the respective paragraph in the manuscript is hard to understand. Therefore, we will revise the whole paragraph and also add quantitative information.

**Change in manuscript (p.14, l.3 - p.16, l.2):** The whole section was revised taking into account also other comments.

***Referee Comment:*** *P13, L17-18: The authors state that the CCN number if overestimated by less than 50% and that this is quite well. This is subjective. Can the authors put this into context, in terms of how being 50% off would impact cloud processes or better explain why they think this is quite well?*

**Author Response:** The lidar-based CCN number concentration has a general uncertainty of a factor of 2-3 (Mamouri and Ansmann, 2016). Therefore, an overestimation (or better deviation) by 50 % is within this range. Furthermore, one cannot conclude that the model is 50 % off, but only that model and observation deviate from each other. Hence, the model estimate is in line with the observation. This is different for the comparison to the ACTOS profiles (which have an uncertainty of ~10 %). Here the model deviates from the observation outside its uncertainty range. The overestimation is up the factor of 2. In the revised manuscript we will not use the qualitative term "quite well" and re-write this section including quantitative measures of the deviation between model and observation.

**Change in manuscript (p16, l.5 - p.17, l.16):** The whole section was revised taking into account also other comments.

***Referee Comment:*** *P13, L19: "decrease considerably." How much is considerably?*

**Author response:** The word "considerably" is just meant to qualitatively describe the obvious vertical decrease of the CCN number concentration from boundary layer to free troposphere seen in Fig. 6 (starting at around 1.3 km height upwards). The important finding is rather that the observations show a sharp decrease at the boundary layer height, whereas the modelled CCN number concentration decrease rather smoothly with height. Therefore, above the boundary layer the deviation between lidar-based and modelled CCN number concentrations increases to more than a factor of 2.

**Change in manuscript (p16, l.5 - p.17, l.16):** The whole section was revised taking into account also other comments.

***Referee Comment:***

*Unclear discussions*

*P12, L5-8: The authors first state that the results are in similar agreement, but then state that the differences are a factor of 2, and then state the difference is about 13%. I found these statements to be quite confusing. They must be comparing different things, but I wasn't sure what was being compared. I think the authors should be careful about making it very clear what they are referring to throughout their manuscript, since there are a lot of different data being presented in this study.*

**Author Response:** The first statement, that the agreement is similar is referring to Fig 2 and A1. The main conclusion here is, that the comparison of modelled and observed CCN delivers similar results for both, 0.2 and 0.3% supersaturation.

The second statement refers to the peak deviation between CCN calculated from modelled and observed aerosol masses (Fig. 2 and A1 upper vs. lower panel). When averaging over the entire time period, a difference of about 13 % was found between the modeled CCN concentrations and the observed CCN concentrations with the CCNC. This can be seen from Table 4 as well (CCNC: $1.1*10^9$ $m^{-3}$, model (2013): $9.4*10^8$ $m^{-3}$).

We agree, that the paragraph should be revised to be less confusing and easier to read. Additionally, we will change Table 4 by moving the information of average measured CCN number concentration ($1.1*10^9$ $m^{-3}$) from the table caption to the table content.

**Change in manuscript (p.14, l.3 - p.16, l.2):** The whole section was revised taking into account also other comments. Table 4 now holds the average measured CCN number concentration, which previously was only given in the table caption.

*Referee Comment: P13, L17-18: The authors state that the ccn number if overestimated by less than 50%, but that seems to only be the case of the lidar measurements but not the ACTOS measurements. Can the authors clarify? A much different picture is present in the ACTOS measurements, which are not really discussed.*

**Author Response:** The lidar-based CCN number concentration has a general uncertainty of a factor of 2-3. Therefore, an overestimation (or better deviation) by 50 % is within this range. Furthermore, one cannot conclude that the model is 50 % off, but only that model and observation deviate. Hence, the model estimate is in line with the observation. This is different for the comparison to the ACTOS profiles (which have an uncertainty of ~10 %). Here the model deviates from the observation outside its uncertainty range. The overestimation is up the factor of 2. In the revised manuscript we will re-write this section including quantitative measures of the deviation between model and observation.

**Change in manuscript (p16, l.5 - p.17, l.16):** The whole section was revised taking into account also other comments.

*Referee Comment: P11, L25-32: One large uncertainty in this analysis and comparison revolves around the size distributions being used in the model conversion from mass to number and whether these are representative for this site. As such, I think this should be included here.*

**Author Response:** It is true that the mass-to-number conversion is a large source for uncertainty. It was, however, developed from data at Melpitz and compared to observations at this site (see Hande et al., 2016). Therefore, it can be considered representative for this site. We will mention the mass-to-number conversion as source for uncertainty in this paragraph.

**Change in manuscript (p.14, l.3 - p.16, l.2):** The whole section was revised taking into account also other comments.

*Referee Comment: P14, L12. The authors state that there is increased variability in CCN number conc. In the free troposphere, which I think is based on the increased 25%75% quartile range in Figure 5. However, the authors state this is "mainly an expression of the considerably increased detection uncertainty". However, the same trend is present in the model data, which does not have this detection uncertainty. Can the authority clarify why they believe this to be the case?*

**Author Response:** The decreasing trend of the median is observed in the model as well, but by far not as pronounced as in the lidar-based CCN observation. This observation method is more uncertain for small CCN concentrations as the lidar observations are close to detection limit. Therefore, the strongly increased variability is mainly due to the increased detection uncertainty and are thus statistically less relevant.

*Referee Comment:*

*Additional Specific Comments*

*P10, L22-24: Since SOA is so important to this region, why wasn't it included in the COSMO MUSCAT simulations?*

**Author Response:** By the time the simulations were conducted, the SOA module in COSMO-MUSCAT was under development and not sufficiently tested. Although, the available measurements at Melpitz point out a significant mass contribution of SOA in the PM1 (see Poulain et al., 2011), the calculated CCN concentration by applying the method described in our manuscript (applied size distributions; activation according to Abdul-Razzak and Ghan, 2000; a factor ~4 lower κ-value of OC compared to ammonium sulfate or nitrate) was not determined by OC (see also Figure 2 c and d). Therefore, we decided not to include SOA. The statement given in the text was meant to express that SOA is important for the total organic mass. We will write this section clearer in the revised manuscript.

**Changes in the manuscript (p.12, l.6-9):** The difference in ==CCN from== OC is ==partly== due to the absence of secondary organic aerosol (SOA) in the model approach. SOA ==generally can contribute a large fraction to== the total concentration of organic aerosol ==mass== (Jimenez et al., 2009) and also ==at Melpitz SOA is known to comprise a major fraction of the PM1 aerosol== (Poulain et al., 2011).

**Referee Comment:** *P2 L2: "For a realistic simulation of cloud adjustment..." What is cloud adjustment?*

**Author Response:** Cloud adjustment in the present context means the change of cloud macroscopic properties and dynamics due to perturbations of the aerosol. Whereas aerosol-cloud interaction describes the complex interplay between aerosols and clouds, cloud adjustments describe adjustments of clouds due to aerosol changes, such as the different aerosol loads between the 1980s and today over Europe. Examples of cloud adjustments are effects on cloud lifetime, cloud fraction or the timing and location of precipitation.

**Change in manuscript (p.3, l.8-9):** For a realistic simulation of ==microphysical aerosol-cloud-interactions and macroscopic cloud adjustment due to aerosol perturbations==, a detailed representation of the aerosol in the models is required.

**Referee Comment:** *P2 L12-14: I do not understand what the authors are describing with these two sentences.*

**Author Response:** We wanted to express that in the COSMO-MUSCAT version described by Sudhakar et al. (2017), an interactive (two-way) online coupling was implemented. Where the aerosol transport part is frequently updated with meteorological fields and the meteorological driver can utilize the aerosol information in order to describe the activation of aerosol particles. The activation of CCN is treated as power laws and therefore, does not consider aerosol microphysical properties.

**Change in manuscript (p.2, l.19-21):** ==This model version is online interactively coupled, making the activation of aerosol mass available for the two-moment scheme==. However, the aerosol ==activation uses the== bulk mass and does not explicitly consider ==aerosol== microphysical properties.

**Referee Comment:** *P2 L32: Why 1985? Can the authors provide background information here as to why one would be interested in the year 1985?*

**Author Response:** The year 1985 is just the middle of the 1980s, which are considered to be the period with highest aerosol concentrations in Germany. Already in the late 1980s, emission reductions were applied. In the text, "1985" is equivalent to "peak aerosol scenario" or "1980s". We will include a clarifying statement in this section in the revised manuscript.

**Change in manuscript (p.1, l.7 and p.3, l.3):** This is now stated in the abstract and introduction.

**Referee Comment:** *P3 L2: "This implies...". Can the authors clarify what this refers to? I am unclear and generally confused by this sentence?*

**Author Response:** "Implies" is the wrong word. Also, the sentence before was confusingly written.

**Change in manuscript (p.3, l.14-17):** The resulting modeled CCN fields can be used in atmospheric models that do not treat aerosol transport explicitly to analyze clouds and their radiation effects. For this purpose, CCN fields of variable degree of complexity can be generated, e.g., temporally and spatially constant CCN profiles, a 3D CCN field as a long-term average or even a 4D CCN field for temporally limited episodes.

**Referee Comment:** *P3 L4-9: The authors state they will be comparing 5 CCN estimates, but they seem to be estimates of different things. However, it is unclear why they are comparing these 5 items? As this is the introduction to the authors study, the authors should make this clear. What is the ultimate goal of this comparison and study?*

**Author Response:** Basically, with this list we aimed to give an overview of the CCN datasets from the different sources (model, chemical aerosol measurements, in-situ measurements) that were compared in this study. In short, the items are:
- CCN from the model for 2013 simulation

- CCN from the model for 1985 simulation ("peak aerosol scenario")

- CCN from chemical aerosol measurements

- ground-based in-situ CCN observation

- vertical CCN profiles from in-situ and remote sensing observations

We will re-structure this part of the text in the revised manuscript and make the description of the five datasets clearer.

The aim of the study is to provide estimates of the CCN concentrations in the 1980s using an offline method and compare to simulations and observations in the year 2013. The description of the study aims will be better summarised in the revised manuscript.

**Change in manuscript (p.2, l.2 - p.3, l.26):** The introduction was revised giving the aim of the study and an overview of the methods.

**Referee Comment:**

*P3 L18: What is a composition cycle?*

*P3 L19: The authors state that the simulations will be reinitialized every 48 hours, and provide other details, without yet describing the basics of the simulation. Therefore, it is difficult to understand why a 48 initialization time is reasonable. It is suggested that the authors provide more details at the beginning of this section to make this section more readable.*

**Author Response:** We agree, that the word "composition cycle" is unclear. COSMO and MUSCAT are coupled online, i.e., MUSCAT is updated with meteorological fields every time step. COSMO is initialized with coarser simulations and is updated only at the boundaries in order to make use of the higher resolved grid. This is also why COSMO is always run with a 24 h spin-up before the coupling to MUSCAT is switched on. After the spin-up period, COSMO and MUSCAT run coupled for another 48 h. In order to keep modelled meteorological fields close to the real atmosphere in these hindcast applications, we re-initialize the simulation cycle every 48 h, i.e. after 72 h for COSMO. We will improve this description in the revised manuscript.

**Change in manuscript (p.3, l.30 - p.4, l.11):** The whole section was revised.

**Referee Comment:** *P4 L8: Why do the authors only use model data up to 8km? The authors should provide a reason to justify or explain why they are not using the full model data?*

**Author Response:** The model simulations use more vertical levels (50) up to a height of 22 km. For the analysis, only the lowest 32 layers (8 km) were saved since we were mainly interested in the lower troposphere. We agree that the information is misleading and we will delete the sentence in the revised manuscript, and instead give the actual number of vertical levels in the simulation.

**Change in manuscript (p.4, l.24):** In the vertical, the model treats 50 layers up to a height of 22km.

*Referee Comment: Figure 1: Why is only Melpitz shown, when Julich is also mentioned? Were measurements taken at the different city, Julich? I was confused about this throughout the manuscript.*

**Author Response:** In the present study, most of the data that was used was measured at Melpitz. From the observations at Jülich, only the derived CCN concentrations from lidar measurements were taken into account. In the revised manuscript we will change to using the terms "spring" and "fall period" instead of HOPE Jülich and HOPE Melpitz and include the Jülich site in Fig. 1.

**Change in manuscript (Figure 1 and p.4, l.21-22):** The caption of Fig. 1 was revised and the Jülich site is now shown in Fig. 1.

In addition, lidar-based CCN concentrations were available during the spring campaign in Jülich, Germany.

*Referee Comment: P4 L2: "in order to be considered" for what?*

**Author Response:** We mean that information on aerosol and CCN needs to be prescribed in order to be used for ICON-LES simulations, e.g., to alter cloud properties.

**Change in manuscript (p.4, l.15-17):** In the ICON-LEM (ICOsahedral Non-hydrostatic Large Eddy Model; Zängl et al., 2015; Dipankar et al., 2015; Heinze et al., 2017), which is the model used in HD(CP)[2], there is no online aerosol transport scheme, which indicates the need of prescribing the aerosol and CCN concentrations in order to be considered for aerosol-cloud interaction.

*Referee Comment: P4 L3: The authors state the model simulations were run. Are these ICON LES simulations, which were just mentioned in the previous sentence or COSMO MUSCAT simulations? I think it is the COSMO MUSCAT simulations, but can the authors make this more clear?*

**Author Response:** Yes, we used COSMO-MUSCAT. The calculated CCN fields were provided for ICON-LEM simulations within the framework of the HD(CP)[2] project. We will make this clearer in the revised manuscript.

**Change in manuscript (p.4, l.18-20):** In order to provide time varying 3D fields of CCN concentrations for ICON-LEM, model simulations with COSMO-MUSCAT covering most of Germany have been carried out for the time period of two intensive measurement campaigns during HD(CP)2: HOPE.

*Referee Comment: P4 L17-18. It seems that the authors imply that total mass concentrations is converted to total number concentrations before the log normal size distribution parameters are set? However, these size distribution would be necessary for the initial conversion to mass to number. So I am ultimately confused by what the authors are actually doing?*

**Author Response:** The process of activation is applied in two steps. First the aerosol particle number size distribution is calculated. Then these size distributions are used in the activation parametrization. We will write this in a clearer manner in the revised manuscript.

**Change in manuscript (p.5, l.1-3):** For ==the== externally mixed aerosols, the total number concentration ==of each species is== calculated from the modeled mass ==of the aerosol species assuming an individual geometric mean radius and standard deviation.==

*Referee Comment: P5 L7: Which of the values are not according to Hande et al. (2016) and why?*

**Author Response:** The value of kappa for sulfuric acid was changed. In Hande et al (2016), a kappa value of 0.236 was assumed, which is far too low. Petters and Kreidenweis (2007) state mean values of kappa of 1.19 (growth factor derived) and 0.9 (CCN derived) for sulfuric acid. Based on these values, a kappa of 1 was chosen for this study.

*Referee Comment: P5, L14: The authors mentioned that ammonium sulfate has a kappa of 0.6, but then use 0.51 in their table. Why?*

**Author Response:** What was meant is that kappa of ammonium sulfate is around 0.6. In Petters and Kreidenweis, there are values from 0.33 up to 0.72 for the growth factor derived kappa and 0.61 for the CCN derived kappa. For consistency, we decided to use the same value as in the previous paper Hande et al. (2016) (taken from Ghan et al., 2001).

*Referee Comment: P5, L33-34:    this should be included in the introduction.*

**Author Response:** We agree and will add this information in the introduction.

**Change in manuscript (p.2, l.2 - p.3, l.26):** The introduction was revised now giving information about why we choose the 1980s and the year 1985 in particular.

*Referee Comment: P5, L7: The authors reference Table 1, but mean to reference Table 2.*

**Author Response:** The sentence is about the parameters of the mass-to-number conversion and therefore correctly references Table 1. During the revision, we now found that on the original p.6, l. 7 the table reference was wrong. This is probably the position the Referee meant.

**Changes in the manuscript (p.6, l.34):** Corrected reference to Table 2.

*Referee Comment: Table 4: How were the measured aerosol mass concentrations obtained? Can the authors include this information in the caption, similar to how they mention that the modeled concentrations came from the COSMO MUSCAT simulations.*

**Author Response:** Yes, we will add according information about the gravimetrical measurements in the table caption.

**Change in manuscript:** Revised caption of table 4.

*Referee Comment: P10, L16-19: The authors discuss why the measured dust was larger than the modeled dust. Can the authors make these statements more clear. For example, why is it OK to assume the difference in total mass from the measured species must be from dust?*

**Author Response:** In the gravimetrical measurements, different species, such as sulfate, nitrate, organic and black carbon, can be detected. The undetectable rest of the mass is generally accounted as "other" dust, i.e. just particulate mass which cannot be further speciated with standard methods. The model does only consider mineral dust from deserts. Additional primary dust sources, such as road dust, are not considered. Therefore, the model tends to underestimate the observed dust concentrations. Overall, the modeled and observed dust contributes only little to the total CCN concentration (see Fig. 2).

*Referee Comment: P11, L8: The authors state the nitrate was overestimated by a factor of 2. Can the authors present this information more clearly, possible on the same figure to make this more clear.*

**Author Response:** The statement refers only to the 3-day period (day of the year 255 – 257), which show a strong overestimation of the ammonium nitrate mass, and therefore CCN from ammonium nitrate (Fig. 2b and 2d). The difference between modelled CCN and CCN from chemical measurements of about a factor of two during those days can clearly be seen (Fig. 2b and 2d). Also, it can be seen that during those days, ammonium nitrate contributes most to the CCN in both observation and model. As for the CCN, the overestimation of ammonium nitrate is about a factor of 2 (CCN from ammonium nitrate: 1.0-1.5 x $10^9$ m$^{-3}$ for the gravimetrical measurements and 1.5-3.0 x $10^9$ m$^{-3}$ for the modelled CCN).

**Change in manuscript (p.12, l.11 - p.13, l.7):** The whole paragraph was revised taking into account also other comments.

*Referee Comment: P11, L8-16: A majority of the discussion is focused on a 3 day period, which is very difficult to see in Figure 2. Can the authors create a new figure that zooms in on this period, to allow the reader to more easily follow along in the figure.*

**Author Response:** We did not intend to focus much on this period. However, we found it necessary to discuss this interesting, yet not important, feature. We will extend the general discussion on the findings displayed in Fig. 2.

**Change in manuscript (p.12, l.11 - p.13, l.7):** The whole paragraph was revised taking into account also other comments.

*Referee Comment: Figure 6: Why isn't the fall period shown here?*

**Author Response:** The fall period is shown in Fig. 6b. To make it clearer, we will add this information also in the figure itself and not only in the figure caption.

**Change in the manuscript (Fig. 6 and p.16, l.5-8 ):** Updated Figure 6 and changed text:

Fig. 6 compares the simulated and observed vertical profiles of the CCN number concentration over Melpitz for the two periods in 2013. Fig. 6a shows the comparison to CCN derived from lidar observation during the spring period at Jülich, and Fig. 6b the comparison to the in-situ observations by the helicopter-based platform ACTOS during the fall period at Melpitz.

*Referee Comment: P14, L8. The authors state that for this calculation a vertical velocity of 1 m/s is used. I don't understand what this is referring too.*

**Author Response:** In contrast to the previous analysis in the manuscript that compared CCN at a fixed supersaturation to observations, for the last discussion part we calculated the CCN at a fixed vertical velocity (1 m s$^{-1}$). Following Abdul-Razzak and Ghan (2000), the maximum supersaturation can be calculated assuming an air parcel rising adiabatically with a given vertical velocity. This maximum supersaturation defines the critical radius of the given size distribution and therefore the number of particles activated to CCN. The maximum supersaturation, hence, depends on the particle composition and the number size distribution. The chosen value of 1 m s$^{-1}$ serves as an example in order to compare the present day and 1980s scenario. In a further model development step, the aerosol composition could now be used to alter cloud microphysical properties under the given modeled grid or sub-grid scale vertical wind velocities.

*Referee Comment: Figure 7: What supersaturation is used for this analysis?*

**Author Response:** Fig. 7 shows the CCN number concentration for an updraft velocity of 1 m s$^{-1}$. This is mentioned in the text (p.14, l.8). We will add this information also in the figure caption. The supersaturation therefore depends on the aerosol chemical composition of the aerosol and the

number size distribution and varies spatially and temporally (see also answer to previous comment).

**Changes in the manuscript (p.18, l.4-5)**: For the calculation, a vertical velocity of 1 m s$^{-1}$ was assumed. Hence, in contrast to the previous analysis the supersaturation depends on the aerosol composition and varies spatially and temporally.

And revised figure caption of Fig. 7.

*Referee Comment:* *Can the authors changes their units to cm$^{-3}$ from m$^{-3}$, such that it is easier to comprehend the values presented?*

**Author Response:** Thank you for this remark. "cm$^{-3}$" are widely used, but we prefer SI units, which is also according to ACP guidelines.

[revised manuscript text omitted]

---

## Referee Report (RR1)

Review of Estimation of Cloud Condensation Nuclei number concentrations and comparisons to in-situ and lidar observations during HOPE experiments by Genz et al.
acp-2019-742

**Summary:** In this study, the authors use a meteorology model, coupled to an aerosol model that predicts aerosol mass concentrations over the course of several months in Germany where observations are present at two sites within their domain. The aerosol mass concentrations that are simulated in the models are converted to CCN using assumed size distributions, and these CCN concentrations are compared to estimates of CCN from various in situ and remotely-sensed observations in Germany during the same time period. Furthermore, the authors apply scaling factors to the emissions representative of the 1980s, and run the 2013 simulations again with these new emissions to get a sense of what the CCN concentrations were in 1980s for comparison. The authors utilize a wide range of methods to quantify CCN, and a comparison of such methods are useful. However, more details on why these different methods vary would provide a much more compelling study and would improve the interpretation and understanding of not only the 2013 results but also the 1985 results. Furthermore, there are also some critical details that are missing and some missing references. As such, I think this study is of interest to the wider community, but needs additional analyses and more details as discussed below.

**Overarching concern**

The main focus of this study is on the use of modelled CCN estimates. However, the authors only speculate reasons why their modelled CCN estimates vary from observations (e.g., ammonium nitrate uncertainties, precipitation impacts, particle size distributions, particle composition) and do not provide concrete results explaining why their modelled estimates compare as they do to various observations. All of these speculations could be actually tested in their framework. Understanding the reasons for their 2013 model-observation comparison biases would be especially helpful in assessing the robustness in their 1985 estimates, for which there are no observations and which are based on meteorology from 2013. Ultimately, I am left questioning on how accurate are these 1985 profiles

As such, I think this manuscript can be improved significantly by providing focused analyses on why their model underestimates and overestimates CCN. These include:

1) Assessment of precipitation and surface winds in the model and at their observation site. The authors state that for a 2 day period, the overestimated in the model compared to observations was due to precipitation not being correctly located in their model as compared to observations. Therefore, this potential problem of comparing the model and observations at one fixed site could help explain a lot of overestimation or underestimation of CCN, and if so, then more credence can be given to the various emissions and aerosol assumptions used in this model and post-model processing.

Similarly, airmass trajectories and the advection of aerosol to their specific site could also lead to similar biases and should be tested.

2) The authors state several times throughout the manuscript that there are large uncertainties likely due to the fact that they assumed a fixed PSD for their aerosol size distribution that is required to estimate CCN. It seems reasonable and possible for the the authors to test this assumption, by using a few other PSDs to understand the magnitude of this sensitivity, which would allow for more scientifically rigorous conclusions. This seems computationally feasible as the calculations are run offline. Specifically, the authors note that there PSDs likely have too few particles below 100nm, so at least the authors should run a test with more particles in this part of the PSD.

3) Why does the aerosol vertical profile of CCN change between the 1985 and 2013? There is no clear explanation given.

4) Why was the meteorology from 1985 not taken into account? Is there no meteorology data available from that time period? How can we know that the differences shown in the authors in their simulated 2013 and their simulated 1985 (using meteorology form 2013) would not be different if they actually used meteorology from 1985? The authors even state in their conclusions that the dynamics and thermodynamics (e.g., meteorology) have a large influence on the CCN distribution. Given that the authors treat their 1985 CCN profiles are realistic representations that can be used in future studies and assess the role of emissions reductions between these years, there should be some at least some support, either references or some analysis of the meteorology from these time periods that justify not considering the meteorology.

**Specific Comments / Questions**

P1, L8: Can the authors extend their CCN values to include a few decimals points, since the units are $10^9$ m$^{-3}$.

P1, L13-14: 'since chemical composition and the size distribution are less important in these ranges'. This was not shown in this manuscript from my understanding and should be excluded from the abstract.

P1, L16: What does 'mid-supersaturation regime' mean? It is suggested the authors use actual numbers here to make it more clear.

P2, L2: It may be a good idea to include the word 'Europe' in the first sentence to make this statement accurate and clear.

P2, L4: Should the last sentence of the first paragraph have a reference, possibly the Cherian et al. 2014 reference?

P2, L5-6: Aerosol particles play an important role in the microphysical processes of cloud formation, should have a reference. I believe the proper, original reference for this is: Köhler, 1936.
Köhler, H. (1936), The nucleus in and the growth of hygroscopic droplets, Trans. Faraday Soc., 32(1152), 1152–1161, doi:10.1039/TF9363201152.

P2, L26: The statement says the particle size distribution were calculated using an offline using the Abdul-Razzak and Ghan parameterization. Is this true? I thought the PSDs were assumed, not calculated. Please check.

P2, L31: It is suggested that the authors include some earlier, more original studies of CCN observations. For example, Squires and Twomey, 1966.
Squires, P., and S. Twomey (1966), A Comparison of Cloud Nucleus Measurements over Central North America and the Caribbean Sea, *J. Atmos. Sci.*, *23*(4), 401–404, doi:10.1175/1520-0469(1966)023<0401:acocnm>2.0.co;2.

P2, L32-33: Again there are earlier, more original references that should be considered. For example, Feingold et al. 1998.
Feingold, G., S. Yang, R. M. Hardesty, and W. R. Cotton, 1998: Retrieving cloud condensation nucleus properties from Doppler cloud radar, microwave radiometer, and lidar. J. Atmos. Oceanic Technol.,

P3, L12: The authors state that the activation of aerosol particles depends on their number, which is not true. The activation of aerosol particles only depends on size, composition, and the amount of supersaturation present.

P4, L3: Is there then a discontinuity in the simulation data every 48 hours when the model is re-initialized?

P5, Figure 1. The Julich site is right next to the model domain boundary. If I understand the model set-up correctly, this is a significant concern since if there are winds coming from the west, the aerosol concentrations in the model could be very inaccurate.

P5, L10: The authors state that particles between 50 and 200 nm are most relevant for estimating CCN. This is not strictly true since all aerosol particles can form CCN. Can the authors be accurate with their statements? For example, maybe here the authors mean that particles between 50 and 200nm are most relevant for estimating CCN for supersaturations between some supersaturation range? For shallow clouds with very low supersaturations, the sizes of CCN particles that are relevant may indeed be larger. Similarly, for high supersaturations, the relevant particle sizes may be lower.

P6, L10: What does "as written above" refer to? I found this paragraph difficult to follow and suggest the authors revise it to make it more clear.

Table 1: Instead of putting the references in the caption, can the authors put the relevant references next to each species? This would allow a reader who was interested in understanding one of the species assumptions to easily find that without having to look through all of the references listed in the caption.

P11, L30: What causes the significant differences (5-8x different) in OC, SS and DU between the 1985 and 2013 modeled values? Since the emissions do not change, it would be very interesting to understand why these other species change and provide better interpretation of these results.

P12, L7-9: What does 'large' and 'major' mean? Can the authors be more specific and include numbers?

P12, L18: Why were ammonium sulfate and nitrate underestimated in the model in the first half of spring?

L12, L19: It is unclear what the "factor of 5" is referring to. I am assuming CCN concentrations, but the way it is written it sounds like ammonium nitrate or ammonium sulfate or both?

P13, L1-5: The authors speculate that the overestimation in the modeled CCN may be due to ammonium nitrate uncertainties. However, then they state that actually for a two day period it was due to differences in precipitation and wet deposition. What is the primary cause for the consistent overestimation?

Figure 2c,d: There is very little discussion in the manuscript about the comparisons of the CCN based on the gravimetric observations as compared to the CCNC, which is shown in Figure 2c,d.

P16, L2: "More important" than what?

P16, L12: "observed and measured" -- Are these meaning the same thing?

P18, L4: The authors state that for this analysis a vertical velocity of 1 ms$^{-1}$ was assumed. All the prior analyses had fixed supersaturations, while for this analysis, the authors instead used a fixed vertical velocity. Why did this change?

P18, L4-5: The authors state that "in contrast to the previous analysis, the supersaturation depends on the aerosol composition and varies spatially and temporally." I think additional statements are needed here or upfront in the manuscript to make this more clear.

P18, L13-14: Why is the difference in shape of the profiles due to differences in aerosol composition (hygroscopicity)? How was this determined, and how did the authors rule out other differences (i.e., differences in aerosol chemistry, differences in meteorology)? There should be much more explanation here.

P18, L15. What do the authors mean when they say the scaling factor represents a mean trend? Are they suggesting there is a linear decrease between the two years? I am not sure the authors can state anything about a mean trend with two points.

P20, L18-19: In their conclusions, the authors state that the thermodynamics and dynamics of the tropopause has a large influence on distribution of aerosol and the vertical profile of CCN. However, the authors state in their manuscript that this was due to aerosol composition hygroscopicity (P18, L13-14). Can the authors clarify what is meant?

P20, L19-20: One of the main results is the differences in the vertical structure of CCN between 1985 and 2013, and the authors state that this vertical structure up to 5km is important for cloud microphysical processes without any references. The two studies that have assessed this are Lebo 2014 and Marinescu et al. 2017 and should be included to support this statement.
Lebo, Z. J. (2014), The Sensitivity of a Numerically Simulated Idealized Squall Line to the Vertical Distribution of Aerosols, *J. Atmos. Sci.*, *71*(12), 4581–4596, doi:10.1175/JAS-D-14-0068.1.
Marinescu, P. J., S. C. van den Heever, S. M. Saleeby, S. M. Kreidenweis, and P. J. DeMott (2017), The Microphysical Roles of Lower-Tropospheric versus Midtropospheric Aerosol Particles in Mature-Stage MCS Precipitation, *J. Atmos. Sci.*, *74*(11), 3657–3678, doi:10.1175/JAS-D-16-0361.1.

Figures 6 and 7. Can the authors make these figures wider, so that readers can more easily see the values that are represented by the lines? Currently, it is very hard to see.

---

## Author Response (AR2)

**Answers to Referee comments of the revised manuscript**

*Summary*

*Referee Comment: In this study, the authors use a meteorology model, coupled to an aerosol model that predicts aerosol mass concentrations over the course of several months in Germany where observations are present at two sites within their domain. The aerosol mass concentrations that are simulated in the models are converted to CCN using assumed size distributions, and these CCN concentrations are compared to estimates of CCN from various in situ and remotely-sensed observations in Germany during the same time period. Furthermore, the authors apply scaling factors to the emissions representative of the 1980s, and run the 2013 simulations again with these new emissions to get a sense of what the CCN concentrations were in 1980s for comparison. The authors utilize a wide range of methods to quantify CCN, and a comparison of such methods are useful. However, more details on why these different methods vary would provide a much more compelling study and would improve the interpretation and understanding of not only the 2013 results but also the 1985 results. Furthermore, there are also some critical details that are missing and some missing references. As such, I think this study is of interest to the wider community, but needs additional analyses and more details as discussed below.*

**Author Response:** The authors would like to thank the anonymous Referee for the comprehensive and critical review, which is helpful in order to improve the manuscript. We have addressed all concerns in this document. Part of the answers of the more general comments can be found at the specific comments in more detail. According to the referee's comments, we have revised the manuscript. All changes in the manuscript are addressed in the following. Text marked in yellow was added in the revised manuscript.

*Overarching concern*

*Referee Comment: The main focus of this study is on the use of modelled CCN estimates. However, the authors only speculate reasons why their modelled CCN estimates vary from observations (e.g., ammonium nitrate uncertainties, precipitation impacts, particle size distributions, particle composition) and do not provide concrete results explaining why their modelled estimates compare as they do to various observations. All of these speculations could be actually tested in their framework. Understanding the reasons for their 2013 model-observation comparison biases would be especially helpful in assessing the robustness in their 1985 estimates, for which there are no observations and which are based on meteorology from 2013. Ultimately, I am left questioning on how accurate are these 1985 profiles*

**Author Response:** The aim of the study was to provide estimates of mid 1980's CCN concentrations over Central Europe and Germany, and not exactly for the year 1985. Apart from uncertainties due to assumptions in the model (e.g., size distribution), the emissions of the 1980's over Germany are very uncertain as there are only insufficient emission records for Eastern Germany for that time. Therefore, it is not realistic to aim for exact concentration estimates of a particular year in the 1980's. The study took part in the framework of the HD(CP)$^2$ project and had the task to provide 3D time-varying CCN fields as input for high-resolution simulations with ICON-LEM. Although, the magnitude of the estimated CCN number concentrations might be uncertain especially for the 1980's scenario, the regional and temporal variations are expected to introduce significantly more realism to the ICON simulations of the partner study (Costa-Surrós et al., 2020) than the previously applied spatially and temporally constant CCN profiles (Heinze et al., 2017).

The applied size distributions were already explained, discussed and compared to measurements great detail in our previous work by Hande et al. (2016) (see also answer to later comment for more details). Any other size distribution assumptions would be less

constrained by observations for this particular site. The effect on the CCN concentration of deviations of the modeled and measured chemical composition (but using the same size distributions to calculate CCN from the modeled and observed mass, respectively) is shown in Figs. 2 and 3. Far from any speculation, we clearly show that the model underestimates the total CCN concentration at least partly due to an underestimation of aerosol mass, mainly by ammonium nitrate and ammonium sulfate, in the spring episode and overestimates CCNs because of an overestimation of these aerosol species in the fall episode. The underestimation of ammonium sulfate and particularly ammonium nitrate mass in the spring episode is also shown in Fig. 2 in the work by Hande et al. (2016) (see below).

[Figure]

Ammonium sulfate and ammonium nitrate concentration for the spring episode. The figure is taken from Hande et al. (2016) (Figure 2) with permission of the authors.

Furthermore, the size distribution assumptions are evaluated against in-situ observations of CCN in section 3.2. This section also provides a comparison of the N_CCN/N_CN ratios, which give insight to both the number of particles in a certain size range and the resulting CCN at a defined supersaturation. From this investigation, it can be seen (Fig. 4) that there is no difference in the comparison to the observation between the spring and fall episode. Hence, the different under- and overestimation of the CCN concentration between the spring and the fall episode seen in Fig. 2 is more likely linked to uncertainties of the modeled aerosol mass than the assumptions made to derive CCN. The behavior during the comparison for different combinations of N_CN thresholds and supersaturations for N_CCN can be explained by the known deviations of the assumed size distributions to the observed ones (N_CN is underestimated by 10 % for particles > 110 nm and 35% for particles > 80 nm).

The recently published paper by Costa-Surós et al. (2020) includes a comparison of the modeled (same simulation data) and satellite-borne observed aerosol optical thickness (AOT). For both the 2013 period and the same time period in 1985, the average AOT in model and observation agree well. That means in particular that the model using the assumptions discussed in the present manuscript is able to represent the clean conditions of the year 2013 and the much more polluted atmosphere of the 1980's. Therefore, it can be concluded that the average total aerosol load is represented sufficiently well also for the 1980's conditions.

However, we are aware that these size distribution assumptions are likely not completely valid for the whole domain and time period in 2013, as well as for the 1980s. This is certainly a shortcoming which is difficult to overcome and to evaluate. The evaluation against today's in-situ observation shows an average deviation for the two periods between -29% to +37%. Further investigations using size-resolved aerosol microphysics would avoid assuming temporally and spatially constant size distributions. Such is planned for the future but is beyond the scope of this study.

Overall, we believe that, despite the simplifying assumptions, this study applying the current state of our model system provides valuable information on the CCN budget in the 1980s, which are of interest for the broader scientific community.

**Change in manuscript:** The introduction has been revised, now providing more information on the motivation and goal of this study as well as the role within the HD(CP)$^2$ project. The label "1985" is replaced by "mid 1980's". Sections 3.1 and 3.2 were revised giving a clearer discussion of the deviations between modeled and observed CCN concentrations due to potential uncertainties of the modeled aerosol mass and the assumption of the number size distribution. More detailed answers and discussion as well as respective changes in the manuscript are addressed at the specific comments below. The comparison of AOT conducted by Costa-Surós et al. (2020) is mentioned in the manuscript.

*Referee Comment: As such, I think this manuscript can be improved significantly by providing focused analyses on why their model underestimates and overestimates CCN. These include:*

*1) Assessment of precipitation and surface winds in the model and at their observation site. The authors state that for a 2 day period, the overestimated in the model compared to observations was due to precipitation not being correctly located in their model as compared to observations. Therefore, this potential problem of comparing the model and observations at one fixed site could help explain a lot of overestimation or underestimation of CCN, and if so, then more credence can be given to the various emissions and aerosol assumptions used in this model and post-model processing. Similarly, airmass trajectories and the advection of aerosol to their specific site could also lead to similar biases and should be tested.*

**Author Response:** As already stated in the manuscript, the misrepresentation of the amount and exact location of precipitation in the COSMO model has been noted in the manuscript. The large-scale air mass transport in the model is expected to be realistic since the model is initialized and driven through the lateral boundaries by reanalysis data for both meteorology and atmospheric chemical composition. A thorough trajectory analysis would be helpful to partly explain the influence of different emission sources on the sampling site, but this is beyond the scope of the current investigation. Nevertheless, as standard procedure during the analysis we carefully go through the simulation results of key species and meteorological fields. In addition to the chemical measurements, also the timeseries of meteorological observations at the measurement station Melpitz were compared to the modeled variables. At the station itself, no meaningful discrepancies were found of meteorological variables were found.

During the mentioned short period the ammonium nitrate reaching the measurement site originated from North-western Germany and The Netherlands, a region with very intense agriculture (see figure below). Especially in the source region, the model underestimated the precipitation and, hence, wet deposition of ammonium nitrate, ammonium sulfate and its precursors on 2013/09/11 and 12 (see figure below). The air mass rich in ammonia and later ammonium nitrate travelled towards the measurement site at Melpitz during the next 2 days, which were the ones where the overestimation was observed.

It was not intended to deeply discuss the mentioned 3-day period since it covers only a short part of the overall simulation period of several months. However, since the overestimation of particularly ammonium nitrate was rather high during these three days, we decided to mention it in a paragraph. From the analysis of air mass transport, ammonium nitrate formation, and precipitation over Germany, we could conclude that at least part of the overestimation can be explained by lack of wet deposition on 2013/09/11 and 12 and subsequent enhanced formation of ammonium nitrate and ammonium sulfate. However, the model is here not completely misrepresenting the situation since also in the gravimetrical aerosol measurements a strong peak could be observed. Therefore, we assume that the general

interplay of emission, formation, and transport is reasonable. However, it cannot be explained which of the processes leads to the overestimation to which extent. A likely major reason presents the missing wet deposition in the ammonia source region in the North-western part of the domain, since observed large amounts of precipitation were not modeled there.

**Change in manuscript:** The section about the 3-day period has been revised including more discussion on the potential causes.

[Figure]

Evolution of the modelled ammonium nitrate mass from 2013/09/12 – 14.

[Figure]

Modelled and observed precipitation sum (24 hours) of 2013/09/11 and 2013/09/12.

**Referee Comment:** *2) The authors state several times throughout the manuscript that there are large uncertainties likely due to the fact that they assumed a fixed PSD for their aerosol size distribution that is required to estimate CCN. It seems reasonable and possible for the the authors to test this assumption, by using a few other PSDs to understand the magnitude of this sensitivity, which would allow for more scientifically rigorous conclusions. This seems computationally feasible as the calculations are run offline. Specifically, the authors note that there PSDs likely have too few particles below 100nm, so at least the authors should run a test with more particles in this part of the PSD.*

**Author Response:** A comparison between observed and measured particle size distribution was already done by Hande et al., 2016, who provide the assumed size distributions. These are based on AMS measurements of ambient concentration of the individual species (Poulain et al., 2011). A comparison of the estimated size distributions to observations can also be found in Hande et al. (2016). As can be seen in their Fig. 3c (see below), the observed average aerosol size distribution at Melpitz is indeed bimodal with peaks at ~30 nm and ~100 nm diameter. The combined size distribution of the different species was found to match the observed size distribution well around 100 nm (i.e., the peak region of the largest mode), which is most relevant for estimating CCN. However, the number of aerosols in the size ranges (diameter) of 200 nm – 1 µm and < ~100 nm is underestimated.

[Figure]

Comparison PNSD, obs. vs. sim., Melpitz 04/2013

Simulation (median, 25 and 75 % quantiles)
Observation (median, 25 and 75 % quantiles)

Modelled and observed particle number size distribution at Melpitz during April 2013. The figure is taken from Hande et al. (2016) (Figure 3c) with permission of the authors.

The following table presents a brief sensitivity for the assumed size distribution assumptions for the example of 1 µg m$^{-3}$ ammonium sulfate, which is a main driver of CCN number concentrations in our simulation. Since the five supersaturations are fixed (no competition for water vapor), the critical radius is independent of the size distribution parameters. First of all, the change of size distribution parameters affects the mass-to-number conversion and therefore the total particle number of this monomodal aerosol distribution. By varying the geometric mean radius by +/- 10 %, the CCN concentration at 0.2 % supersaturation varies by +/- ~15 % in the particular case. The effect is larger for the higher supersaturations shown here. Widening the distribution causes a ~25 % decrease in CCN number concentration at 0.2 %. The effect is different for the aerosol mixture, which changes both temporally and spatially.

| r [µm] | σ | N$_{tot}$ [cm$^{-3}$] | N$_{CCN}$ [cm$^{-3}$] at supersaturation of | | | | |
|---|---|---|---|---|---|---|---|
| | | | 0.1 % | 0.2 % | 0.3 % | 0.5 % | 0.7 % |
| | | | which are equivalent to critical radii [µm] of | | | | |
| | | | 0.077 | 0.048 | 0.037 | 0.026 | 0.021 |
| 0.05 | 1.6 | 409 | 74 | 215 | 302 | 373 | 395 |
| 0.045 | 1.6 | 560 | 71 | 246 | 371 | 490 | 531 |
| 0.055 | 1.6 | 306 | 73 | 186 | 246 | 289 | 301 |
| 0.05 | 1.7 | 310 | 65 | 163 | 223 | 276 | 295 |

Hence, keeping the mass that is supposed to be distributed over the particle size distribution and the chemical composition constant, the underestimation of particle number in the diameter range 200 nm – 1 µm (i.e., a too narrow distribution, or shifted to smaller sizes than reality) results in an overestimation of CCN at fixed supersaturation. The underestimation of particle number of particles with diameters < ~100 nm (i.e., distribution shifted towards larger sizes than reality) would result in an underestimation of CCN at fixed supersaturations (in the investigated range).

Also, in the current manuscript, the modeled particle number and CCN number concentration was compared to the in-situ observation at Melpitz. As stated in the text (p.15, l.4-6), the number of particles larger than 110 nm is underestimated by 10 %, and the number particles

larger than 80 nm is underestimated by 35 %. Furthermore, the N_CN / N_CCN ratios that were investigated in section 3.2 provide a comparison to in-situ observations and take into account the effect of the chosen size distribution assumptions.

**Change in manuscript:** Sections 3.1 and 3.2 were revised now giving a clearer discussion of the deviations between modeled and observed CCN concentrations due to potential uncertainties of the modeled aerosol mass and the assumption of the number size distribution.

*Referee Comment:* 3) *Why does the aerosol vertical profile of CCN change between the 1985 and 2013? There is no clear explanation given.*

**Author Response:** This is due to the different chemical composition of the aerosol in 1985 and 2013. It was stated that way in the manuscript (p.18, l.13-14). The different scaling factors applied in order to estimate the concentration of the aerosol constituents in the mid 1980's lead to a different chemical composition. The chemical composition is not constant with height and the comparison this comment refers to is done for a vertical velocity of $1 \ m \ s^{-1}$. Therefore, the different aerosols compete for the available water vapor. Hence, the supersaturation is not fixed but depends on the chemical composition of the aerosol. Here, we finally utilize the full parameterization of Abdul-Razzak and Ghan, 2000.

**Change in manuscript:** The respective text in section 3.4 is updated.

*Referee Comment:* 4) *Why was the meteorology from 1985 not taken into account? Is there no meteorology data available from that time period? How can we know that the differences shown in the authors in their simulated 2013 and their simulated 1985 (using meteorology form 2013) would not be different if they actually used meteorology from 1985? The authors even state in their conclusions that the dynamics and thermodynamics (e.g., meteorology) have a large influence on the CCN distribution. Given that the authors treat their 1985 CCN profiles are realistic representations that can be used in future studies and assess the role of emissions reductions between these years, there should be some at least some support, either references or some analysis of the meteorology from these time periods that justify not considering the meteorology.*

**Author Response:** The meteorology of the year 1985 is for sure different from 2013. It was not the goal to describe the real year 1985 but rather to estimate the general aerosol and CCN conditions of the mid 1980's over Central Europe. The reason for keeping the 2013 meteorology is that we did not want to have additional effects due to different meteorological patterns. Instead of simulating many years to rule out meteorological effects, we applied scaling factors to our concentrations of the 2013 simulation. These scaling factors are derived based on the difference of emission estimates of the years 2013 and 1985. Since the concrete emission strength, the location of emitters and other assumptions like size distributions are uncertain for the 1980's in general, this approach is expected to deliver a valid estimate of general mid 1980's CCN concentrations to be included in the mentioned high-resolution simulations.

The way the study is designed, we can be sure to see only effects caused by the altered emission estimates.

**Change in manuscript:** The introduction has been revised, now providing more information on the motivation and goal of this study. The label "1985" is replaced by "mid 1980's". It is made clearer that we did not conduct a separate 1980's simulation, but rather scaled the concentrations of aerosol constituents based on the difference of emission estimates of the years 2013 and 1985.

***Specific Comments / Questions***

***Referee Comment:*** *P1, L8: Can the authors extend their CCN values to include a few decimals points, since the units are $10^9$ $m^{-3}$.*

**Author Response:** Yes.

**Change in manuscript:** "At ground level, average ==values between 0.7 -1.5== x $10^9$ CCN $m^{-3}$ at a supersaturation of 0.2% ==were== found with the different methods."

***Referee Comment:*** *P1, L13-14: 'since chemical composition and the size distribution are less important in these ranges'. This was not shown in this manuscript from my understanding and should be excluded from the abstract.*

**Author Response:** We agree that it is not well formulated here. We meant that at these two supersaturations (0.1 and 0.7%) either almost none or all of the particles are activated, respectively (since we do not consider for Aitken and nucleation mode particles here). However, we now think that it is better to remove the statement from the abstract, and refer to effects of the size distribution assumptions in the text.

**Change in manuscript:** "The discrepancies between model and in-situ observations were lowest for the lowest ==(0.1 %)== and highest supersaturations ==(0.7 %)==."

***Referee Comment:*** *P1, L16: What does 'mid-supersaturation regime' mean? It is suggested the authors use actual numbers here to make it more clear.*

**Author Response:** We agree and have changed the sentence accordingly

**Change in manuscript:** "==For== supersaturation==s between 0.3 % - 0.5 %==, the model overestimated the potentially activated particle fraction by around 30%."

***Referee Comment:*** *P2, L2: It may be a good idea to include the word 'Europe' in the first sentence to make this statement accurate and clear.*

**Author Response:** We agree and have changed the paragraph also according to the next Referee Comment.

**Change in manuscript:** "Compared to today, in the 1980s the ==anthropogenic emission of aerosols and precursor gases such as $SO_2$ in Central Europe== was much higher (Vestreng et al., 2007; Smith et al., 2011). Presumably, during this time the ==loads of such== aerosols ==over this region== were at their maximum. At least since the 1990s anthropogenic ==emissions of aerosols and precursor gases== in Central Europe have been decreasing ==(e.g., Smith et al., 2011)==."

Vestreng, V., Myhre, G., Fagerli, H., Reis, S., and Tarrasón, L.: Twenty-five years of continuous sulphur dioxide emission reduction in Europe, Atmos. Chem. Phys., 7, 3663–3681, https://doi.org/10.5194/acp-7-3663-2007, 2007.

***Referee Comment:*** *P2, L4: Should the last sentence of the first paragraph have a reference, possibly the Cherian et al. 2014 reference?*

**Author Response:** Yes, we agree. The paragraph was revised also according to the Referee Comment above.

**Change in manuscript:** "Compared to today, in the 1980s the ==anthropogenic emission of aerosols and precursor gases such as $SO_2$ in Central Europe== was much higher (Vestreng et

al., 2007; Smith et al., 2011). Presumably, during this time the ==loads of such== aerosols ==over this region== were at their maximum. At least since the 1990s ==anthropogenic emissions of aerosols and precursor gases== in Central Europe have been decreasing ==(e.g., Smith et al., 2011).=="

Vestreng, V., Myhre, G., Fagerli, H., Reis, S., and Tarrasón, L.: Twenty-five years of continuous sulphur dioxide emission reduction in Europe, Atmos. Chem. Phys., 7, 3663–3681, https://doi.org/10.5194/acp-7-3663-2007, 2007.

**Referee Comment:** *P2, L5-6: Aerosol particles play an important role in the microphysical processes of cloud formation, should have a reference. I believe the proper, original reference for this is: Köhler, 1936.*

*Köhler, H. (1936), The nucleus in and the growth of hygroscopic droplets, Trans. Faraday Soc., 32(1152), 1152–1161, doi:10.1039/TF9363201152.*

**Author Response:** Agreed.

**Change in manuscript:** The reference Köhler (1936) was added to the respective sentence.

**Referee Comment:** *P2, L26: The statement says the particle size distribution were calculated using an offline using the Abdul-Razzak and Ghan parameterization. Is this true? I thought the PSDs were assumed, not calculated. Please check.*

**Author Response:** We apologize for ambiguous writing. The reviewer is correct. The size distributions are assumed and applied to the aerosol mass. After this step, the CCN concentration is calculated using the parameterization of Abdul-Razzak and Ghan (2000). We have revised the sentence.

**Change in manuscript:** "Then, the CCN number concentration was calculated offline using the parametrization of Abdul-Razzak and Ghan (2000)==, utilizing assumed== number ==size== distributions ==and the modeled== chemical composition of the aerosol."

**Referee Comment:** *P2, L31: It is suggested that the authors include some earlier, more original studies of CCN observations. For example, Squires and Twomey, 1966.*

*Squires, P., and S. Twomey (1966), A Comparison of Cloud Nucleus Measurements over Central North America and the Caribbean Sea, J. Atmos. Sci., 23(4), 401–404, doi:10.1175/1520-0469(1966)023<0401:acocnm>2.0.co;2.*

**Author Response:** Yes, we agree.

**Change in manuscript:** We added the references Hoppel et al. (1973), Squires and Twomey (1966), and Twomey and Squires (1959) and adapted the paragraph slightly.

Hoppel, W. A., J. E. Dinger, and R. E. Ruskin (1973), VERTICAL PROFILES OF CCN AT VARIOUS GEOGRAPHICAL LOCATIONS, *J. Atmos. Sci.*, *30*(7), 1410-1420, doi:10.1175/1520-0469(1973)030<1410:vpocav>2.0.co;2.

S. Twomey & P. Squires (1959), The Influence of Cloud Nucleus Population on the Microstructure and Stability of Convective Clouds, Tellus, 11:4,408-411, DOI: 10.3402/tellusa.v11i4.9331

**Referee Comment:** *P2, L32-33: Again there are earlier, more original references that should be considered. For example, Feingold et al. 1998.*

*Feingold, G., S. Yang, R. M. Hardesty, and W. R. Cotton, 1998: Retrieving cloud condensation nucleus properties from Doppler cloud radar, microwave radiometer, and lidar. J. Atmos. Oceanic Technol.,*

**Author Response:** We added a few more references and adapted the paragraph slightly.

**Change in manuscript:** "Also the derivation of vertical profiles of CCN with ground-based remote sensing methods is possible (e.g., Ghan et al., 2006; Shinozuka et al., 2015; Mamouri and Ansmann, 2016; Lv et al., 2018) with the development of first approaches in the late 1990's (Feingold et al., 1998)."

Shinozuka, Y., Clarke, A. D., Nenes, A., Jefferson, A., Wood, R.,McNaughton, C. S., Ström, J., Tunved, P., Redemann, J., Thorn-hill, K. L., Moore, R. H., Lathem, T. L., Lin, J. J., and Yoon, Y.J.: The relationship between cloud condensation nuclei (CCN) concentration and light extinction of dried particles: indications of underlying aerosol processes and implications for satellite-based CCN estimates, Atmos. Chem. Phys., 15, 7585–7604,https://doi.org/10.5194/acp-15-7585-2015, 2015.

Lv, M., Wang, Z., Li, Z., Luo, T., Ferrare, R., Liu, D., Wu, D., Mao,J., Wan, B., Zhang, F., and Wang, Y.: Retrieval of cloud condensation nuclei number concentration profiles from lidar extinction and backscatter data, J. Geophys. Res.-Atmos., 123, 6082–6098,https://doi.org/10.1029/2017JD028102, 2018

*Referee Comment: P3, L12: The authors state that the activation of aerosol particles depends on their number, which is not true. The activation of aerosol particles only depends on size, composition, and the amount of supersaturation present.*

**Author Response:** The reviewer is correct. The sentence as it was written in the original manuscript is wrong. We actually meant that the number of activated particles is depending on the physical and chemical properties of the underlying aerosol population, which includes the number size distribution. We have revised the sentence accordingly.

**Change in manuscript:** "Based on the modeled aerosol mass concentrations and assumed particle number size distributions for each aerosol species, the CCN number concentrations were calculated offline using the activation parametrization by Abdul-Razzak and Ghan (2000). The parameterization calculates the number of activated aerosol particles for an aerosol population of multiple lognormal aerosol size distributions and multiple aerosol types. The number of activated aerosol particles depends on the number size distributions of the aerosol population, its chemical composition as well as the applied supersaturation (e.g., fixed or derived from updraft velocities)."

*Referee Comment: P4, L3: Is there then a discontinuity in the simulation data every 48 hours when the model is re-initialized?*

**Author Response:** Yes, there is a usually small discontinuity in the meteorological fields in the center of the domain. The aerosol and trace gas fields are kept for the next cycle and are not re-initialized. Due to hourly update of the chemical and meteorological boundary conditions, near the domain boundaries, no discontinuity is present since the boundaries originate from continuous reanalysis data. Wolke et al. (2012) tested the impact of the cycle length on the resulting concentration fields of aerosol constituents and gas phase chemical compounds. It was found that there is no huge difference between a cycle length of 48 or 96 hours. For COSMO-MUSCAT, the differences between 24- and 48-hours cycles were previously investigated at TROPOS, which showed almost the same result (not published). Therefore, a cycle length of 48 hours presents a good compromise between accuracy and computational costs.

**Change in manuscript:** "After a spin-up phase for COSMO of 24 hours, both models run coupled online for 48 hours. To ensure ==that the meteorology== stay==s== close to the real meteorological conditions, the ==meteorological fields are== then re-initialized for the next simulation cycle. ==The trace gas and aerosol fields are kept from the last time step of the previous cycle to ensure a continuous simulation.=="

*Referee Comment: P5, Figure 1. The Julich site is right next to the model domain boundary. If I understand the model set-up correctly, this is a significant concern since if there are winds coming from the west, the aerosol concentrations in the model could be very inaccurate.*

**Author Response:** The large rectangle in the Fig. 1 shows the approximate location of the domain. This was for simplistic reasons. In the actual domain the site Jülich is located farther away from the western domain boundary. We have exchanged Fig. 1 now showing the real domain. That the domain is not rectangular in this projection is due to that the model is run on a grid with rotated pole.

The model was run in nested mode, i.e. the atmospheric chemical composition was calculated on a coarser European domain first (horizontal resolution of 14km), which for itself is driven by reanalysis data. For sake of simplicity, this information was left out in the original manuscript. We have now revised this paragraph providing the information of the boundary data and generally make the model setup clearer. The inner domain, which was mentioned in the text, is driven by boundary data from this larger domain run. Therefore, reliable chemical composition is provided at the boundaries, which only lacks of fine-scale structure.

**Change in manuscript:** Updated Figure 1.

"The model domain investigated in this study is displayed in Fig. 1 and covers the area between 6-15°E and 48.25-54°N. The horizontal resolution was set to 7 km. In the vertical, the model treats 50 layers up to a height of 22 km. ==As lateral boundary conditions for the trace gases and aerosol species, modeled fields of atmospheric chemical composition originating from a coarser simulation on a European domain are utilized. This coarser surrounding simulation is driven by reanalysis data for meteorology (reanalysis product of DWD using the GME model) and atmospheric chemical composition (CAMS reanalysis product, Innes et al., 2019).=="

Inness, A., Ades, M., Agustí-Panareda, A., Barré, J., Benedictow, A., Blechschmidt, A.-M., Dominguez, J. J., Engelen, R., Eskes, H., Flemming, J., Huijnen, V., Jones, L., Kipling, Z., Massart, S., Parrington, M., Peuch, V.-H., Razinger, M., Remy, S., Schulz, M., and Suttie, M.: The CAMS reanalysis of atmospheric composition, Atmos. Chem. Phys., 19, 3515–3556, https://doi.org/10.5194/acp-19-3515-2019, 2019.

*Referee Comment: P5, L10: The authors state that particles between 50 and 200 nm are most relevant for estimating CCN. This is not strictly true since all aerosol particles can form CCN. Can the authors be accurate with their statements? For example, maybe here the authors mean that particles between 50 and 200nm are most relevant for estimating CCN for supersaturations between some supersaturation range? For shallow clouds with very low supersaturations, the sizes of CCN particles that are relevant may indeed be larger. Similarly, for high supersurations, the relevant particle sizes may be lower.*

**Author Response:** The reviewer is right. We had the investigated supersaturation range in mind. In this supersaturation range, the critical size of activation is often located in the size range of 50 – 200 nm in diameter. Furthermore, in this range particles are usually more numerous than in larger sizes. We wanted to express that, for the given supersaturation range, an underestimation of the number of much larger particles and much smaller particles does

usually not influence the CCN number concentrations substantially since they are either small in number (large particles) or are not activated (small particles).

**Change in manuscript:** "The calculations have been compared to observational data and showed a good agreement to the observed total size distribution at Melpitz between 50 and 200 nm (Hande et al., 2016), which is a very relevant size range for estimating CCN in the supersaturation range investigated in this study (0.1 – 0.7 %)."

*Referee Comment:* P6, L10: What does "as written above" refer to? I found this paragraph difficult to follow and suggest the authors revise it to make it more clear.

**Author Response:** The statement refers to the section on p.5, l.8-13, which describe the paper by Hande et al., 2016. We agree that the reader benefits from a revision of this paragraph.

**Change in manuscript:** The respective paragraph is revised.

*Referee Comment:* Table 1: Instead of putting the references in the caption, can the authors put the relevant references next to each species? This would allow a reader who was interested in understanding one of the species assumptions to easily find that without having to look through all of the references listed in the caption.

**Author Response:** We agree. However, putting the references in the table would suggest to the reader that the references also refer to the other parameters. Instead, we have now included the species name to the respective reference in the table caption.

**Change in manuscript:** Updated caption of Table 1.

"Several laboratory and model studies served as basis for the κ values used in this study (ammonium sulfate: Ghan et al. (2001), Petters and Kreidenweis (2007); ammonium nitrate: Duplissy et al. (2011); sulfate: Petters and Kreidenweis (2007); OC: Ghan et al. (2001), Wex et al. (2009); sea salt, dust, EC: Ghan et al. (2001))."

*Referee Comment:* P11, L30: What causes the significant differences (5-8x different) in OC, SS and DU between the 1985 and 2013 modeled values? Since the emissions do not change, it would be very interesting to understand why these other species change and provide better interpretation of these results.

**Author Response:** The concentration of OC, dust and sea salt did not change. Table 4 presents their relative contribution to the CCN number concentration. Due the higher emission of $SO_2$ in 1985 the concentration of ammonium sulfate and sulfate is much higher and hence the CCN provided by these species. Therefore, the relative contribution to the total CCN number of the unchanged components OC, dust and sea salt is reduced.

*Referee Comment:* P12, L7-9: What does 'large' and 'major' mean? Can the authors be more specific and include numbers?

**Author Response:** Poulain et al. (2011) did not report SOA concentrations but concluded from their investigations of chemical properties of the organic aerosol that in summer at Melpitz "the organic particulate matter seemed to be heavily influenced by regional secondary formation". Overall, organic matter is the most important particulate fraction in PM1 at Melpitz in summer (59 %).

**Change in manuscript:** "SOA generally can contribute a large fraction to the total concentration of organic aerosol mass with an average contribution over Europe ranging from

~20 % to more than 50 % (Jimenez et al., (2009). Also at Melpitz in summer, organic matter is the major fraction (59 %) of the PM1 aerosol and is strongly influenced by SOA (Poulain et al., 2011)."

**Referee Comment:** *P12, L18: Why were ammonium sulfate and nitrate underestimated in the model in the first half of spring?*

**Author Response:** The exact reasons are not known. The concentration of ammonium nitrate in agricultural regions is depending on the available ammonia. The ammonia emissions are in the short term uncertain since the exact timing of bringing out manure is usually not known. Hence, in particular the magnitude and timing of observed ammonium nitrate and sulfate concentration patterns cannot be expected to match by applying ammonia emission databases (time variation covers only the general seasonal cycle). Other potential causes of uncertainty such as precipitation did not show obvious discrepancies during the first half of the spring episode.

**Referee Comment:** *L12, L19: It is unclear what the "factor of 5" is referring to. I am assuming CCN concentrations, but the way it is written it sounds like ammonium nitrate or ammonium sulfate or both?*

**Author Response:** It refers to the concentration of ammonium nitrate, where we see the strongest deviation from the observed aerosol mass. Therefore, the statement is put in brackets behind ammonium nitrate only. We have separated the information about ammonium nitrate into its own sentence to avoid misunderstanding.

**Change in manuscript:** "However, during the fall period the model often overestimates the concentration of ammonium sulfate and ammonium nitrate and hence the CCN concentrations. In particular ammonium nitrate is sometimes strongly overestimated by up to factor of 5."

**Referee Comment:** *P13, L1-5: The authors speculate that the overestimation in the modeled CCN may be due to ammonium nitrate uncertainties. However, then they state that actually for a two day period it was due to differences in precipitation and wet deposition. What is the primary cause for the consistent overestimation?*

**Author Response:** The respective paragraph was meant to discuss the fall period in Fig. 2, which actually presents the CCN calculated from the different compounds using the modeled and the measured aerosol masses and the same activation parameterization. It can be seen that both ammonium nitrate and ammonium sulfate are overestimated (compare Fig. 2b (model) and d (chemical aerosol measurements)).

The 3-day episode was not supposed to be investigated in detail in this study, since it does not give hints on the overall performance during the whole period covering several months. We mentioned the short episode in the text since the exact location of such small-scale low pressure systems and of the formed precipitation is not controlled by the driving boundary data, but develops due to the model physics. The analysis causes leading to uncertainties of the meteorological model for this selected short period is beyond the scope of the study. The 3-day period is meteorologically an exception and the overall performance of the reanalysis-driven meteorological model is mostly satisfactory.

The lack of wet deposition due to missing precipitation in North-western Germany and The Netherlands could explain at least part of the overestimation (see also Figure at comment 1) in the section "Overarching Concerns"). For the three days, the ammonium nitrate concentration is stronger overestimated (factor of 2-3) than ammonium sulfate (< factor of 2).

Assuming the overestimation of ammonium sulfate is due to the missing wet deposition, would imply still ~50% overestimation in ammonium nitrate. However, this is also very speculative. Since the true emission events are not known, it is not possible to reliably quantify the effect that arises from overestimation of the emissions. Nevertheless, the chemical observations show the same peak in ammonium nitrate and ammonium sulfate, but with smaller magnitude. This implies that the general pattern of emission, formation, and transport is represented well.

The potential uncertainties of the observations (temperature bias) and modelling (unknown timing of ammonia emissions) of ammonium nitrate were given in the manuscript since we know that these issues exist. However, the magnitude of these effects is not known. Therefore, the primary cause of the overestimation is not known. Since the overestimation is primarily seen for ammonium nitrate, uncertainties in the emissions seems likely, but is also speculative. We therefore intended to only mention potential uncertainties in the manuscript.

**Change in manuscript:** The section about the 3-day period has been revised including more discussion on the potential causes.

*Referee Comment: Figure 2c,d: There is very little discussion in the manuscript about the comparisons of the CCN based on the gravimetric observations as compared to the CCNC, which is shown in Figure 2c,d.*

**Author Response:** We agree that a statement on the comparison between CCN from the gravimetrical measurements and in-situ CCN measurements improves the discussion of Fig. 2. The discussion was initially meant to focus on the performance of the model against the in-situ observations and against the CCN estimated based on the gravitational measurements.

**Change in manuscript:** The discussion of Fig. 2 is revised.

*Referee Comment: P16, L2: "More important" than what?*

**Author Response:** This sentence was unclear and not a valid conclusion from Fig. 5.

**Change in manuscript:** The sentence is deleted in the revised manuscript and the whole paragraph is revised.

*Referee Comment: P16, L12: "observed and measured" -- Are these meaning the same thing?*

**Author Response:** We are sorry for this typo with strong confusing effect. Instead of "measured" we mean "modeled". This is changed accordingly in the manuscript.

**Change in manuscript:** "Above, the observed and modeled CCN concentrations start to decrease considerably, ..."

*Referee Comment: P18, L4: The authors state that for this analysis a vertical velocity of 1 ms$^{-1}$ was assumed. All the prior analyses had fixed supersaturations, while for this analysis, the authors instead used a fixed vertical velocity. Why did this change?*

**Author Response:** This was done, since the partner study that used the CCN fields in high-resolution simulations required CCN concentrations for a set of different vertical velocities. Furthermore, the usage of the vertical velocity in order to derive the number of activated particles is what would be done when the activation parameterization is applied to cloud droplet activation in a simulation with cloud interactive aerosols. Here, we finally utilize the full

parameterization of Abdul-Razzak and Ghan (2000). In contrast to a fixed supersaturation, the usage of vertical velocity leads to a competition between the aerosol particles for the available water vapor and, hence, a variable supersaturation which depends also on the aerosol chemical composition.

**Change in manuscript:** In the revised manuscript, the following sentence was added:

"For the calculation, a vertical velocity of 1 ms$^{-1}$ was assumed. This is an example for the CCN fields that are required as input for the ICON-LEM simulations within the HD(CP)$^2$ project."

**Referee Comment:** *P18, L4-5: The authors state that "in contrast to the previous analysis, the supersaturation depends on the aerosol composition and varies spatially and temporally." I think additional statements are needed here or upfront in the manuscript to make this more clear.*

**Author Response:** Assuming a fixed vertical velocity, instead of fixed supersaturation, leads to variable supersaturation, which depends on the current size-resolved chemical composition of the aerosol population. Since the size distribution is fixed in our case, the supersaturation is still varying with chemical composition.

**Change in manuscript:** "In contrast to the previous analysis, the applied supersaturation, and hence the critical size of activation, is not fixed but now result from the competition of the aerosol particles for the available water vapor. Therefore, the supersaturation and the critical size of activation depends on the aerosol composition, due to the different hygroscopicities and assumed size distributions of the different aerosol species, and therefore varies temporally and spatially."

**Referee Comment:** *P18, L13-14: Why is the difference in shape of the profiles due to differences in aerosol composition (hygroscopicity)? How was this determined, and how did the authors rule out other differences (i.e., differences in aerosol chemistry, differences in meteorology)? There should be much more explanation here.*

**Author Response:** Differences in meteorology can be ruled out since only the 2013 scenario was simulated. The 1985 concentrations were estimated from the 2013 concentrations (see Table 3) based on the scaling factors. These are derived from the difference in countrywide emissions between 1985 and 2013. Moreover, model assumptions on aerosol chemistry, i.e., how the model forms ammonium nitrate and ammonium sulfate aerosol, were used to derive the scaling factors. The procedure is explained in detail in section 2.2.2. Therefore, only different aerosol composition can be the reason for the different shape of the vertical profiles. That the contribution from different species to the total CCN number is different between the 1980's and 2013 case is also shown in Table 4.

Since the chemical composition is not constant with height and the comparison this comment refers to is done for a vertical velocity of 1 m s$^{-1}$. Therefore, the different aerosols compete for the available water vapor. Hence, the supersaturation and the critical size of activation are not fixed but depend on the chemical composition of the aerosol.

**Change in manuscript:** Due to a different vertical distribution of the aerosol constituents, the aerosol composition and, hence, aerosol hygroscopicity deviates between mid 1980's and 2013. Therefore, and since Fig. 7 presents the CCN concentration for a fixed vertical velocity leading to variable supersaturations, the shape of the CCN profiles in the two scenarios differs.

**Referee Comment:** *P18, L15. What do the authors mean when they say the scaling factor represents a mean trend? Are they suggesting there is a linear decrease between the two years? I am not sure the authors can state anything about a mean trend with two points.*

**Author Response:** We agree that the term "mean temporal trend" is misleading. We mean that the derived scaling factor could be used to roughly scale CCN concentrations between the 2010's and the 1980's over Europe.

**Change in manuscript:** "This scaling factor describes the ==difference in== CCN number concentration between ==the== past peak aerosol in the 1980s and present-day conditions in Europe and ==is== useful for sensitivity studies."

**Referee Comment:** *P20, L18-19: In their conclusions, the authors state that the thermodynamics and dynamics of the tropopause has a large influence on distribution of aerosol and the vertical profile of CCN. However, the authors state in their manuscript that this was due to aerosol composition hygroscopicity (P18, L13-14). Can the authors clarify what is meant?*

**Author Response:** We mean that, under the assumption that vertical transport (as part of the atmospheric dynamics) of aerosol between 1980's and today does not change generally, strong concentration changes of CCN due to altered emission are estimated also in the lower free troposphere. However, we agree that the sentence is actually misleading and have replaced it.

**Change in manuscript:** "The scaling factor for estimating the CCN concentrations during the 1980s from current simulations is not vertically homogeneous. Close to the ground, a scaling factor of 2 was determined, increasing to 3.5 between 2 and 5 km height. Towards the upper troposphere at around 8 km height, the scaling factor decreases again to 1. ==The vertical variability of the CCN scaling factor is caused by the changed chemical composition of the aerosol due to the 1980's emission estimates.=="

**Referee Comment:** *P20, L19-20: One of the main results is the differences in the vertical structure of CCN between 1985 and 2013, and the authors state that this vertical structure up to 5km is important for cloud microphysical processes without any references. The two studies that have assessed this are Lebo 2014 and Marinescu et al. 2017 and should be included to support this statement.*

*Lebo, Z. J. (2014), The Sensitivity of a Numerically Simulated Idealized Squall Line to the Vertical Distribution of Aerosols, J. Atmos. Sci. , 71 (12), 4581–4596, doi:10.1175/JAS-D-14-0068.1.*

*Marinescu, P. J., S. C. van den Heever, S. M. Saleeby, S. M. Kreidenweis, and P. J. DeMott (2017), The Microphysical Roles of Lower-Tropospheric versus Midtropospheric Aerosol Particles in Mature-Stage MCS Precipitation, J. Atmos. Sci. , 74 (11), 3657–3678, doi:10.1175/JAS-D-16-0361.1.*

**Author Response:** Thank you for these suggestions. The two references are now included in the revised manuscript.

**Change in manuscript:** Especially the height range of up to 5km, where a very high CCN number concentration during the 1980s was found, is important for cloud and precipitation formation in the mid-latitudes ==(e.g., Lebo, 2014; Marinescu et al., 2017)==.

**Referee Comment:** *Figures 6 and 7. Can the authors make these figures wider, so that readers can more easily see the values that are represented by the lines? Currently, it is very hard to see.*

**Author Response:** Yes, we have made the figures wider, and also increased the font size of the figure text.

**Change in manuscript:** Updated Figures 6 and 7.

[revised manuscript text omitted]